# Influence of infrastructure on water quality and greenhouse gas dynamics in urban streams

Rose M. Smith[1,2], Sujay S. Kaushal[2], Jake J. Beaulieu[3], Michael J. Pennino[4,5], Claire Welty[5]

[1]Department of Biology, University of Utah, Salt Lake City, UT, 84112, USA

[2]Department of Geology & Earth System Science Interdisciplinary Center, University of Maryland, College Park, MD, 20742, USA

[3]U.S. Environmental Protection Agency, Office of Research and Development, National Risk Management Research Laboratory, Cincinnati, OH, 45220, USA

[4]U.S. Environmental Protection Agency National Health and Environmental Effects Research Lab, Corvallis, OR, 97333, USA

[5]Department of Chemical, Biochemical, and Environmental Engineering and Center for Urban Environmental Research and Education, University of Maryland Baltimore County, Baltimore, MD, 21250, USA

*Correspondence to*: Rose M. Smith (rose.smith@utah.edu)

**Abstract.** Streams and rivers are significant sources of nitrous oxide ($N_2O$), carbon dioxide ($CO_2$), and methane ($CH_4$) globally,

and watershed management can alter greenhouse gas (GHG) emissions from streams. We hypothesized that urban infrastructure significantly alters downstream water quality and contributes to variability in GHG saturation and emissions. We measured gas saturation and estimated emission rates in headwaters of two urban stream networks (Red Run and Dead Run) of the Baltimore Ecosystem Study Long-Term Ecological Research Project.. We identified four combinations of stormwater and sanitary infrastructure present in these watersheds, including: 1) stream burial, 2) inline stormwater wetlands, 3) riparian/ floodplain

preservation, and 4) septic systems. We selected two first-order catchments in by each of these categories, and measured GHG concentrations, emissions, and dissolved inorganic and organic carbon and nutrient concentrations bi-weekly for one year. From a water quality perspective, the DOC: $NO_3^-$ ratio of streamwater was significantly different across infrastructure categories. Multiple linear regressions including DOC: $NO_3^-$ and other variables (DO, TDN, and temperature) explained much of the statistical variation in nitrous oxide ($N_2O$, $r^2= 0.78$), carbon dioxide ($CO_2$, $r^2=0.78$), and methane ($CH_4$, $r^2=0.50$) saturation in

stream water. We measured $N_2O$ saturation ratios, which were among the highest reported in the literature for streams, ranging from 1.1 - 47 across all sites and dates. $N_2O$ saturation ratios were highest in streams draining watersheds with septic systems and strongly correlated with TDN. $CO_2$ saturation ratio was highly correlated with $N_2O$ saturation ratio across all sites and dates, and $CO_2$ saturation ratio ranged from 1.1 to 73. $CH_4$ was always super-saturated with saturation ratios ranging from 3.0 to 2,157. Longitudinal surveys extending form headwaters to third order outlets of Red Run and Dead Run took place in spring and fall.

Linear regressions of this data yielded significant negative relationships between each gas with increasing watershed size, as well as consistent relationships between solutes (TDN or DOC, and DOC: TDN ratio) and gas saturation. Despite a decline in gas saturation between the headwaters and stream outlet, streams remained saturated with GHGs throughout the drainage network, suggesting that urban streams are continuous sources of $CO_2$, $CH_4$, and $N_2O$. Our results suggest that infrastructure decisions can have significant effects on downstream water quality and greenhouse gases, and watershed management strategies may need to consider coupled impacts on urban water and air quality.

**Key Words**: Greenhouse Gases, Urban Streams, Infrastructure, DOC, Nitrate, Methane, Carbon Dioxide, Methane, Nitrous Oxide

## 1 Introduction

Streams and rivers are dynamic networks that emit globally significant quantities of $CO_2$, $CH_4$ and $N_2O$ to the atmosphere. $CO_2$ emissions via flowing waters are equivalent to half of the annual terrestrial carbon sink (1.2 Pg $CO_2$-C $yr^{-1}$, Cole et al. 2007; Battin et al. 2009). Stanley et al. (2016) recently demonstrated that flowing waters are significant $CH_4$ sources as well, emitting approximately 28 Tg $yr^{-1}$, which is equivalent to between 10 and 35% of emissions from wetlands globally (Bridgham et al. 2013). Approximately 10% of global anthropogenic $N_2O$ emissions are emitted from river networks due to nitrogen contamination of surface and groundwater (UNEP 2013; Ciais et al. 2013). There is evidence that these $N_2O$ estimates, based on IPCC guidelines, might be too low, given growing evidence of high denitrification rates in small streams with high $NO_3^-$ loads (Beaulieu et al. 2011).

While much of the research on GHG emissions from streams has taken place in agricultural watersheds, urban-impacted river networks receive similar N loads and have also shown elevated GHG concentrations and emissions (e.g. Daniel et al. 2001; Beaulieu et al. 2010; Beaulieu et al. 2011; Kaushal et al. 2014; Gallo et al. 2014). As urban land cover and populations continue to expand, it is critical to understand the impacts on downstream waters, including C and N loading and GHG emissions. While $N_2O$ emissions from both urban and agricultural sources are taken into account in models based on estimated watershed DIN loading (Nevison et al. 2000; Seitzinger et al. 2000), measurements validating these estimates or estimates of $CO_2$ and $CH_4$ in urban watersheds are rare. Quantifying the variability, drivers, and sources of GHG emissions from streams will illuminate the biogeochemical processes and potential role of urban infrastructure on nutrient cycling, water quality, and GHG budgets.

## 1.2 Role of sanitary infrastructure

The form and age of stormwater and sanitary infrastructure within a watershed can influence stream water GHG emissions in several ways. GHGs may enter urban streams directly through buried stormwater and sanitary infrastructure, or form increased production within streams in response to nutrient loading and/or geomorphic changes. We investigated the role of infrastructure on GHG emissions from streams in order to evaluate these potential drivers of heterogeneity within urban watersheds. Sanitary infrastructure encompasses a wide array of systems to manage human waste. In developed countries, sanitary infrastructure includes a combination of septic systems, sanitary sewers, and sometimes, combined stormwater/sanitary sewers. Storm and sanitary sewer lines are present in areas with medium-to-high density development. Sanitary sewer or combined sewer network delivers waste to centralized wastewater treatment plants (WWTPs), which treat influent and release effluent into larger rivers or coastal zones. Sanitary, storm, and combined sewers tend to follow stream valleys (i.e. low points in the landscape), are often made of erodible materials such as terra cotta or concrete, and tend to crack or develop leaks. Leaks in sanitary sewer infrastructure can lead to chronic nutrient loading throughout stream networks (Divers et al. 2013, Kaushal et al. 2011; Pennino et al. 2016; Kaushal et al. 2015). Septic systems, primarily used in low-density residential areas, are designed to settle out waste solids and leach N-rich liquid waste into subsurface soils and groundwater. Sanitary sewer infrastructure may influence GHG abundance and emission from streams directly via diffusion of gases out of gravity sewer lines (Short et al. 2014), or indirectly by microbial processing along surface and subsurface flowpaths (Yu et al. 2013; Beaulieu et al. 2011). While the present study focuses mainly on first to third-order streams influenced by sanitary sewer lines or septic systems, it is also worth mentioning that WWTPs are known to be a source of $CH_4$ and $N_2O$ in urban areas, and contribute point-source GHG loading to larger rivers and coastal areas (Beaulieu et al. 2010; Strokal and Kroeze 2014; Alshboul et al. 2016).

Sewage leaks are likely the primary source of $N_2O$ emissions from small urban streams (Short et al. 2014). Several studies have documented that wastewater leakage from municipal sewers often accounts for more than 50% of dissolved N in urban streams (Kaushal et al. 2011; Pennino et al. 2016; Divers et al. 2013). While sanitary sewer lines are known to leak dissolved N, $N_2O$ losses are not accounted for in greenhouse gas budgets of large WWTPs that these pipes feed into. Short et al. (2014) measured intake lines from three municipal WWTPs and estimated that $N_2O$ emissions from sewer lines alone on the same order of magnitude (1.7g $N_2O$ person $yr^{-1}$) as current IPCC estimates for per-capita emissions from secondary WWTPs. Their study demonstrates the importance of constraining biogenic gas emissions from streams, which flow alongside and may receive gaseous inputs from aging sanitary sewer lines.

**1.3 Role of stormwater infrastructure**

Stormwater infrastructure varies widely across and within cities. From stream burial in pipes to infiltration-based green infrastructure (GI) designs, stormwater management designs have evolved over time (Collins et al. 2010, Kaushal et al. 2014). In Baltimore, where this study took place, stormwater management installed prior to the 1970s consisted of concrete-lined channels and buried streams (Baltimore County Department of Planning, 2010). Areas developed during the 1990s and 2000s are characterized by a more GI-based design approach, including but not limited to upland detention ponds, infiltration basins, wetlands and bio-swales. Stream restoration projects and riparian zone protections have also been established, restricting development within 100m of the stream corridor for new developments (Baltimore Department of planning, 2010).

The form of stormwater infrastructure – whether stream burial, infiltration wetland, or restored riparian zone – may contribute to GHG saturation of groundwater and streams. Stormwater-control wetlands and riparian/floodplain preservation may increase or decrease $CH_4$ and $N_2O$ emissions from streams, depending upon how watershed C and N inputs are routed along hydro-biogeochemical flowpaths. For instance, if these forms of GI are successful at removing excess N inputs to streams, GI may reduce $N_2O$ emissions from flowing waters. Alternatively, GI may increase both $N_2O$ and $CH_4$ inputs to streams and thus emissions by facilitating anaerobic microbial metabolism (Søvik et al. 2006; VanderZaag et al. 2010). The form of GI (i.e. stormwater control wetland vs. riparian/floodplain preservation) may also influence GHGs due to 1) differences in water residence time and oxygen depletion in wetland vs. floodplain soils, and 2) differences in watershed-scale N removal capacity of the two different approaches.

**1.4 Variables controlling GHG production in urban watersheds**

Reach-scale studies in streams across biomes have demonstrated that GHG production and emission is sensitive to changes in nutrient stoichiometry, organic matter quality, redox state, and temperature (e.g. Bernot et al. 2010; Kaushal et al. 2014a; Beaulieu et al. 2009; Dinsmore et al. 2009; Baulch et al. 2011; Harrison and Matson 2003). Several studies have shown that infrastructure can influence solute loading and stoichiometry of streams, which could in turn increase GHG production. For instance, Newcomer et al. (2012) measured higher rates of N uptake and denitrification potential in streams with restored riparian zones compared with degraded, incised urban streams. In-stream N uptake is also consistently higher in daylighted streams compared with streams buried in pipes (Pennino et al. 2014; Beaulieu et al. 2015). Upland or inline stormwater wetlands and retention ponds provide additional locations for focused N removal in urban watersheds (Newcomer et al. 2014; Bettez et al.

2012). Sanitary infrastructure (i.e. leaky sewer lines and septic systems) can also be a source of N via leaching into groundwater (Shields et al. 2008; Kaushal et al. 2015; Pennino et al. 2016).

In previous studies, carbon quantity and/or organic matter quality was correlated with N uptake or removal in urban streams and wetlands (Newcomer et al. 2012; Pennino et al. 2014; Beaulieu et al. 2015; Bettez et al. 2012; Kaushal et al. 2014). Inverse relationships between dissolved organic carbon (DOC) and nitrate ($NO_3^-$) concentrations have been found to persist across a wide variety of ecosystems ranging from soils to streams to oceans (e.g., Aitkenhead-Peterson and McDowell 2000; Dodds et al. 2004; Kaushal and Lewis 2005; Taylor and Townsend 2010). Recently, inverse relationships between DOC and $NO_3^-$ have also been reported for urban environments ranging from groundwater to streams to river networks (Mayer et al. 2010; Kaushal and Belt 2012; Kaushal et al. 2014a). A suite of competing biotic process may control this relationship, by either 1) assimilating or reducing $NO_3^-$ in the presence of bioavailable DOC, or 2) producing $NO_3^-$ regardless of DOC status (Hedin et al. 1998; Dodds et al. 2004; Kaushal and Lewis 2005; Taylor and Townsend 2010). The former category includes heterotrophic denitrification, which oxidizes organic carbon to $CO_2$ and reduces $NO_3^-$ to $N_2O + N_2$ (Knowles, 1982), and assimilation of inorganic N (Wymore et al. 2015; Caraco et al. 1998; Kaushal and Lewis 2005). In the second category, nitrification chemoautotrophically produces $NO_3^-$ by oxidizing $NH_4^+$, and consuming $CO_2$. Nitrification also yields $N_2O$ as an intermediate product, and has been shown to dominate N cycling processes in low-DOC environments (Schlesinger 1997; Taylor and Townsend, 2010; Helton et al. 2015).

In urban watersheds, denitrification is often limited by DOC due to increased N loading and/or decreased connectivity with carbon-rich soils in the riparian zone (Mayer et al. 2010; Newcomer et al. 2012). C:N stoichiometry are likely to be affected by stormwater and sanitary sewer infrastructure designs as well (Søvik et al. 2006; Collins et al. 2010; Kaushal et al. 2011). Stormwater wetlands may promote anoxic conditions and increase C:N ratio of stream water by increasing flow through carbon-rich soils (e.g. Søvik et al. 2006; Newcomer et al. 2012). Stream burial can reduce C:N ratios, if streams are buried in storm drains (Pennino et al. 2016; Beaulieu et al. 2014). Leaky sanitary infrastructure may additionally reduce the C:N ratio, and/or alter the form of carbon in streams (Newcomer et al. 2012).

**1.5 Study goals**

The goal of the present study was to identify patterns and drivers related to GHG dynamics in urban headwater streams draining different forms of infrastructure (stream burial, septic systems, inline SWM wetlands and riparian/floodplain preservation). Although less considered compared with nutrient loading, increased GHG emissions may be an unintended consequence of

urban water quality impairments and biogeochemical processes occurring within and downstream of urban infrastructure. A growing body of work has shown that nutrient and carbon loads to streams are related not only to land cover metrics (% impervious surface, urban density, etc.) but also urban infrastructure (Shields et al. 2008; Kaushal et al. 2014). Connectivity between runoff-generating water sources (groundwater, overland flow, shallow subsurface flow) and urban infrastructure (sanitary sewer lines, storm sewers, drinking water pipes, constructed wetlands, etc.) is likely to influence nutrient export and biogeochemical function of waterways. An improved understanding of the relationship between infrastructure type and biogeochemical functions is critical for minimizing unintended consequences of water quality management, especially as growing urban populations place greater burden on watershed infrastructure (Doyle et al. 2009; Foley et al. 2005; Strokal and Kroeze 2014).

## 2.1 Sampling Methods

### 2.1.1 Study Sites

This study took place in collaboration with the Baltimore Ecosystem Study Long-Term Ecological Research (LTER) project (www.beslter.org). We identified four categories based on distinct combinations of stormwater and sanitary infrastructure dominating the greater Baltimore region, based on maps of stormwater control structures, housing age, and intensive field surveys. We then selected eight first-order streams paired across the four categories. The first order stream sites each were located in half in Red Run and half in Dead Run, sub-watersheds of the Gwynns Falls (Fig. 1). We have abbreviated the categories based on the dominant infrastructure feature as follows: 1) stream burial, 2) inline stormwater management (SWM) wetlands, 3) riparian/floodplain preservation, and 4) septic systems (Table 1).

Sites in the 'stream burial' category (DRAL and DIRS) drain watersheds with streams contained in storm sewers. Sanitary infrastructure in these watersheds is composed of aged sanitary sewer lines, installed prior to 1970 (Baltimore County Department of Planning, 2010). Streams in the 'inline stormwater management' category (DRKV and DRGG) originate in stormwater ponds or wetlands and also flow adjacent to aging sanitary sewer lines. Streams in the 'riparian/floodplain preservation' category (RRRM, RRSM) drain watersheds with newer development (after 2000), upland infiltration wetlands, and 100 m wide undeveloped floodplains (Baltimore County Department of Planning, 2010). Sanitary sewers were constructed in these watersheds between 2000 and 2010 (Baltimore County Department of Planning, 2010). Sites in the 'septic systems'

category (RRSM, RRSD) drain lower density development with stormwater management in the form of stormwater sewer pipes (Fig. 1). All eight first order stream sites were sampled every two weeks for dissolved carbon and nitrogen concentrations.

### 2.1.2 Temporal Sampling of Dissolved Gases and Stream Chemistry

Headwater stream sites were sampled every two weeks for solutes (DOC, TDN, HIX, BIX) and dissolved gas ($CO_2$, $CH_4$ and $N_2O$) concentrations. Chemistry sampling took place for two years, between January 2013 and December 2014, and gas sampling took place between July 2013 and July 2014. Sites were visited between the hours of 9 AM and 2 PM. Five dissolved gas samples were collected per stream on each date, along an established 20 m study reach either upstream adjacent to the gaging station. Gas samples were collected at 0, 5, 10, 15, and 20 m from the fixed starting point of the study reach. Samples were collected by submerging a 140 mL syringe with a 3-way luer-lock and pulling 115 mL of stream water into the syringe. We added 25 mL of ultra-high purity helium to the syringe in the field, then shook syringes vigorously for 5 minutes to promote equilibration of gases between aqueous and gas phases. After equilibration, 20 mL of the headspace was immediately transferred into a pre-evacuated glass vial capped with screw-top rubber septum (LabCo Limited, Lampeter, UK), then transported to the laboratory, where samples were stored at room temperature for up to four weeks prior to analyses. Water temperature and barometric pressure during the equilibration were recorded in the field. We collected three helium headspace blanks by injecting 25mL of helium into pre-evacuated vials in the field.

We collected stream water samples in a 250 mL high-density polyethylene bottles, one sample per site. One sample duplicate sample was collected on each sampling date, and the site for duplicate sample collection rotated among sampling dates. Dissolved oxygen (DO) concentration and pH were measured at the upstream end of each study reach using a handheld YSI 550-A dissolved oxygen meter (YSI Inc. Yellow Springs, OH) and an Oakton handheld pH meter (Oakton Instruments, Vernon Hills, IL).

### 2.1.3 Longitudinal Sampling of Dissolved Gases

Longitudinal surveys were conducted in June 2012, March 2014, and December 2014 in Red Run and Dead Run. Longitudinal sampling started at the outlet of each major tributary (Dead Run or Red Run), and extended every 500 m upstream to include the four bi-weekly sampled headwater sites in each watershed (Fig. 1). During spring and fall months, solute and gas samples were collected along all major tributaries (>5% main stem flow) as well as every 500 m along the main stem of Dead Run and Red Run. Minor tributaries (< 5% of main stem flow) were not sampled. Stream discharge was measured at each sampling point

using a Marsh- McBirney Flo-Mate hand held velocity meter (Marsh McBirney Inc., Frederick, MD, USA). We used cross-sectional measurements of stream velocity and water depth to calculate instantaneous discharge at each sampling site. We measured velocity and depth at a minimum of 10 points at each cross section in order to properly characterize flow across the channel. Discharge data was provided by USGS when sampling sites were co-located with a USGS gaging station (U.S. Geological Survey 2017).

We calculated the watershed contributing area above each sampling point and flow length from each sampling point to the watershed outlet using Hydrology toolbox in ArcMap 10. Sampling locations were designated pour points in the hydrology tools workflow. Because sampling points were always co-located with road crossings, we were able to acquire the latitude and longitude of sampling sites using Google Earth software (Google Inc. 2009). Watersheds were delineated using a 2-m resolution DEM (Baltimore County Government, 2002). We first corrected the DEM for spurious depressions using the "Fill" tool in the ArcMap10.0 hydrology toolbox. Next, we calculated flow direction for each pixel of this filled DEM raster. We then used the Flow Accumulation tool to evaluate the number of pixels contributing to each downstream pixel. After ensuring that each pout point was co-located on the map streams (i.e. areas with flow accumulation > 500 pixels), we used the 'Watershed' tool to delineate the pixels draining into each sampled location.

## 2.2  Laboratory Methods

### 2.2.1 Dissolved Gas Concentrations

Samples of headspace equilibrated gas concentrations ($CO_2$, $CH_4$, and $N_2O$) were stored at room temperature for up to 1 month in airtight exetainer vials and transported to the EPA National Risk Management Research Laboratory, Cincinnati, Ohio for analysis. Concentrations of $CO_2$, $CH_4$, and $N_2O$ were measured using a Bruker 450 (Billerica, MA, U.S.A) gas chromatograph equipped with a methanizer, flame ionization detector (FID), and electron capture detector (ECD). Instrument detection limits were 100 ppb for $N_2O$, 10 ppm for $CO_2$, and 0.1 ppm for $CH_4$.

### 2.2.2 Solute Concentrations

Water samples were transported on ice to the University of Maryland, College Park and filtered using pre-combusted 0.7 μm glass fiber filters within 24 hours. A Shimadzu TOC analyzer (Shimadzu Scientific, Kyoto Japan) was used to measure total dissolved nitrogen (TDN) and dissolved organic carbon (DOC). The non-purgeable organic carbon (NPOC) method was utilized for DOC, despite potential underestimation of volatile compounds because the NPOC method is insensitive to variations in DIC

(Findlay et al. 2010). TDN was measured on the same instrument using the 'TDN' method, which consists of high temperature combustion in the presence of a platinum catalyst. Nitrate ($NO_3^-$) concentrations were measured via colorimetric reaction using a cadmium reduction column (Lachat method 10-107- 04-1-A) on a Lachat flow injection analyzer (Hach, Loveland, CO).

### 2.2.3. DOM Characterization

Filtered water samples were analyzed for optical properties in order to characterize dissolved organic matter (DOM) sources. After filtering (0.7 µm GF/F), samples were stored in amber glass vials at 4ºC for a maximum of two weeks prior to analyses. Detailed methodology for optical properties and fluorescence indices can be found in Smith and Kaushal (2015), and numerous other studies have followed a similar filtration and storage procedure (Singh et al. 2014, Singh et al. 2015, Huguet et al 2009, Dubnick et al. 2010, Gabor et al. 2014). Fluorescently active DOM constitutes a wide range of lability. While some highly labile compounds may break down within hours of sample collection, more recalcitrant forms can remain stable for months. The two-week window is a convention meant to facilitate comparisons between sites, rather than a biologically based limit to storage (Personal communication, Dr. Rachel Gabor & Dr. Shuiwnag Duan). Briefly, fluorescence and absorbance properties of DOM were measured in order to evaluate the relative abundance of terrestrial and aquatic sources to the overall DOM pool.

A FluoroMax-4 Spectrofluorometer (Horiba Jobin Yvon, Edison NJ, USA) was used to measure the emission spectra of samples in response to a variety of excitation wavelengths. Excitation-emission matrices (EEMs) were used for characterizing indices of terrestrial vs. aquatic DOM sources. The humification index (HIX) is defined as the ratio of emission intensity of the 435-480 nm region of the EEM to the emission intensity of the 300-345 nm region of the EEM at the excitation wavelength of 254 nm (Zsolnay et al. 1999; Ohno 2002). HIX varies from 0 to 1, with higher values signifying high-molecular weight DOM molecules characteristic of humic terrestrial sources. Lower HIX indicates DOM of bacterial or aquatic origin (Zsolnay et al. 1999). The autochthonous inputs index (BIX) is defined as the ratio of fluorescence intensity at the emission wavelength 380 nm to the intensity emitted at 430 nm at the excitation wavelength of 310 nm (Huguet et al. 2009). Lower BIX values (< 0.7) represent terrestrial sources, and higher BIX values (> 0.8) represent algal or bacterial sources (Huguet et al. 2009).

### 2.3. Calculations

Dissolved gas concentrations were calculated using equations 2- 4. First, we used Henry's law to convert measured mixing ratios (ppmv) to the molar concentration of each gas in the headspace vial [Cg] (µmol L$^{-1}$) following Eq. 2,

$$[C] = \frac{PV}{RT} \tag{2}$$

where P is pressure (1 atm), V is the measured partial pressure of the gas of interest (ppmv), R is the universal gas constant (0.0821 L atm mol$^{-1}$ K$^{-1}$), and T is the temperature of a water sample (Kelvin) during headspace equilibration. We used Henry's law and a temperature-corrected Bunsen solubility coefficient to calculate $[C_{aq}]$, the concentration of residual gas remaining in water following headspace equilibration (Eq. 3, Stumm and Morgan 1981)

$$[C_{aq}] = \frac{V*Bp*Bunsen}{RT} \tag{3}$$

where V is measured gas mixing ratio (ppmv), Bp is the barometric pressure (atm), and Bunsen is the solubility coefficient in the vessel (L L$^{-1}$ atm$^{-1}$). Calculations of the Bunsen coefficient were based on Weiss (1974) for $CO_2$, Weiss (1970) for $N_2O$, and Yamamoto et al., (1976) for $CH_4$.

The final stream water concentration $[C_{str}]$ was then calculated using mass balance of these two pools, described in Eq. (4), where Vaq and Vg were the volumes of water and gas respectively in a water sample with helium headspace.

$$[C_{str}] = \frac{[Caq]*Vaq+[Cg]*Vg}{RT} \tag{4}$$

Because gas solubility is temperature dependent, it was useful to display gas concentrations as the percent saturation, or the ratio of the measured dissolved gas concentration to the equilibrium concentration. To determine gas saturation, the equilibrium concentration, $[C_{eq}]$, was calculated based on water temperature, atmospheric pressure, and an assumed value for the current atmospheric mixing ratios of each gas following Eq. (3). We obtained current ratios for $CO_2$ from The Keeling Curve (Scripps Institution of Oceanography, 2013), and $N_2O$ and $CH_4$ from the NOAA Earth Systems Research Laboratory (NOAA ESRL 2013; Dlugokencky, accessed 2013). Saturation ratio is defined as a ratio $[C_{str}]$ / $[C_{eq}]$, and excess (i.e $xsCO_2$) is described as a mass difference ($[C_{str}]$ - $[C_{eq}]$). Supersaturation is the condition when the saturation ratio is greater than 1, or gas excess (i.e. $xsCO_2$) is greater than 0.

### 2.3.2 Apparent Oxygen Utilization

Apparent oxygen utilization is defined as the difference between the $O_2$ concentrations (μM) at equilibrium with the atmosphere vs. ambient measured $O_2$ concentrations in the stream. A positive value of AOU represents net oxygen consumption conditions along the soil-groundwater-stream flowpath, while negative AOU (μM) represents net $O_2$ production within the stream. Because aerobic respiration and photosynthesis couples $CO_2$ production and $O_2$ consumption, we can assume that AOU is equivalent to the $CO_2$ produced / consumed along the same flowpath (Richey et al. 1998). Under aerobic conditions, respiration of organic

matter consumes $O_2$ and produces $CO_2$ at approximately a 1:1 molar ratio (Schlesinger 1997). Therefore, 1 mole of AOU should result in 1 mol of $xsCO_2$ (Measured - equilibrium $CO_2$ concentration). This ratio was then used, with an offset to 1.2:1 to account for differences in diffusion constants for the two gases (Stumm and Morgan 1981; Richey et al. 1988), to determine the proportion of $CO_2$ produced by aerobic respiration. When $CO_2$ concentrations are greater than AOU, the difference between

measured $CO_2$ and AOU ($xsCO_2$-AOU) represents additional sources from either anaerobic respiration or abiotic sources. We split our analysis of $CO_2$ into these two categories (AOU and $xsCO_2$-AOU) in order to determine whether patterns in $CO_2$ saturation were solely represented aerobic reparation or other processes and sources as well.

**2.3.3 Greenhouse gas emissions**

We calculated the gas flux rate using Eq. (5) where $F_{GT}$ is the flux (g m$^{-2}$ d$^{-1}$) of a given gas (G) at ambient temperature (T) and d

is water depth (m). $K_{GT}$ (day$^{-1}$) is the re-aeration coefficient for a given G at ambient T. Measured and equilibrium gas concentrations [$C_{str}$] and [$C_{eq}$] were calculated following equations 3 and 4, then converted to units of g m$^{-3}$.

$$F_{GT} = K_{GT} * d * ([C_{str}] - [C_{eq}]) ,\tag{5}$$

We modeled $K_{GT}$ for each site and sampling date using the energy dissipation model (Tsivoglou and Neal 1976). The energy dissipation model predicts K from the product of water velocity (V, m day$^{-1}$), water surface gradient (S), and the escape

coefficient, $C_{esc}$, (m$^{-1}$, Eq. 6).

$$K = C_{esc} * S * V \tag{6}$$

$C_{esc}$ is a parameter related to additional factors other than streambed slope and velocity that affect gas exchange, such as streambed roughness and the relative abundance of pools and riffles. The $C_{esc}$ value used in this study was derived from 22

measurements of K, made using the $SF_6$ gas tracer method, carried out across a range of flow conditions in four streams within 5 km of our study sites and reported in Pennino et al. (2014). $C_{esc}$ was calculated as the slope of the regression of K vs. S*V from data in Pennino et al (2014) and was assumed to be representative of our headwater stream sites in Dead Run and Red Run. We calculated $C_{esc}$ to be 0.653 m$^{-1}$ (n=22, r$^2$=0.42, p= 0.001). The 95% confidence interval of this $C_{esc}$ based on measured $K_{20,O2}$ values was ±0.359 m$^{-1}$, which corresponds to ±55% of a given gas flux estimate. This estimate of $C_{esc}$ from these nearby sites

was assumed to be representative of the 8 stream reaches investigated in this study. Given the moderate range of uncertainty in

$C_{esc}$, as well as additional uncertainties associated with slope estimation and relating $C_{esc}$ to different stream sites, gas flux estimates must be interpreted with caution.

Measurements of K were converted to K for each GHG (as well as $O_2$ for general comparisons) by multiplying by the ratio of their Schmidt numbers (Stumm and Morgan 1981). K measured at ambient temperature was converted to K at 20C ($K_{20}$) following Eq. 7.

$$K_{20} = \frac{KT}{11.0421^{T-20}} \tag{7}$$

In order to compare re-aeration rates across sites and prior studies, we calculated the gas transfer velocity, $k_{600,}$ which is defined as $K_{20,O2}$ multiplied by water depth, with units of m d$^{-1}$.

We estimated S of headwater streams with GHG sampling sites by measuring the change in elevation along the stream above and below stream gaging stations. We determined the latitude and longitude of the stream gage, which was co-located with GHG sampling sites in Red Run and Dead Run using a Trimble GeoXH handheld 3.5G edition GPS unit (10cm accuracy). We then plotted this location atop a 1-m resolution LiDAR-based DEM (Baltimore County Government, 2002) in ArcMap 10. Using low points in the DEM to represent the stream channel, we then selected one point above and one point below the stream gaging station and measured the distance between these two points along the stream channel with the 'Measure' tool. We calculated S based on the change in elevation divided by distance. The slope measurement reach overlapped with, but did not coincide exactly with the gas sampling reach in order to ensure measureable differences in elevation. We followed the same protocol to estimate S for reaches in Pennino et al (2014), except rather than estimating points above and below a gaging station, we determined the change in elevation over the specific reach where $SF_6$ injections took place. Pennino et al (2014) provided data on the latitude and longitude of their $SF_6$ injection reaches.

Pennino et al's (2014) measurements of V during gas injections ranged from 0.02 to 0.15 m s$^{-1}$. V measured at headwater gaging stations in our sites ranged from undetectable to 0.34 m s$^{-1}$. In order to avoid extrapolation, we limited our estimation of gas fluxes to sampling sites and dates with V in the range measured by Pennino et al. (2014). These conditions corresponded to 37 measurements total, spread unevenly across the four headwater sites with complete rating curves (DRAL, DRKV, RRRB, DRGG). K estimates were restricted to five dates at DRAL, 18 dates at DRKV, 11 dates at RRRB, and three dates at DRGG.

## 2.4 Statistical Analyses

### 2.4.1 Role of infrastructure and seasonality

A linear mixed effects modeling approach was used to determine the significant drivers of each gas across streams in different headwater infrastructure categories. Due to uncertainties in the gas flux parameters, GHG saturation ratios were used rather than GHG emissions to compare spatial and temporal patterns across sites. Mixed effects modeling was carried out using R (R Core Team, 2014) and the *nlme* package (Pinheiro et al. 2012) following guidance outlined in Zurr et al. (2009). Separate mixed effects models were used to detect the role of infrastructure category and date on each response variable. Response variables included saturation ratios for each gas ($CO_2$, N2O, and CH4), solute concentrations (DOC, DIC, TDN, $NO_3^-$), and organic matter source indices (HIX, BIX). Fixed effects were 'infrastructure category' and 'sampling date,' as well as an interaction term for the two. The effect of a random intercept for 'site' was included in each model. The statistical assumptions of normality, and equal variances were validated by inspecting model residuals. When necessary, variances were weighted based on infrastructure category to remove heteroscedasticity in model residuals (Zuur et al. 2009). The assumption of temporal independence was examined by testing for temporal autocorrelation in each response variable. This test was performed using the function 'corAR1(),' which is part of the package 'nlme' in R. The significance of random effects, weighting variances, and temporal autocorrelation was tested by comparing Akaike information criterion (AIC) scores for models with and without each of these attributes. Additionally, pairwise ANOVA tests were run to determine whether each additional level of model complexity significantly reduced the residual sum of squares. Final model selection was based on meeting model assumptions, minimizing the AIC value, and minimizing residual standard error. Pairwise comparisons among infrastructure categories were examined using the Tukey HSD post-hoc test (*lsmeans* package, Lenth, 2016) for each response variable where 'infrastructure category' had a significant effect. Where 'infrastructure category' did not have a significant effect on a response variable after incorporating 'site' as a random effect, a separate set of linear models was run with 'site' and 'date' as main effects rather than 'infrastructure category'. The role of 'site' was evaluated in these cases to determine the degree to which site-specific factors overwhelmed the effect of infrastructure category.

**2.4. Role of environmental variables on gas saturation**

A stepwise linear regression approach was used to examine the role of multiple environmental variables on $CO_2$, $N_2O$, and $CH_4$ saturation across sites and dates. Predictor variables were selected via backward stepwise procedure, using the 'Step' function in R. This involves first running a model that includes all potential driving factors, then running sequential iterations of that model after removing one variable at a time until the simplest and most robust combination of predictors was achieved. Model fit at

each step was evaluated using the AIC score. Parameters that did not reduce AIC when comparing models were removed until the model had the best fit with the minimum number of factors. The initial list of potential drivers included temperature, DO, DOC, TDN, DIC, HIX, and the BIX. Prior to the stepwise regression, we calculated the variance inflation factor (VIF) for each response variable to test for multicolinearity. VIF > 3 was the cut off for assessing multicolinearity. All variables in this study were below the VIF > 3 threshold (Zuur et al. 2010).

Analysis of covariance (ANCOVA) was carried out to determine whether relationships among gases ($CO_2$ vs. $N_2O$, $CO_2$ vs. $CH_4$) and solutes (log of DOC:$NO_3^-$ ratio) varied systematically across infrastructure categories. ANCOVA involved comparing two generalized least squares models. The first linear model included an interaction term between one of the predictor variables (i.e. DOC or $CO_2$) and infrastructure category to predict the response variable ($N_2O$ or $CH_4$). The second was a linear model with the same two independent variables but no interaction term. When infrastructure category had a significant influence on both the intercept (first model) and slope (second model) of a relationship, this refuted the null hypothesis that infrastructure category had no influence on a relationship.

Because we used three separate models to evaluate variations in three GHG concentrations (for across infrastructure categories, continuous variables, and ANCOVA), we used a Bonferroni correction for the 95% confidence level. We determined the new confidence level by dividing the 95% level (0.05) by the number of models used on all gases across headwater stream sites (6). This new p-value (0.008) was then used to determine significance rather than 0.05.

### 2.4.3 Longitudinal variability in gas saturation

We analyzed longitudinal data using multiple linear regressions in order to evaluate whether patterns observed in headwater sites were representative of the broader stream network. We compiled data from four surveys – Red Run and Dead Run in spring and fall – and used a stepwise linear regression approach to determine the significant drivers for each gas (Table 6). Covariates included log of drainage area above each point, watershed (Red Run vs. Dead Run), season (spring vs. fall), DOC concentration, DIC concentration, TDN concentration, log of discharge, location (tributary vs. main stem), DOC: TDN molar ratio, a TDN by Drainage are interaction term, and a DOC by drainage are interaction term. We used the stepAIC() function in R to determine the optimal model formulation, selecting the model with minimum AIC.

## 3 Results

### 3.1 Effect of infrastructure on water quality and DOC: $NO_3^-$ ratios

We detected significant differences among TDN, $NO_3^-$, and DOC: $NO_3^-$ ratios across infrastructure categories (Table 2). TDN concentrations ranged from 0.12 to 8.7 mg N $L^{-1}$ (Table 3). Pairwise comparisons yielded significantly higher TDN concentrations in sites in the typology of 'septic systems', compared with the 'inline SWM wetlands' typology, and sites in the 'riparian/floodplain preservation' typology. Sites in the 'stream burial' typology fell within the mid-range of TDN concentrations and were not different from any other category. DOC concentrations varied widely from 0.19 to 16.89 mg $L^{-1}$, but were not significantly predicted by infrastructure typology (Table 2). DOC: $NO_3^-$ ratios varied over four orders of magnitude, from 0.02 to 112 (Fig. 2). Infrastructure typology was a significant predictor of DOC: $NO_3^-$, with the lowest ratios in sites with septic systems and highest in sites with riparian/floodplain preservation (Fig. 2). Pairwise comparisons showed no difference in DOC: $NO_3^-$ ratios between in the inline SWM wetland and complete stream burial typologies, however (Fig. 2).

### 3.2 Effect of urban infrastructure on DOM quality

Measurements of HIX ranged from 0.30 to 0.90 while BIX ranged from 0.40 to 1.15 across all sites and sampling dates in headwater streams. Streams draining septic system infrastructure had significantly lower HIX values than any other infrastructure typology. BIX values showed no significant pattern across infrastructure typologies (Table 2).

### 3.3 Effect of urban infrastructure on gas concentrations

Mixed effects models did not detect significant influence of infrastructure typology alone on $CO_2$, $CH_4$, and $N_2O$ saturation in streams. There was, however, a significant interaction effect between sampling date and infrastructure typology on the saturation ratios of all three gases (Table 2). This indicated that sampling date was important to GHG saturation for some infrastructure typologies, or that the effect of infrastructure is dependent upon sampling date. The second set of linear models, which used site rather than infrastructure category as a main effect, yielded significant differences across all sites for $N_2O$ (Fig. 3). Similarly, for $CO_2$, there were significant differences in 25 out of 28 pairwise comparisons. Pairwise comparisons across sites for $CH_4$

saturation were significant in 23 out of 28 cases. These patterns suggest that site-specific effects overwhelmed the role of infrastructure categories on GHG saturation.

**3.4 Effect of environmental variables on gas concentrations**

Stepwise model parameter selection yielded several variables that correlate with each GHG saturation ratio (Table 4). TDN was the strongest predictor of $N_2O$ saturation, followed by DO. The final model for $N_2O$ ($r^2=0.78$) also included temperature, HIX, BIX, %SWM, and $DOC:NO_3^-$. $CO_2$ saturation had a similar pattern of predictors and nearly identical model fit ($r^2=0.78$). $DOC:NO_3^-$ ratio was the strongest predictor of $CH_4$ saturation followed by DO and temperature. HIX, %IC, and %SWM were also related to $CH_4$ saturation, but TDN and BIX were not.

**3.5 Covariance among GHG abundance and C: N Stoichiometry**

AOU ranged from -180.9 to 293.9 across all sites and sampling dates, however AOU was only negative (net oxygen production along surface and subsurface flow paths) in 6% of samples, or 43 out of 691 measurements. $N_2O$ was significnatly but weakly correlated with AOU ($p<0.008$, $r^2=0.12$), and strongly correlated with $xsCO_2$-AOU ($p<0.008$, $r^2=0.87$). Log of $CH_4$ saturation ratio was very weakly correlated with AOU ($p<0.008$, $r^2=0.01$ as well as $xsCO_2$-AOU ($p<0.008$, $r^2=0.07$). The relationships between $xsCO_2$ –AOU and both $N_2O$ and $CH_4$ saturation ratios were also significantly different between categories (Fig. 4). There was an overall negative relationship between DOC and $NO_3^-$ with a significant interaction with infrastructure category (Fig. 4c; ANCOVA p-value < 0.008).

**3.6 Longitudinal Patterns in GHG saturation**

Spatial variability in GHG saturation was examined in order to evaluate whether concentrations measured in tributaries were consistent between headwaters and larger 3[rd] order watersheds of Red Run and Dead Run respectively (Fig 5). Multiple linear regressions yielded a set of distinct controlling factors on saturation of each gas. The optimal models for $CO_2$ and $N_2O$ were similar and included the log of drainage area, TDN concentration, log of discharge, and TDN x discharge interaction term. The $CO_2$ model also included DOC: TDN molar ratio. The optimal model for $CH_4$ saturation was slightly different, and included log

of drainage area, season (spring vs. fall), DOC concentration, and DOC: TDN molar ratio (Table 6). TDN concentration was not included in the optimal model for $CH_4$. Watershed location (tributary vs. main stem) was not included in the final model for any of the three gases.

### 3.7 Greenhouse gas emissions

GHG emission rates were sensitive to differences in modeled $k_{600}$. Despite having medium to low gas saturation ratios compared with other sites, DRKV had the highest GHG emission rates on all dates. This is due in part to having the highest slope (0.10 m/m), and thus the highest modeled $k_{600}$ (m day$^{-1}$). Our 37 estimates of $k_{600}$ ranged from 2.4 to 122.6.1 m d$^{-1}$. Site-averages for $k_{600}$ varied from 5.39± 0.73 to 28.0± 7.0 m day$^{-1}$. The median value for all $k_{600}$ estimates was 13.24 m day$^{-1}$. This range of values and site-averaged values extends beyond that measured by Pennino et al. (2014) of 0.5 to 9.0 m d$^{-1}$. The discrepancy between

Pennino et al. (2014)'s $k_{600}$ measurements is driven by differences in channel gradient. Gradients in the present study ranged from 0.01 to 0.1, while Pennino's ranged from 0.001 to 0.016 m d$^{-1}$. Channel gradient (S) is also the parameter with the greatest uncertainty, thus warranting cautious interpretation of our gas emission estimates.

Site-average $CO_2$ emissions ranged 6.4± 2.3 g C m$^{-2}$ day$^{-1}$ at DRAL (± standard error) to 134 ± 30.2 at DRKV. Mean emission rates for DRGG and RRRB were 11.5 ± 6.1 and 10.3 ± 1.7 respectively. Site-average $CH_4$ emissions ranged from 2.6 ± 1.1 at

DRAL to 102.5± 75.6 mg C m$^{-2}$ day$^{-1}$ at DRKV. $N_2O$ emissions ranged from 5.1± 0.8 at RRRB to 149 ±33.9 mg N m$^{-2}$ d$^{-1}$ at DRKV. The full range of values and standard errors for fluxes are listed in Table 5.

## 4 Discussion

### 4.1 Overview

This study showed strong relationships between urban water quality and GHG saturation across streams draining different forms of urban infrastructure. $N_2O$ and $CO_2$ saturation was correlated with nitrogen concentrations, but did not differ between infrastructure typologies. DOC: $NO_3^-$ did differ among the four infrastructure categories, however (Table 2). While infrastructure

categories did not show a significant predictor of GHG saturation in streams, the gradients in DOC: $NO_3^-$ found across all categories was strongly correlated with GHG saturation. Stoichiometric variation may thus serve as a predictor of GHG saturation downstream where land cover and infrastructure does not. While direct GHG loading to streams from leaky sanitary and / or stormwater infrastructure may play a role, the strongest predictors of GHGs in this study were continuous/ environmental variables (i.e. TDN and DOC concentrations, DO, temperature), rather than categorical (infrastructure category). Relationships between anaerobic $xs$CO$_2$ – AOU and N$_2$O saturation further suggest that anaerobic metabolism contributes to N$_2$O production along hydrologic flowpaths (Fig. 4).

### 4.2 C:N Stoichiometry as an Indicator of Microbial Metabolism

By comparing various forms of infrastructure, results from this study support a growing understanding of the biogeochemical consequences of expanded hydrologic connectivity in urban watersheds. Strong inverse relationships between DOC and $NO_3^-$ present across all infrastructure categories (Fig. 4c) suggest that organic carbon availability modulates inorganic nitrogen loading to streams. DOC availability has been shown to control $NO_3^-$ concentrations across terrestrial and aquatic ecosystems through a variety of coupled microbial processes (Hedin et al. 1998, Kaushal and Lewis 2005, Taylor and Townsend 2010). Additionally, the average DOC: $NO_3^-$ ratio, (i.e. the slope of this relationship) varied significantly across categories. Variation in this relationship is likely driven by a combination of differential N loading across categories as well as different capacities for microbial N uptake and removal.

We speculate that the location of infrastructure on the landscape may affect the relative importance of direct anthropogenic loading vs. microbial processes on DOC: $NO_3^-$ ratios of stream water. For instance we found high concentrations of and $NO_3^-$ and low DOC in streams draining septic systems. Much of this excess $NO_3^-$ is likely from septic plumes, but the lack of DOC may be the result of microbial C mineralization along subsurface flowpaths. On the other end of the spectrum, very low $NO_3^-$ and TDN in streams draining watersheds in the floodplain preservation category, which were also newly developed. In this case, the higher C:N may have been driven by lower N leakage rates as well as improved ecological function of the preserved floodplain wetlands to remove any N that does enter the groundwater from stormwater or sewage leaks.

Understanding the spatial variability in N$_2$O concentrations, as well as the processes responsible for N$_2$O production and $NO_3^-$ removal in watersheds is useful for informing watershed management. The relationship between N$_2$O and CO$_2$ can provide insight into production mechanisms because nitrification consumes CO$_2$ while denitrification simultaneously produces N$_2$O and

$CO_2$. We found a strong positive relationship between $N_2O$ saturation and $CO_2$ concentrations, suggesting that denitrification was the primary source of $N_2O$ (Figure 5c). By contrast, very low DOC: $NO_3^-$ ratios (Figure 2) in stream water with highest $N_2O$ saturation (Figure 3a) suggest that nitrification was the dominant process at these sites. Taylor and Townsend (2010) suggest that the ideal DOC: $NO_3^-$ stoichiometry for denitrification is 1:1, and that persistent conditions below that are more ideal for nitrification. DOC: $NO_3^-$ was consistently below 1 in streams in septic system infrastructure, suggesting that in-stream denitrification would be carbon limited. We measured DOC: $NO_3^-$ consistently above 1 at sites in riparian/floodplain preservation typology, suggesting $NO_3^-$ was limiting for in-stream denitrification this infrastructure category. Conversely, the mean stoichiometric ratio was consistently near 1 in sites with inline SWM wetlands and stream burial, suggesting that denitrification may be occurring within the stream channel at these sites. While DOC: $NO_3^-$ stoichiometry in watersheds with septic systems appeared more favorable for nitrification, the positive $xsCO_2$ –AOU vs. $N_2O$ relationships in these streams suggest that these gases were produced anaerobically (by denitrification). One possible explanation for this discrepancy is that the $N_2O$ and $CO_2$ observed in the stream were produced under stoichiometric conditions more favorable for denitrification along groundwater flow paths prior to emerging in the stream channel. Denitrification occurring along groundwater flowpaths may draw down the DOC concentration as it is converted to $CO_2$, however the initial N load in septic plumes may be too high to noticeably decline. Pabich et al. (2001) documented this phenomenon, in which DOC concentrations in a septic plume were quite high (>20 mg L$^{-1}$) in the upper part of the plume, and declined exponentially resulting in a very low DOC:$NO_3^-$ ratio at depth.

Overall, the relationships between $CH_4$ and $CO_2$ were much weaker and more variable than the relationships between $CO_2$ and $N_2O$ (Figure 4). While $CO_2$ and $CH_4$ are sometimes correlated in wetlands and rivers with low oxygen (Richey et al. 1998), this was not the case for our study sites. Instead, $CO_2$ and $N_2O$ were highly coupled, suggesting prevalence of $NO_3^-$ as a terminal electron acceptor over $CO_2$.

**4.3 Effects of infrastructure on $N_2O$ Saturation and Emissions**

The present study documents some of the highest $N_2O$ concentrations currently reported in the literature for streams and rivers, ranging from 0.009 to 0.55 µM, with a median value of 0.07µM and mean of 0.11 µM $N_2O$-N. This range of concentration is greater than that reported for headwater agricultural and mixed land use streams in the Midwestern United States (0.03 – 0.07 µM, Werner et al. 2012; 0.03 to 0.15 µM, Beaulieu et al. 2008). A similar range of dissolved $N_2O$ concentrations was reported for macrophyte- rich agriculturally influenced streams in New Zealand (0.06 to 0.60µM, Wilcock and Sorrell, 2008). The only

report of higher dissolved $N_2O$ concentrations in streams is from a subtropical stream receiving irrigation runoff, livestock waste, and urban sewage (saturation ratio max of 60 compared with 47 in this study; Harrison et al. 2005).

Average daily $N_2O$ emissions were high, ranging from 5.1 to 149.6 mg $N_2O$-N $m^2$ $d^{-1}$. Our values rates fall on the high end compared with numerous studies of $N_2O$ emission from urban and agriculturally influenced waterways, including agricultural drains in Japan (max= 179 mg N $m^2$ $d^{-1}$; Hasegawa et al. 2000) or the Humber Estuary, UK (max= 121 mg N $m^2$ $d^{-1}$ Barnes and Owens 1998). When the highest site (DRKV) is removed, these average daily fluxes remain high (range= 5.1 to 12.3 mg N $m^2$ $d^{-1}$) compared with estimates reported for nitrogen enriched agricultural and mixed land use streams in the Midwestern U.S. from Beaulieu et al. 2008 (mean= 0.84 and maximum = 6.4 mg $N_2O$- N $m^2$ $d^{-1}$). Laursen and Seitzinger (2004) reported higher maximum rates (20 mg N $m^2$ $d^{-1}$) to our overall median $N_2O$ emission rates (13.8 mg N $m^2$ $d^{-1}$), and the maximum daily rates measured in tropical agricultural streams in Mexico (mean = 1.2 max= 58.8 mg $N_2O$-N $m^2$ $d^{-1}$, Harrison and Matson 2003). While our measured $N_2O$ saturation ratios were highly correlated solute concentrations and redox conditions (Table 4), emission rates sensitive to the gas transfer velocity ($k_{600}$), which varied by two orders of magnitude in our study (Table 6).

Correlations between TDN and $N_2O$ concentrations in this study highlight the role of urban N loading on GHG production along urban flowpaths- which include groundwater, within pipes, and along the stream network (Tables 3 & 4). While urban streams receive a mixture of different N sources including fertilizer, wastewater, atmospheric deposition (e.g. Kaushal et al. 2011; Pennino et al. 2016), the location of aging gravity sewers adjacent to stream channels is likely to influence the relative importance of sewage on N and $N_2O$ loading to streamwater. While this source of $N_2O$ emission is likely a small portion of the global budget, gaseous losses of N can contribute significant portion of watershed-scale N budgets, which are relevant to nutrient management (Gardner et al. 2015). $N_2O$ emissions from uncollected human waste (i.e. leaky sanitary sewer lines, septic system effluent, dug pits) are largely unmeasured globally (Strokal and Kroeze 2014; UNEP 2013) and warrant further study in the context of watershed management as well as local GHG accounting. Direct emissions from wastewater treatment plants (WWTPs) are well documented (Foley et al. 2010; Townsend-Small et al. 2011; Strokal and Kroeze 2014; UNEP 2013), however the upstream losses of $N_2O$ from delivery pipes into streams and rivers are not (Short et al. 2014). Short et al. (2014) measured $N_2O$ concentrations in WWTP influent in Australia and determined that sanitary sewers are consistently super-saturated with $N_2O$, with concentrations in excess of equilibrium by as much as 3.5μM. Average daily sewer pipe $xsN_2O$ concentrations were 0.55 μM, which is nearly identical to the maximum $xsN_2O$ measured in the present study (0.54 μM). While wastewater only contributes a portion of excess N in urban streams, further accounting for this source is necessary to improve municipal $N_2O$ budgets.

Synoptic surveys of $N_2O$ saturation in Red Run and Dead Run in this study provide evidence that the entire network is a net source of $N_2O$ (Fig. 5). $N_2O$ saturation shows a significant decline with increasing drainage area (Table 6, Fig. 5), suggesting that emissions outpace new sources to the water column. Variability in gas concentrations headwater sites and along the 3rd order stream networks are largely explained by a combination of discharge and/or drainage area, as well as N concentrations and C:N stoichiometry in streamwater.

### 4.4 Effects of infrastructure on $CH_4$ Saturation and Emissions

Methane was consistently super-saturated across all streams in this study, and varied significantly across headwater infrastructure categories. The highest $CH_4$ saturation ratios were measured in sites with riparian reconnection (RRRM and RRRB) followed by streams draining inline SWM wetlands (DRKV and DRGG) (Fig. 3 As with $CO_2$). $CH_4$ saturation was negatively correlated with DO, however $CH_4$ was positively correlated with DOC: $NO_3^-$. $CO_2$ and $N_2O$, by contrast, were more strongly and positively correlated with TDN (Table 4). These patterns suggest that, along with redox conditions, carbon availability may modulate $CH_4$ production as well.

$CH_4$ concentrations in our study ranged from 0.06 to 6.08 μmol $L^{-1}$, equivalent to the mean +/- standard deviation of concentrations reported by a meta-analysis by Stanley et al. (2016). Saturation ratio (3.0 to 2157) fell within the lower range of previously measured values in agricultural streams in Canada (sat. ratio 500 to 5000, Baulch et al. 2011a). Mean daily $CH_4$ emissions estimates in this study ranged from 2.6 to 103.5 mg $CH_4$-C $m^2$ $d^{-1}$ and are comparable to measurements in agricultural streams of New Zealand (Wilcock and Sorrel, 2008; 17-56 mg $CH_4$-C $m^2$ $d^{-1}$) and southern Canada (20-172mg C $m^2$ $d^{-1}$, Baulch et al. 2011), however these studies also measured ebullitive (i.e. bubble) fluxes, whereas the present study only examined diffusive emissions. Stanley et al. (2016) reported the average of all current $CH_4$ emission rates to be 98.7 mg $CH_4$-C $m^2$ $d^{-1}$ with a minimum of -125.3 and a maximum of 5,194 overall. While the $CH_4$ emission estimates in the present study have a large margin of uncertainty due to the nature of estimating gas flux parameters as well as the lack of ebullitive flux measurements, our sites were consistently sources to the atmosphere throughout the year at both headwater sites (Figure 3) and throughout 3rd order drainage networks (Figure 5b). Differences in $CH_4$ abundance across infrastructure categories, as well as the negative relationship between $CH_4$ saturation and TDN, suggest that $CH_4$ may increase if TDN declines with the addition of stormwater wetlands and floodplain reconnection in urban areas.

**5 Conclusions**

Urban watersheds are highly altered systems, with heterogeneous forms of infrastructure and water quality impairment. The present study demonstrates that $N_2O$ and $CH_4$ saturation and emissions from urbanized headwaters are on the high end of estimates currently reported in the literature. Variations in urban infrastructure (i.e. SWM wetlands, riparian connectivity, septic systems) influenced C:N stoichiometry and redox state of urban streams. These in-stream variables, along with potential direct sources from leaky sanitary sewer lines may contribute to increased GHG production and/or delivery to streams. Our results suggest that N from septic plumes and sanitary sewer lines is the principal source of $N_2O$ saturation in our study sties. Dissolved inorganic N is highly correlated with $N_2O$ in our study sites, and the highest values are only present in watersheds with aging sanitary sewer infrastructure or septic systems. Our observations of $N_2O$ saturation and emissions from urban and suburban headwater streams are comparable with streams and ditches in intensive agricultural watersheds (Harrison and Matson. 2003; Outram et al. 2012). These results suggest that streams draining medium to low-density suburban or exurban land cover are comparable to those in intensively managed agricultural areas in terms of $N_2O$ emissions.

**Code availability:** The authors are happy to share any and all codes used to produce this manuscript. Please contact the corresponding author with inquiries about the codes used.

**Data availability:** The authors have provided tables of all raw data collected for this study in the supplementary information files. These datasets will additionally be available as part of the Baltimore Ecosystem Study LTER Site archive (www.beslter.org).

**Author Contributions:**

R. Smith, S. Kaushal, C. Welty and M. Pennino selected sampling sites based on infrastructure typology. R. Smith, S. Kaushal and J. Beaulieu designed the gas and solute sampling design. R. Smith and J. Beaulieu analyzed samples for solute and gas concentrations respectively. C. Welty collected continuous flow data from headwater gaging stations. J. Beaulieu provided key insights into interpretation of gas concentrations and statistical analyses and gas flux estimations. M. Pennino provided data used for estimating $K_{20}$. S. Kaushal and C. Welty provided funding for the project. All coauthors provided feedback on multiple versions of the manuscript.

**Competing interests:** The authors declare that they have no conflict of interest.

**Acknowledgements**

The authors gratefully acknowledge funding from the National Science Foundation Water Sustainability and Climate program (NSF grants CBET-1058038 and CBET-1058502), as well as scientific infrastructure provided by the Baltimore Ecosystem Study LTER (www.beslter.org, NSF grant DEB-1027188). Field data collection was also partially supported by NOAA grant NA10OAR431220 to the Center for Urban Environmental Research and Education (www.cuere.umbc.edu) and the Water Resources mission area of the U. S. Geological Survey (water.usgs.gov). Daniel Jones provided advice on spatial analyses and numerous individuals including Tamara Newcomer, Tom Doody, Evan McMullen, John Urban, Shahan Haq, Julia Gorman, Julia Miller, John Kemper, Erin Stapleton, and Joshua Cole provided field assistance and/or feedback on drafts of this manuscript. The views expressed in this article are those of the authors and do not necessarily reflect the views or policies of the U.S. Environmental Protection Agency.

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

**Tables**

Table 1. Summary of site characteristics including drainage area (km$^2$), percent impervious cover (%IC), and percent of the watershed drained by GI stormwater best management practices (i%GI SWM drainage).

| Infrastructure feature | Site | Drainage area (km$^2$) | % IC cover | % GI SWM drainage | Description |
|---|---|---|---|---|---|
| Septic Systems | RRSD | 0.23 | 7.9 | 0.00 | Low-density residential development with septic systems, minimal stormwater management with some stream burial. |
| | RRSM | 0.68 | 3.78 | 13.97 | |
| Floodplain Preservation | RRRM | 0.63 | 16.4 | 100.00 | Suburban and commercial low-impact development converted from agriculture in early 2000s. Stormwater wetlands in upland + wide riparian buffer zones surround each stream and sanitary sewer infrastructure. |
| | RRRB | 0.21 | 22.81 | 54.67 | |
| Inline SWM Wetlands | DRKV | 0.31 | 39.16 | 100.00 | Older suburban development (1950s) with GI located inline with stream channels, rather than dispersed across the landscape. Watershed is serviced by sanitary sewers. |
| | DRGG | 0.6 | 36.68 | 47.60 | |
| Stream Burial | DRAL | 0.26 | 41.9 | 1.10 | Older suburban and commercial development (1950s) with piped headwaters upstream of the sampling point. Watershed is serviced by sanitary sewers. No management of stormwater other than the pipe network, which also contains buried streams. |
| | DRIS | 0.18 | 30.57 | 0.00 | |

Table 2 Summary of results (main effects p-values) from mixed effects models examining the role of infrastructure typology and date on the following response variables: $CO_2$, $N_2O$ and $CH_4$ saturation ratios; TDN and DOC concentrations (mg $L^{-1}$), BIX, and HIX (unitless).

| Main Effects | $CO_2$ | $CH_4$ | $N_2O$ | TDN | DOC | BIX | HIX | DOC: $NO_3^-$ |
|---|---|---|---|---|---|---|---|---|
| Infrastructure typology p-value | 0.496 | 0.298 | 0.488 | 0.068 | 0.200 | 0.441 | 0.020 | <0.008* |
| Date p-value | 0.957 | <0.008* | <0.008* | 0.086 | 0.387 | 0.155 | 0.765 | 0.492 |
| Date by Infrastructure Typology Interaction p-value | <0.008* | <0.008* | <0.008* | 0.114 | 0.978 | 0.490 | 0.899 | 0.894 |

Table 3. Mean with standard error in parentheses of GHG saturation ratios, TDN and DOC concentrations (mg L$^{-1}$), BIX values and HIX values for each site.

| Infrastructure Typology | Site | $CO_2$ | $CH_4$ | $N_2O$ | TDN | DOC | BIX | HIX | DOC: $NO_3^-$ |
|---|---|---|---|---|---|---|---|---|---|
| Septic Systems | RRSD | 52.9 (1.1) | 14.9 (0.5) | 28.0 (0.7) | 6.40 (0.20) | 0.76 (0.12) | 0.89 (0.02) | 0.74 (0.01) | 0.06 (0.01) |
| | RRSM | 13.5 (0.5) | 25.6 (1.5) | 5.9 (0.2) | 3.49 (0.13) | 1.40 (0.25) | 0.70 (0.02) | 0.782 (0.015) | 0.27 (0.04) |
| Riparian/ Floodplain Preservation | RRRM | 6.6 (0.3) | 207.3 (36.2) | 1.7 (0.04) | 0.59 (0.08) | 2.89 (0.27) | 0.67 (0.01) | 0.85 (0.02) | 12.16 (3.45) |
| | RRRB | 9.6 (0.4) | 103.6 (8.6) | 3.6 (0.1) | 0.35 (0.02) | 1.58 (0.18) | 0.716 (0.01) | 0.85 (0.01) | 9.24 (2.43) |
| Inline SWM | DRKV | 28.1 (1.0) | 50.8 (8.5) | 19.1 (0.6) | 2.52 (0.16) | 2.65 (0.24) | 0.75 (0.01) | 0.86 (0.003) | 2.38 (0.67) |
| | DRGG | 16.3 (1.1) | 225.8 (31.9) | 7.9 (0.4) | 1.16 (0.07) | 5.32 (0.60) | 0.73 (0.02) | 0.83 (0.01) | 8.72 (2.23) |
| Stream Burial | DRAL | 7.9 (0.3) | 11.3 (0.6) | 5.1 (0.2) | 2.68 (0.09) | 2.64 (0.37) | 0.81 (0.01) | 0.83 (0.01) | 1.42 (0.40) |
| | DRIS | 22.6 (1.0) | 78.4 (5.8) | 10.7 (0.5) | 2.42 (0.09) | 2.51 (0.27) | 0.79 (0.01) | 0.82 (0.01) | 1.82 (0.44) |

Table 4. Main effects, model coefficients, adjusted $r^2$, and overall model p-value for stepwise regression models examining the relationship between continuous variables and GHG saturation ratios. The model coefficient is the main effect of each parameter, and the absolute value of this coefficient signifies the relative contribution of each predictor. A * indicates the predictor with the greatest influence for each response variable ($CO_2$, $CH_4$, and $N_2O$). Rows with with 'n.a.' indicate that the predictor variable was not retained in the final model.

| Predictor | $CO_2$ Coefficient | $CH_4$ Coefficient | $N_2O$ Coefficient |
|---|---|---|---|
| TDN | 1.08* | n.a. | 1.10* |
| Temperature | -0.22 | 0.25 | -0.26 |
| DO | -0.46 | -0.27 | -0.37 |
| HIX | 0.09 | -0.15 | 0.13 |
| BIX | 0.11 | n.a. | 0.15 |
| %IC | n.a. | -0.16 | 0.14 |
| %SWM | 0.18 | 0.16 | 0.31 |
| $log(DOC:NO_3^-)$ | 0.32 | 0.55* | 0.19 |
| *Overall Model Fit* | | | |
| Adjusted $r^2$ | 0.78 | 0.5 | 0.78 |
| P-value | <0.008* | <0.0008* | <0.008* |

Table 5. Summary of gas flux estimations for the four sites with continuous flow data. Average, standard error (s.e.), and number of measurements (n) are listed for $CO_2$ (g C $m^{-2}$ $day^{-1}$), $CH_4$ (mg C $m^{-2}$ $day^-$), $N_2O$ (mg N $m^{-2}$ $day^-$), and predicted $k_{600}$ (m $day^{-1}$)

| Infrastructure typology | Site | Parameter | Minimum | Maximum | Mean | s.e. | n |
|---|---|---|---|---|---|---|---|
| Stream Burial | DRAL | | 2.37 | 23.12 | 11.51 | 6.12 | 5 |
| Inline SWM | DRGG | $CO_2$ g C $m^{-2}$ $d^{-1}$ | 53.28 | 548.01 | 134.55 | 30.18 | 3 |
| Inline SWM | DRKV | | 3.39 | 23.81 | 10.30 | 1.74 | 18 |
| Floodplain Preservation | RRRB | | 0.61 | 5.51 | 2.55 | 1.10 | 11 |
| Stream Burial | DRAL | | 7.71 | 23.67 | 14.09 | 4.88 | 5 |
| Inline SWM | DRGG | $CH_4$ mg C $m^{-2}$ $d^-$ | 2.27 | 1339.62 | 102.51 | 75.57 | 3 |
| Inline SWM | DRKV | | 3.26 | 62.98 | 16.80 | 5.29 | 18 |
| Floodplain Preservation | RRRB | | 2.19 | 12.11 | 6.69 | 2.19 | 11 |
| Stream Burial | DRAL | | 2.13 | 24.21 | 12.33 | 6.43 | 5 |
| In-line SWM | DRGG | $N_2O$ mg N $m^{-2}$ $d^-$ | 60.45 | 565.17 | 149.63 | 33.91 | 3 |
| In-line SWM | DRKV | | 1.90 | 8.61 | 5.14 | 0.79 | 18 |
| Floodplain Preservation | RRRB | | 2.57 | 16.98 | 7.03 | 2.63 | 11 |
| Stream Burial | DRAL | | 3.84 | 19.20 | 10.97 | 4.47 | 5 |
| Inline SWM | DRGG | $k_{600}$ m $d^{-1}$ | 12.82 | 122.59 | 28.02 | 7.06 | 3 |
| Inline SWM | DRKV | | 2.40 | 8.89 | 5.39 | 0.73 | 18 |
| Floodplain Preservation | RRRB | | 2.57 | 13.91 | 6.45 | 2.33 | 11 |

5 Table 6. Covariates and model fit parameters for linear models describing drivers of gas saturation ratios ($CO_2$, $CH_4$ and $N_2O$) from longitudinal surveys of Dead Run and Red Run. 'X's denote that a given parameter was used in the final model while dashes (-) denote parameters not used.

| Covariates Tested | $CO_2$ Sat. Ratio | $CH_4$ Sat. Ratio | $N_2O$ Sat. Ratio |
|---|---|---|---|
| log of drainage area ($km^2$) | X | X | X |
| Watershed (Dead Run vs. Red Run) | - | - | - |
| Season | - | X | X |
| DOC ($mg\ L^{-1}$) | - | X | - |
| DIC ($mg\ L^{-1}$) | - | - | - |
| TDN ($mg\ L^{-1}$) | X | - | X |
| Log of Q ($m^3\ s^{-1}$) | X | - | X |
| Location (tributary vs. main stem) | - | - | - |
| DOC:TDN molar ratio | X | X | - |
| TDN x log of drainage area interaction | X | - | X |
| DOC x log of drainage area interaction | - | - | - |
| | | | |
| Model AIC | 336.85 | 542.14 | 263.59 |
| Overall model $r^2$ | 0.789 | 0.153 | 0.795 |
| Overall Model p-value | <0.008 | 0.0082 | <0.008 |

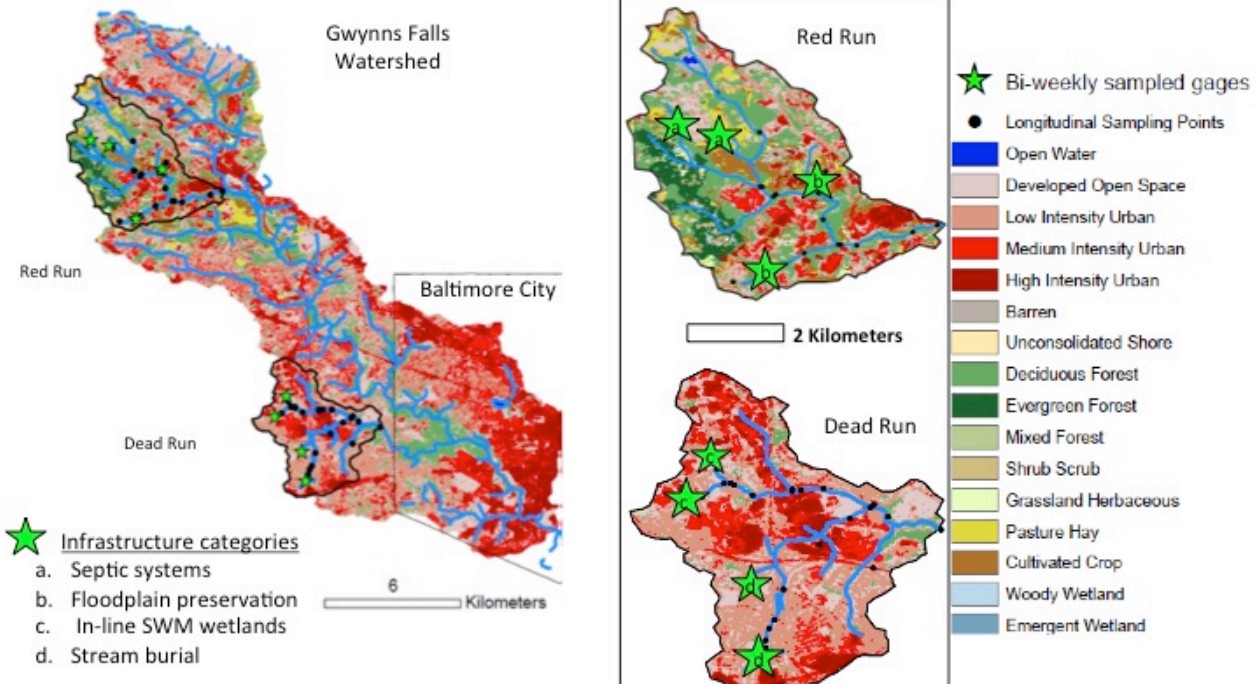

**Figure 1: Site map of headwater stream sites within Red Run and Dead Run watersheds. Green stars signify bi-weekly sampling sites, and black dots signify longitudinal sampling points sampled seasonally. Land cover categories are colored based on the National Land Cover Database, with dark red areas signifying dense urban land cover, light red signifying medium urban land cover, and green colors signifying forested or undeveloped areas. Close-up views of Dead Run and Red Run on the right represent the study watersheds, with areas that are captured by stormwater management structures (detention basins, wetlands, sand filters, etc.) shaded in gray.**

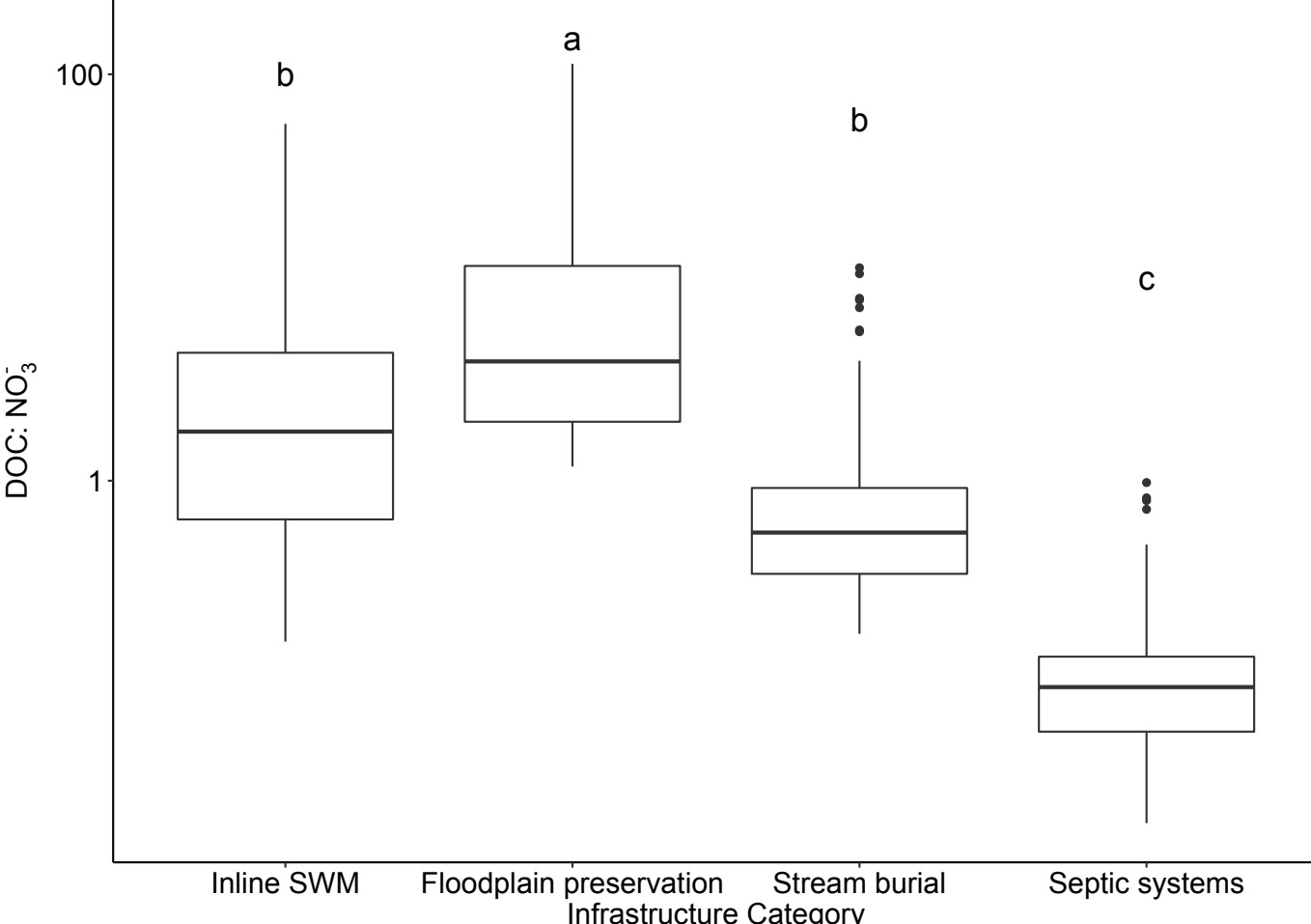

Figure 2: Boxplot of molar DOC: NO$_3^-$ ratio across sites in watersheds with differing infrastructure typologies. The median of each dataset is signified by the middle horizontal line for each category. Boxes signify the range between first and third quartiles (25[th] and 75[th] percentiles). Vertical lines extend to the minimum and maximum points in the dataset that are within 1.5 times the inter-quartile range. Points signify data that fall above or below this range. Letters represent significant ($p < 0.01$) differences between infrastructure typologies for DOC:NO$_3^-$ across all sampling dates, determined using a linear mixed effects model.

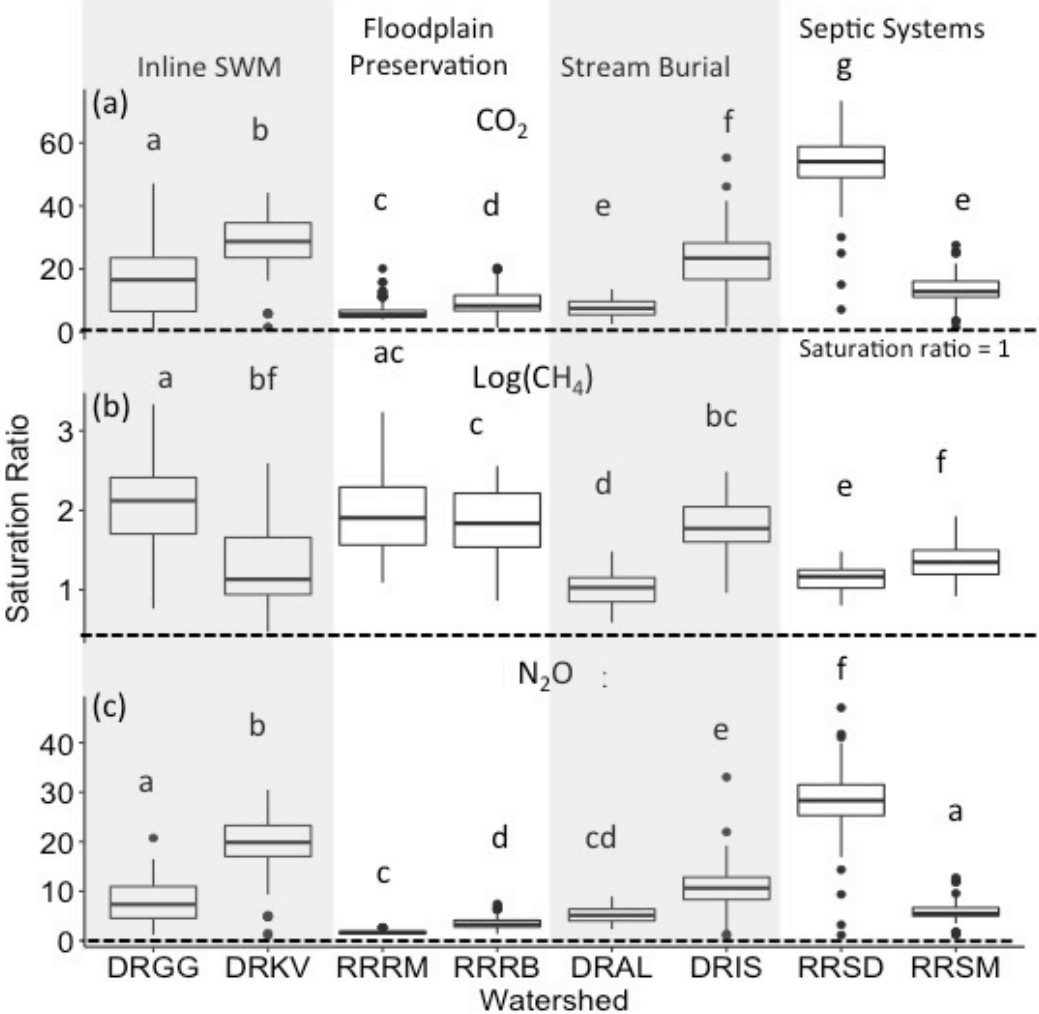

**Figure 3: Boxplot of $CO_2$, $CH_4$, and $N_2O$ saturation ratios across stream sites in varying infrastructure categories. Letters denote significant pairwise differences across streams for a given gas from linear mixed effects models with 'watershed' as a main effect. Boxes signify the range between first and third quartiles (25th and 75th percentiles). Vertical lines extend to the minimum and maximum points in the dataset that are within 1.5 times the inter-quartile range. .Points signify outliers outside of 1.5 times the interquartile range.**

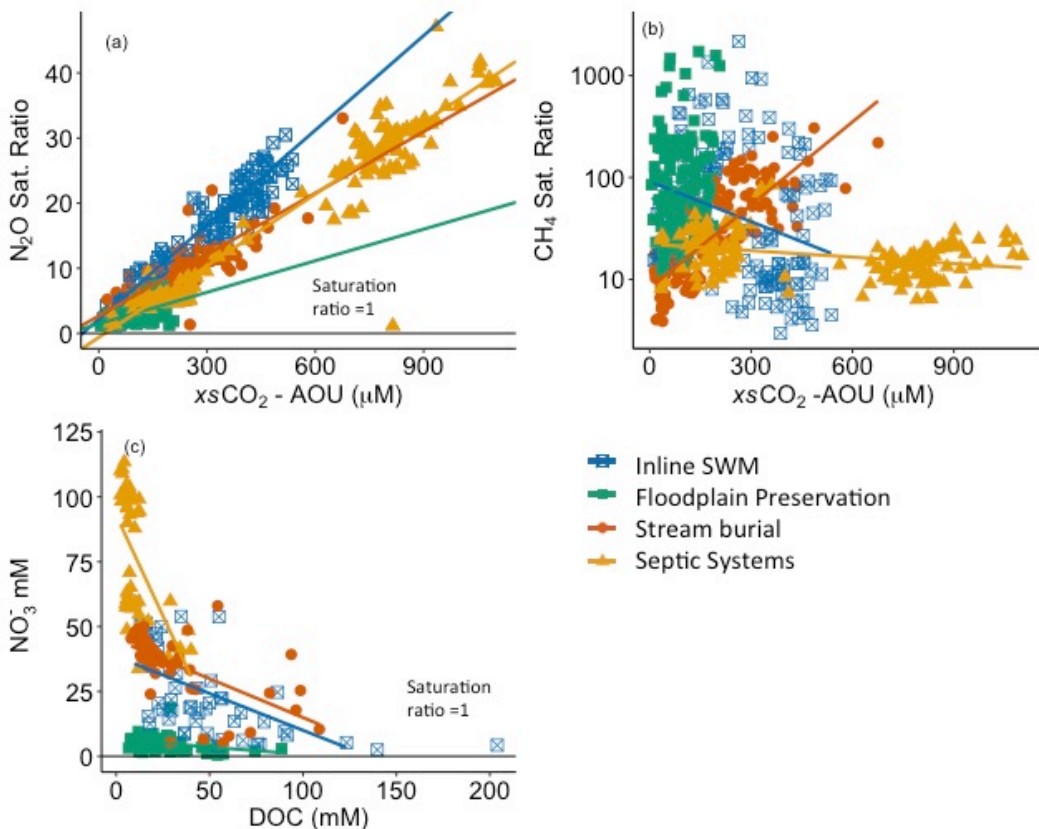

**Figure 4: Scatterplots of a) $N_2O$ saturation vs. $xsCO_2$-AOU ($\mu M$) $CH_4$ saturation vs. anaerobic $CO_2$, and c) relationships between $NO_3^-$ and DOC. Lines denote significant (p<0.01) correlations among gas or solute concentrations, which vary by infrastructure category.**

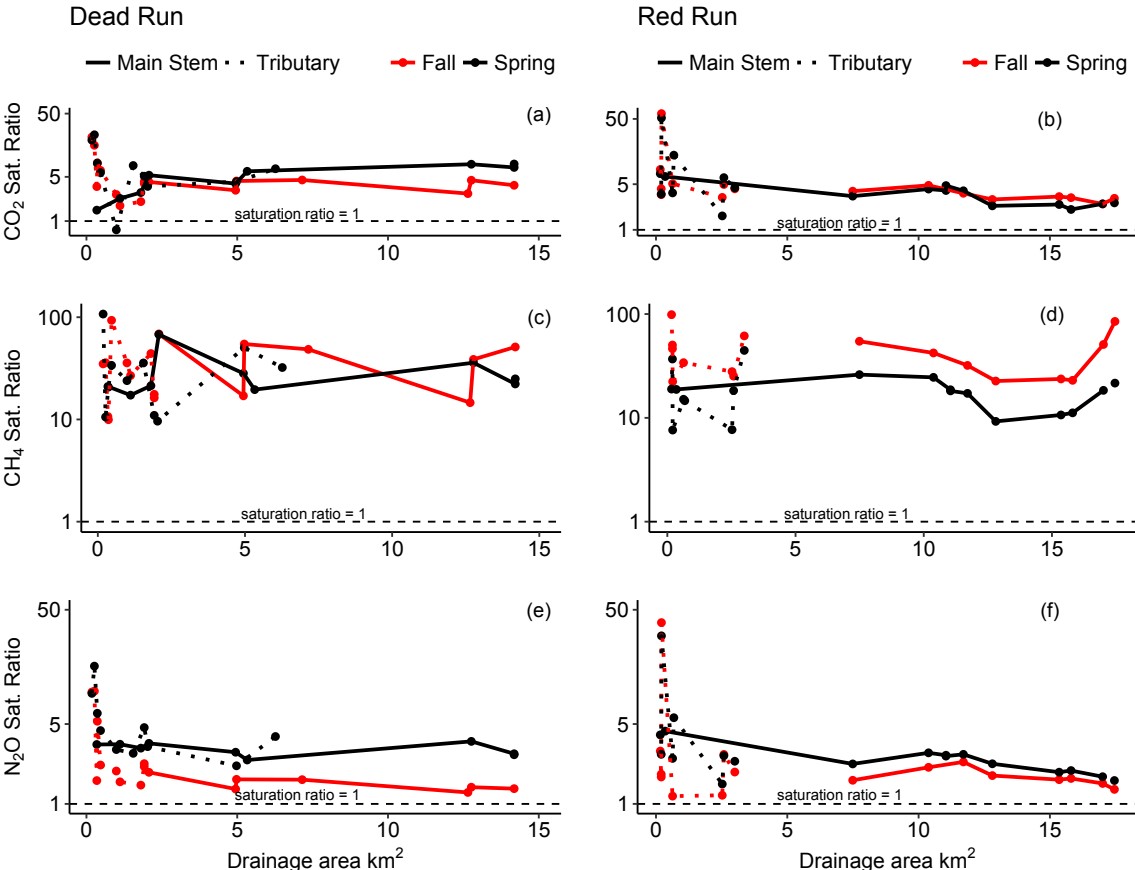

**Figure 5: Longitudinal variability in CO$_2$ (a-b), CH$_4$ (c-d), and N$_2$O (e-f) saturation ratios from spring and fall synoptic surveys of Dead Run and Red Run. Dotted lines denote tributaries to each watershed, while straight lines denote the main stem sites.**