# Peer review of "Influence of infrastructure on water quality and greenhouse gas dynamics in urban streams"

_Biogeosciences, 2016_

## Referee Comment (RC1) · Anonymous Referee #1 · 27 Oct 2016

Review for the following manuscript:
Journal: BG
Title: Influence of urban infrastructure on water quality and greenhouse gas dynamics in streams
Author(s): R. M. Smith et al.
MS No.: bg-2016-380
MS Type: Research article

First, my apologies for taking so long on this review. I had a personal crisis and lost track of my work responsibilities for a bit. I also apologize for the length; please respond in writing to comments only, and not all supporting examples.

**General comments:**
Summary for editor: This paper examines greenhouse gas saturation and emissions in urban streams, both across outflows of different green infrastructure, and spatially, across stream orders, within two stream networks. It finds super-saturation and emission levels comparable to or above those previously found in agricultural headwaters. Spatial variation is apparent, but not clearly analyzed.

Comments: This paper makes a substantial new contribution to understanding of greenhouse gas (GHG) emissions from urban headwaters. It is not clear what proportion of global GHG emissions might come from urban headwaters, and perhaps this global climate change framing used by the authors is not the most or only significant implication of their project data anyway. That said, the paper does point to biogeochemical and infrastructure controls on those emissions in a novel way. The data and its apparent trends as shown graphically are interesting. However, the paper as currently written has severe flaws, including substantial missing pieces of methodology, a multiple comparisons problem and lack of statistical analysis of part of the data that make any statistical interpretation of the results questionable at best, and significant room for improvement in clarity of the writing. I cannot recommend that this paper be published without heavy revision. It definitely appears to contain biogeoscientific information meriting publication, but it is impossible to say for sure from the current presentation of that information.

**Specific comments:** Numbers preceding examples refer to page number (line number). Please do not respond in writing to every example; just responding to each comment will suffice. The comments are roughly ordered from most to least concerning.

Comment 1: This paper appears to lack some methodological information, some of which is important and makes it difficult to assess what you did. For example:
- 4(22-24) and Table 1: You may want to explain why you decided to treat these watersheds as four categories of two replicates each, rather than eight watersheds varying continuously along a few axes (impervious surface cover, development age, etc.). I think the reason of different discrete stormwater infrastructure design types going with developments built at different times makes sense; you just might want to state it a little more explicitly.
- 4(26-28): Over what time period (i.e. year(s), season(s)/month(s)?, times of day?) Actually, you should probably give much of this this information earlier than this section, and I don't think you did.
- 5(5): How did you define a study reach? Approximately how long were study reaches? This information should come up in the previous section.

- 5(8): Is this the timing for the methods described in section 2.1.2 also?  If so, say so, earlier.
- 5(17) & 8(21): "Estimated using Google Earth software" sounds a bit sketchy.  If you must mention Google Earth, include a citation for the program.  Ditto at 8(21)), and also, what's the precision on the Google DEM, and why didn't you use the lidar one mentioned in 5(19-20); is it not more precise?
- 5(17-20): There are multiple ways to make these calculations; what actual commands or tools did you use to do this?
- 8(14, 17, & 24) & 9(1-4): What is $K_{20}$?  You did not previously explain what GT (from $K_{GT}$) means in general terms, so if that explanation was supposed to translate, it does not do so effectively.  Ditto with $K_{SF6}$ and plain K; are those at ambient temperature?
- 8(20) You say you, "measure[ed] the change in elevation over a reach with a handheld GPS unit."  Isn't elevation from GPS units usually rather unreliable?  Describe the precision of your GPS unit.
- 12(26): Can you not distinguish (or at least venture an educated guess) between "C and N inputs and/or microbial metabolism," based on measurements/calculations of these gases individually, together with those of other gases?
- Missing: How did you analyze "longitudinal variability," or the effect of "distance from watershed outlet," on any of the response variables, i.e., the output of the method described in section 2.1.3?  You make claims about the results of this survey in section 3.6 and display graphs derived from the data in Fig. 5, and then about the significance of these findings in 14(10-18).  However, it's never apparent that you did more than eyeball that data to assess spatial trends.  Moreover, my eyeballing does not match your eyeballing; I don't see Fig. 5 as reflecting the patterns you describe in the text.

Comment 2: In your statistical methods (section 2.4, "Statistical Analyses,") you execute a number of models (linear mixed effects, stepwise linear regression, etc., yielding all the results in Table 2 and 5) testing similar or related things.  This may constitute a statistical multiple comparisons problem, i.e. increased chance of Type I error (https://xkcd.com/882).  Consider either combining models (e.g. in a structural equations modeling framework or similar) or correcting for this risk of error.  At the very least, try to combine your categorical and continuous variables for into a single model for each gas.

Comment 3: Some interpretations of your results, most but not all minor, don't entirely make sense, or seem incomplete.  For example:
- Section 3.6 and 14(10-14):
- 12(16-17): Are you sure the "influence" is actually "indirect" on "biogeochemical processes in streams," or does the "indirect" part really only apply to GHGs?  It seems like those things listed are directly related to biogeochemistry in general.
- 12(23): Plain "nitrogen" or "*inorganic* nitrogen?"
- 13(9-10): "stoichiometric conditions more favorable for denitrification" would be a DOC:nitrate ratio closer to 1:1?  If that ratio is different in incoming groundwater, wouldn't the $N_2O:CO_2$ ratio from that groundwater be correspondingly different as well?
- 13(24-25): You've made a big jump here, from relatively high emissions in certain places to "globally significant."  Consider reminding your reader ("reminding" insofar as this should go in the introduction first; currently it's all just missing) what it would take for these locally high emissions to be globally significant- what's the

relative global contribution of streams in general; how much of global streams is urban stream, etc. It might make more sense to think of the impacts of $NO_2$ emissions in the city in terms of local air pollution than global GHGs. You might also think about if your findings suggest anything new for general biogeochemistry, as opposed to just the GHG emission application.

- Section 3.5 and Fig. 4b: Why do you think the slope directions of the lines in Fig. 4b so variable? Address this in discussion.
- 14(10-18): See comment after "missing," in Comment 1; it is unclear if you did a statistical analysis to support these claims.
- 15(17-18): "Variation in nonpoint sources and flowpaths" is not really an independent variable you tested; you don't know what in the watershed, but outside the stream, is driving anything, beyond a bit of inference about groundwater.
- Table 5: You never interpret your $K_{20}$ results in the discussion.
- To put your results in context a bit better, see Gallo et al. 2014 ("Physical and biological controls on trace gas fluxes in semi-arid urban ephemeral waterways" in Biogeochemistry 121(1) pp.189-207). They did related measurements in ephemeral streams in urbanized deserts, with similar results. For just nitrous oxide emissions from urban streams, there are several more relevant papers; try searching "nitrous oxide urban stream," in Web of Science if you can. (No, I am not Gallo et al.)

Comment 4: You refer several times to a gradient or continuum of stormwater infrastructure, but you never elucidate the relationships between or ordering of the infrastructure types that makes them constitute a gradient or continuum. Explain, up front and early. For example:
- 2(29): Is the "along the urban watershed continuum" significant? Does something change along this gradient about the effect of the wetlands, or do you just mean "in urban watersheds?"
- 1(16): It is not immediately clear how these seemingly discrete categories constitute a "a gradient of stormwater and sanitary infrastructure"- gradient along what axis, what variable?
- 3(20-21): "Urban watershed continuum" again- is that just a way to refer to the stretch from the infrastructure in the headwater downstream a bit, or are the different kinds of infrastructure arranged along a continuum, or what?
- 5(22): This is the closest thing to an explanation you've made so far, and it still doesn't really make sense.

Comment 5: You could improve this paper by reducing vague and occasionally careless diction. Sometimes this problem makes your meaning somewhat unclear. For example:
- 1(22): "These variables" refers to the "drivers of GHG dynamics," "infrastructure categories," or both? If it's the former, I guess this line just verifies that "nitrogen stoichiometry" etc. *are* in fact "drivers of GHG dynamics" in this context (as expected); if "these variables" are the "infrastructure categories," then it's a much more novel finding.
- 2(16-17): When you talk about GI here, are you proposing that all GI will have the same effects, at least in terms of direction of change in GHGs, or might effects differ depending on GI type?

- 2(20): "Source of uncertainty" for what?  Do you just mean "uncertain," or do you mean that this role could change our understanding of global fluxes from rivers, or what?
- 3(5-7): Reconsider word choice of "control;" option 2 doesn't seem to be an instance of control per se.  "Determine?"  Here is a spot where you could probably get away with one of those less specific verbs.
- 3(10): Specify *anaerobic* nitrification; this is unclear until 12(29). With plain "nitrification," it at first seems like $N_2O$ must be a typo for $NO_2^-$. You also need a source here for the description of nitrification; I don't think Taylor and Townsend 2010 suffices.
- 3(18): "GHG emissions"- what about them?  "Increased GHG emissions?"
- 3(28): Go ahead and be more specific than "water chemistry" if you can do so concisely.
- 4(8): "Reflects" what?  I think you mean the timing of development.  Maybe rephrase: "…developed in the 2000s *with* more infiltration-based designs…"
- 4(18): Maybe "…exists in various forms*, including* gravity sewers and septic systems*,* as well as a gradient…" or "…exists *as both* gravity sewers and septic systems *along* a gradient…"rather than the current, more ambiguous, "…exists in varying forms (gravity sewers and septic systems) as well as a gradient…."
- 11(28): "Consistent along the drainage network for Red Run and Dead Run": do you mean looking intra-Red Run drainage network and intra-Dead Run drainage network, or are you looking at both together as part of a larger drainage network?  I think you mean the former, but your phrasing is unclear?
- 12(25): Instead of "Varying forms," just "form."
- 12(27) & 15(13): Provide a citation for "'hot spots'" if you're going to put it in quotes, so we can verify which definition of "hot spot" you mean.  Also, decide if you're going to say, "'hot spot'" or just "hotspot;" be consistent.
- 12(30): "The source," or just "the primary source," or "a source?"
- 15(1): By "variations" you mean "differences?"
- 15(6): "Methodology" or "assumptions" (or "methodological assumptions")?
- 15(20): "Ecological?"  What does that mean here?
- 15(23): "Role" or "influence?"

Sometimes your point could be stronger if you provided concrete numbers to back up your assertions.  For example:
- 2(3-4): Consider fleshing out "globally significant" with some actual numbers?  Also, if you have space, it might not hurt to explain very briefly how this impact of rivers and streams on GHGs was determined.  It is unclear here whether the figures you cite include urban streams or not, and why.  In other words, could knowing about urban stream GHGs make these fluxes more or less "globally significant?"  Without this piece of information, it is unclear if all of the potentially contributing factors to urban stream GHG emissions that you describe in the rest of the paragraph are already accounted for in the currently accepted stream GHG numbers and you're just partitioning sources, or if you might revise the numbers on stream GHG fluxes as a result of this study.
- 2(13): What does "substantially" mean?  Can you provide numbers as to the relative contributions of nonpoint and point sources?
- 3(22-24):  How is human population relevant?  Also, please contextualize "fastest form of land use change;" that statement alone isn't really enough to ascertain significance.  Is the magnitude of the change (i.e. first derivative of land use rather

than second derivative) large? Is urban land use large, relative to other uses? Or do you think urban watersheds contribute disproportionately much to GHGs for their size, and so are significant globally even if small?

- 12(6): Which were the "three high-flow sampling dates?"

Sometimes you waste valuable space by not going ahead and saying what you actually mean. For example:

- 1(27-29): Your concluding sentence is rather vague; for a start, "influenced" could mean almost anything. Could you be a bit more specific about what the "influence" was and what the "implications" are?
- 2(9): Again, on "implications," try to be less vague if you can do so concisely. "Increase or decrease" or "change the magnitude of?" "Alter seasonality of?" Etc.
- 14(5): By "relative proportion of different gases," do you actually mean "methane production?"
- 15(1): By "typologies" you mean "types?"

Comment 6: Remember to maintain coherence and clarity of the paper through clear transitions, linking similar ideas, defining terms the first time you mention them, etc. For example:

- Abstract: You don't describe your "longitudinal" results here (the ones along stream length).
- 2(21-23): How do these numbers/methods for calculating global fluxes that you cite here compare to the ones in 2(2-3)?
- 2(24-25): Consider "Some key differences *between the watershed types that might affect this relationship* include," for clarity. Alternatively, "…may differ substantially *between* urban and agricultural watersheds *due to contrasting biogeochemistry and hydrology*. Some key differences…"
- 2(25-26): For clarity, consider something like, "*In urban watersheds, these factors likely vary with* stormwater and sanitary sewer…"
- 3(5): Consider ending this sentence with an "as well," or similar to tie back to previous sentence.
- 3(29)-4(2): The final sentence in this paragraph seems out of place. Maybe shift it to the start of the next paragraph and end with, ", which facilitated site selection," or something? If you don't move the sentence, at least go ahead and explain why this information store matters. I mean, I can guess, but I shouldn't have to do so, or to wait until you bring it up again later. Maybe just collapse the first two paragraphs into one?
- 4(5-6): Clarify timing. Everything was put in place in the 1950s-1970s, and the aging and cracking is now (or rather, when this study was conducted)? Also, "between" or "from?"
- 4(13): Remind us *which* eight streams- "…the eight streams *studied* drained…?"
- 4(14-20): Some of this description of what types of infrastructure were built when might go better in the introduction. Or at least, you might want to introduce the concept of change in design through time in the introduction.
- 4(12-16): This sentence has a bit of a run-on feel; consider breaking down. Also, does "stormwater infrastructure… encompass older designs" *and* the newer GI ones? The way the sentence breaks doesn't suggest so. You could say, "We define stormwater infrastructure broadly to encompass older designs such as stormwater drainage networks and newer forms of 'green' stormwater infrastructure (GI)," and then define each in a sentence (or so) each.

- 5(20): Unclear how GIS calculations in previous sentence are used; abrupt transition back to "these surveys" is hard to follow.
- 5(25): "Relative contributions of inflow" *to groundwater*?
- 12(30)-13(1): Consider referencing figures here (and more elsewhere in the discussion) to make it easy for readers to look back at the ratios etc. that you mention.
- 13(27)-14(9) & 15(5-9): Most of this information should go in the introduction. You can refer back to it here insofar as your findings update or add to it, but it's unclear that they do. It does not seem entirely relevant here.
- 14(16): "Detailed information" is not in itself a "step;" you need a verb, e.g. "*Finding* detailed information."
- 14(3): You could use "however" or another transition word before "these."
- 14(31-32): You do not make it clear how this information about plants is relevant. Are you saying that some other type of plant within the waters you surveyed might be releasing methane in this way, but you didn't measure it? There are no transitions into or out of this part about the plants, either.
- 15(9-11): This sentence goes with the end of the last paragraph.
- 15(26-27): It is unclear how exactly this part about wastewater relates to your results. Either make your transitions more clear, or move this sentence to a different section.
- 15(28): You have not brought up the concept of mitigation before, and it isn't immediately obvious if mitigation per se is the goal, or how your results translate to doing mitigation. Elaborate.

**Technical corrections:** Again, numbers preceding comments refer to page number (line number). Please do not feel obligated to respond to all of these; just make sure you have them the way you want them in the final version.
- 1(30): "Infrastructure" misspelled. Also, consistent capitalization of keywords?
- 3(9-10): Instead of, "nitrification is a chemoautotrophic process that produces," you could just say, "nitrification chemoautotrophically produces," (and then switch ", and consumes" to "and consuming") for brevity.
- 3(27), 5(20), & 7(22-23): Is just sticking a web link in here appropriate? For 5(20) and 7(22-23) especially, I think you need proper citations.
- 3(28): ", which" would be more grammatically appropriate than "that."
- 4(7): "In-line?" Repeats throughout document- just make sure you want "in-line" and not "inline" or "in line."
- 4(11-12): Maybe "and" instead of "that are;" the phrasing of this sentence is a bit awkward. Also, I think you could avoid the passive tense of "are located" ("exist?").
- 4(26): "First-order streams" instead of "first order streams," yes?
- 4(27-28): I'm not sure why you repeat all the categories when you just said them and even *said* that you just said them. Also, here you capitalized the categories and put apostrophes around them, whereas you didn't in the last sentence; pick a format, and be consistent.
- 4(32): "Septa" or "septum?"
- 5(3), 7(16), 10(11), & 13(26): Remove tab for consistent paragraph formatting.
- 5(3-4): Consider rephrasing for clarity and brevity, e.g.: "A single stream water sample was collected in a 250 mL high-density polyethylene bottle at each site. One sample duplication rotated site each sampling date."
- 5(10): Unnecessary "to."

- 5(15-16): Can shorten slightly by removing passive tense, i.e. "USGS provided discharge data." Also, consider providing a citation for the USGS data here.
- 6(9): "To *the* University?"
- 6(12): "Underestimates" or *underestimations*? Also, what "it" refers to is a bit unclear.
- 6(13 & 24), 10(3), 11(23), & 13(7): "*Via*" and "*vs.*" need not be italicized.
- 6(16 & 19): Move "(DOM)" up to first use.
- 6(19-20 & 27-28): You essentially describe what molecular weight characterizes which source twice in a row, and do it better the second time; condense.
- 7(4): "Eq.'s?" Maybe just write it out.
- 7(4): "Rations" or "ratios?" (Pretty sure you mean "ratios.")
- 7(5): If you must put a comma before "($\mu$mol L$^{-1}$), I think you need one after too.
- 7(11) & Table 1: Combine things in parentheses in "(Eq. 3) (Stumm and Morgan 1981)." Similar change needed at end of caption for Table 1.
- 7(19): "-" may be unnecessary.
- 8(4): "In" or "at?"
- 8(7): "From" or "by?"
- 8(8): "Were," not "where."
- 8(8-9): "Would be indicative of" can be shortened to "would indicate" or even "indicates." You could also remove, "other $CO_2$ sources, namely."
- 8(26): "P= " or "p=?"
- 8(27): Provide units again for "±0.058."
- 9(13): Escaped ")."
- 9(19-20): Lost sentence fragment.
- 11(9): Second comma unnecessary. Also, why "may be," and only in second alternative explanation?
- 12(16): "Typologies however," should probably be, "typologies, however."
- 12(22): You can shorten, "were present across all four infrastructure typologies (Fig. 4c), which suggests," to "present across all four infrastructure typologies (Fig. 4c) suggest."
- 12(30): "Concentrations suggest that" should be, "concentrations, suggesting that."
- 13(24): "Warrants," not "warrant."
- 14(23): "With DOC:NO$^{3-}$ while other" could use a comma in the middle (i.e. "with DOC:NO$^{3-}$, while other."
- 15(1-2): Isn't there just the one negative relationship? ("The negative relationship" instead of "negative relationships.")
- Table 1: Header word spacing is awkward.
- Table 4: In caption, "* Indicate" should be something like, "A '*' indicates," based on comparable sentences elsewhere.
- Table 5: You may be missing some commas towards the end of the list in the caption.
- Figure 1: "Sampling sites and black dots signify" should have a comma after "sites."
- Figure 2: "Points signify data points," in the caption is a bit confusing; consider removing the second "points."
- Figure 3c: I know it will mess with the clarity of your outliers, but consider some kind of log scale here; the differences between the actual boxes and whiskers are almost completely unapparent.

- Figure 3: In caption, "box and whiskers signify the median, first and third quartiles," is unclear phrasing.  At minimum, I think "box" needs to be plural.
- Figure 5: Consider combining identical keys for panels (e) and (f), and perhaps some of the identical axes across panels as well.  Unpunctuated letters representing figure panels within the caption text, e.g. "in panels a through d signify a saturation," are confusing; "a" is also a word.  Also, more specific date here?
- 15(16): "Of aquatic ecosystems" is in the middle of a list which relates to it (either end would make more sense), and the "as well as" and "significantly alter" seem unnecessary; commas would do.
- 15(25): "Include" not "includes."

---

## Referee Comment (RC2) · Anonymous Referee #2 · 4 Nov 2016

GENERAL COMMENTS The manuscript examines greenhouse gas (GHG) dynamics in urban streams, a relevant but relatively understudied topic in biogeochemistry, with potentially relevant implications for global GHG budgets. The manuscript presents interesting results on $CO_2$, $CH_4$ and $N_2O$ concentrations and emissions in several streams with different types of urban infrastructures. It is found that potential drivers of GHG dynamics (e.g. carbon, nitrogen, oxygen concentrations) differed among infrastructure types and were related to $CO_2$, $CH_4$ and $N_2O$ supersaturation in stream water. Moreover, $N_2O$ saturation ratios measured in these urban streams were among the highest ever reported for streams. In general, the manuscript is well written and potentially interesting for the readers of the journal Biogeosciences. However, there are some important caveats, which I briefly list here and develop more in specific comments below: - Some strange terms are used throughout the text that could be avoided

(e.g. "watershed continuum", anaerobic concentration) - The role of external (non-in-stream) and non-biological sources of GHG is not well considered in the manuscript. This may also make some calculations such as the index of aerobic and anaerobic respiration inaccurate. - Some parts of the methods need clarification (e.g. supersaturation, DOM sample preservation). In addition some parts of the methods seem unnecessary given the results that are presented - The dynamics of CO2 are not considered in the discussion section - Reference to relevant recent studies on GHG dynamics in urban streams are missing (e.g. see Alshboul et al. 2016 Environmental Science & Technology 50: 5555-5563 DOI: 10.1021/acs.est.5b04923 and references therein).

SPECIFIC COMMENTS Title: I have the feeling that something is missing in the title. Maybe the word "of" before "urban"? P1, L17: Unclear what is meant by "watershed continuum". I think it would be more correcto to speak about river network. This study focuses on the river and not on the whole watershed. This should be clear throughout the manuscript. P1, L23: Not sure these r2 values are helpful here. It is not clear which statistical test was used. P1, L26: Again, unclear use of r2 value. P1, L29: This last sentence of the abstract does not seem appropriate. It refers to emissions, which are not the focus of the manuscript. I would rather include a more conclusive sentence here. P2, L4: Land use can alter GHG emissions from streams not only through changes in drivers of stream metabolism. Changes in external GHG sources (e.g. groundwater inputs, soil leaching, point sources) and some geochemical reactions may also be important. In general, only part of GHG emissions from streams come from in-stream metabolism. This relevant aspect is not made sufficiently clear in this manuscript. P3, L20-24: Yes, but how much do streams contribute to whole watershed GHG fluxes? P5, L1: Please specify what blanks are here. P5, L5: Unclear what is meant by "study reach". It has not been defined. P5, L26: Not sure this equation and the associated text are necessary according to the results shown later. P5, L29: What about minor tributaries? Define better what you mean by major tributary. P6, L10-12: Specify how TDN and DOC were analyzed. P6, L16: $0.7\mu$m-filtered samples stored for 2 weeks seems inappropriate for a DOM composition analysis. $0.2 \mu$m fil-

tering is usually preferred. P6, L29: Why use a new name for this index if BIX is the name normally used? P7, L25 to P8, L11: This index seems controversial and needs clarifications. Not sure it can be really applied because apparently, it does not take into account external (non-in-stream) GHG sources and non-biological GHG sources. P7, L13: Remove "and" before "flux"? P7, L23-25: Unclear. Please explain better how Cesc was estimated from SF6 additions. P11, L1: This subtitle is repeated 3 times in this page. P11, L21: The term "anaerobic $CO_2$ concentration" seems erroneous. It does not make much sense. The same applies for anaerobic $N_2O$ or $CH_4$ concentrations. P15, L25-28: I suggest the authors try to include more results-based conclusions and implications at the end of the paper. It also seems confusing that the authors emphasize wastewater here, when the paper is about streams and GIs. Tables & Figures: For greater clarity, I suggest keeping the same order for the 3 solutes ($CO_2$, $CH_4$ and $N_2O$) in all tables and figures as well as in the text. Table 1: I do not think so many decimals are necessary for most of these variables. Table 2: "0.000" = "<0.001" or "<0.0001"? Table 4: If some variables were log-transformed (e.g. logDOC:NO3), this should be indicated in the methods section.

---

## Author Comment (AC1) · 18 Jan 2017

**BG 2016-380**
**Responses to Comments by Anonymous Reviewers 1 & 2**

"Influence of infrastructure on water quality and greenhouse gas dynamics in urban streams'

*We would like to thank the reviewers for their time in providing detailed, constructive comments regarding this manuscript. We have combined our responses to both reviewers' comments below, and believe that their contributions will lead to significant improvements. Both reviewers raised concerns about: 1) methodological details, 2) interpretation of results, and 3) terminology and clarity of ideas. Both reviewers outlined some overall comments with specific examples along with some technical line-by-line edits. As requested by the reviewers, we have combined our responses to each overarching comment with responses to specific examples where we deemed necessary below. We have not included direct responses to each line-by-line comment, but will incorporate these edits in the revised version of the manuscript.*

**I. METHODOLOGICAL INFORMATION**

*Both reviewers have expressed concern about the level of detail provided in the methods section, and certain specific methodologies used. We have compiled their general comments and replied to their examples where more than a simple textual response was deemed necessary.*

**R1 Comment 1: This paper appears to lack some methodological information, some of which is important and makes it difficult to assess what you did. Some of these examples of this are listed below.**

*And*

**R2 Comment 3) Some parts of the methods need clarification (e.g. supersaturation, DOM sample preservation). In addition some parts of the methods seem unnecessary given the results that are presented**

*In the submitted version of this paper, we described the different terminology for gas saturation in stream water (saturation ratio, or $xsCO_2$, $xsCH_4$, and $xsN_2O$) on page 7, lines 19-24.*

*In the revised version, we will add the following text to this section. "Super-saturation is defined as having a saturation ratio >1 or when $xsCO_2$, $xsCH_4$, or $xsN_2O$ is >0."*

*We are not sure which aspects of DOM sample preservation the reviewer finds to be missing from the manuscript; however, we will clarify our preservation methods with citations below. We will additionally review our methods section again after completing*

*all other text edits to ensure that only the relevant methods are reported.*

R2, P6, L16: 0.7micron-filtered samples stored for 2 weeks seems inappropriate for a DOM composition analysis. 0.2 micron filtering is usually preferred.

*In the original manuscript, we describe DOM sample preservation and analysis on page 6, lines 16-21. Following filtration through pre-combusted 0.7µM glass fiber filters, samples were stored in amber glass vials at 4°C and analyzed within 2 weeks following collection. To the authors' knowledge, this is an appropriate and commonly utilized filtration procedure for DOM fluorescence metrics. Glass fiber filter pore sizes are not available below 0.7µM, and smaller filter materials (such as 0.2 µM nylon) have the potential to leach out fluorescently active compounds and/or measureable amounts of dissolved organic carbon during filtration.*

*Numerous references are available outlining this filtration and storage procedure (Sing et al. 2014, Sing et al. 2015, Huguet et al 2009, Dubchick et al. 2010, Gabor et al. 2014). While none of these papers specifically discuss the length of time that samples can be stored, the 'two week rule' is a commonly used convention rather than a biologically based limit to storage (Personal communication, Rachel Gabor, Shuiwang Duan). However, we can acknowledge in the revised text that some highly labile compounds can break down within hours of collection, while recalcitrant DOM can take months to break down contributing to some uncertainty.*

R1: P4 L22-24 and Table 1: You may want to explain why you decided to treat these watersheds as four categories of two replicates each, rather than eight watersheds varying continuously along a few axes (impervious surface cover, development age, etc.). I think the reason of different discrete stormwater infrastructure design types going with developments built at different times makes sense; you just might want to state it a little more explicitly.

*The reviewer's understanding of our reasoning for treating watersheds as replicates of different categories is correct. We do attempt to explain the reasoning for development of infrastructure types (page 4, lines 22-24), and we will clarify this section as follows:*

*" ... We selected eight headwater streams, each of which drained one of four distinct groupings of infrastructure types. Watersheds drained by these streams fell into four categories, which were based on development age, stormwater infrastructure design, and sanitary infrastructure. These headwater stream sites were treated as four discrete categories rather than eight sites across a gradient based on similarities in the form and age of stormwater and sanitary infrastructure in the watershed of each stream site. A comprehensive description of attributes in each infrastructure type can be found in Table 1; however, for simplicity we have abbreviated the types based on the dominant infrastructure feature as follows: 1)*

*stream burial, 2) in- line stormwater management (SWM) wetlands, 3) riparian/floodplain preservation, and 4) septic systems.*

*We will additionally review the remainder of the text to ensure that the infrastructure groupings are not described as a gradient.*

R1: P4 L26-28: Over what time period (i.e. year(s), season(s)/month(s)?, times of day?) Actually, you should probably give much of this this information earlier than this section, and I don't think you did.

*We will add the following information to the methods section of the manuscript* (P4 L26-28)*: "Headwater stream sites were sampled every two weeks for both water chemistry and dissolved gas concentrations. Chemistry sampling took place for two years, between January 2013 and December 2014. Dissolved gas sampling took place every other week between July 2013 and July 2014. Sites were visited between the hours of 9am and 2pm."*

R1: P5 L5: How did you define a study reach? Approximately how long were study reaches? This information should come up in the previous section.

R2, P5, L5: Unclear what is meant by "study reach". It has not been defined.

*Both reviewers pointed out the need to define the 'study reaches' established at each headwater stream sampling site. We will clarify our study design in the methods section as follows: "At each headwater stream site, we took five gas samples along a 20-m reach characterized by uniform bed forms either downstream or upstream of a stream gaging station. Gas samples were collected with 120-mL syringes at 0, 5, 10, 15, and 20m from the fixed starting point.*

R1: P5 L17 & P8 L21: "Estimated using Google Earth software" sounds a bit sketchy. If you must mention Google Earth, include a citation for the program. Ditto at 8(21)), and also, what's the precision on the Google DEM, and why didn't you use the lidar one mentioned in 5(19-20); is it not more precise?

*This analysis was originally completed using estimations of elevation and distance along the stream within Google Earth Software. Due to uncertainties in the precision of the Google DEM, as pointed out by the reviewer, we will re-calculate these values using a 2-meter resolution LiDAR – based digital elevation model procured by the Baltimore Ecosystem Study. This will remove the reference to Google Earth here.*

R2: P5 L17-20: There are multiple ways to make these calculations; what actual

commands or tools did you use to do this?

*We will clarify in the methods section that we used Hydrology tools in the Spatial Analyst toolbox of ArcGIS in order to delineate watersheds above each sampling point for our headwater sites, as well as longitudinal sampling locations. We mapped each sampled location using latitude and longitude and used these as pour points in the hydrology tools workflow. Because sampling points were always co-located with road crossings in this urban watershed, we were able to acquire the latitude and longitude of sampling sites using Google Earth software (Google Inc. 2009). Headwater stream sites were mapped based on the top of the 20m reach. Watersheds were delineated using a 2-meter resolution DEM created from LiDAR collected by Baltimore County in 2002. We first corrected the DEM for spurious depressions using the "Fill" tool in the hydrology toolbox. Next, we calculated flow direction for each pixel of this filled DEM raster. We then used the Flow Accumulation tool to evaluate the number of pixels contributing to each downstream pixel. After ensuring that each pout point was co-located on the map streams (i.e. areas with flow accumulation >500 pixels), we used the 'Watershed' tool to delineate the pixels draining into each sampled location.*

R1: P8 L14, 17, & 24) & P9 L1-4: What is $K_{20}$? You did not previously explain what GT (from $K_{GT}$) means in general terms, so if that explanation was supposed to translate; it does not do so effectively. Ditto with $K_{SF6}$ and plain K; are those at ambient temperature?

*We will clarify our description of each equation accordingly. $K_{20}$ is K normalized to 20C for a given gas. $K_{GT}$ is K for a specific gas at ambient temperature. Equation (7) describes the relationship between $K_{20}$ and $K_T$.*

R1: P8 L20 You say you, "measure[ed] the change in elevation over a reach with a handheld GPS unit." Isn't elevation from GPS units usually rather unreliable? Describe the precision of your GPS unit.

*GPS units were used to determine the location of two points along the stream network. The GPS points were mapped and the distance between points was determined using GIS tools. A 2-meter resolution DEM (based on 2002 LiDAR provided by Baltimore County Government) was used to estimate the change in elevation between the two points.*

R1: P3 L28: Go ahead and be more specific than "water chemistry" if you can do so concisely.

*We will change this to "were sampled every two weeks for dissolved carbon and nitrogen concentrations as well as and dissolved gases."*

R2, P5, L1: Please specify what blanks are here.
*We will add detail here that we collected three gas blanks by pulling 25mL of helium from the same tedlar bag used for headspace equilibration*

R2, P5, L26: Not sure this equation and the associated text are necessary according to the results shown later.
*Reviewer 2 is correct that we do not discuss the results from this mass balance calculation later on and it could justifiably be removed. We will remove this, as well as panels (e) and (f) from Figure 5.*

R2, P5, L29: What about minor tributaries? Define better what you mean by major tributary.
*We will clarify in the text that major tributaries were those contributing more than 5% of the discharge to the main channel at a given point along the stream network.*

R2, P6, L10-12: Specify how TDN and DOC were analyzed.

*We describe how DOC and TDN concentrations were analyzed on page 6 lines 9-14. We will clarify that 'TDN' was measured using the 'TDN' method, which consists of high temperature combustion in the presence of a platinum catalyst, and clarify that the Shimadzu instrument was a "TOC Analyzer." If the reviewer would be willing to provide more detail about which information is missing we would be happy to clarify the methods further.*

R2, P6, L29: Why use a new name for this index if BIX is the name normally used?

*We will replace 'index of autochthonous inputs' with 'BIX' throughout the manuscript.*

R2 Comment 2) The role of external (non-in stream) and non-biological sources of GHG is not well considered in the manuscript. This may also make some calculations such as the index of aerobic and anaerobic respiration inaccurate.

R2, P7, L25 to P8, L11: This index seems controversial and needs clarifications. Not sure it can be really applied because apparently, it does not take into account external (non-in-stream) GHG sources and non-biological GHG sources.

*Reviewer 2 is correct that AOU does not account for non-biological sources of GHGs. We will clarify this assumption about using the index on page 8, lines 10-11 where we define AOU. We will clarify that that AOU differentiates between aerobic $CO_2$ and $CO_2$ of anaerobic or abiotic origin (and not anaerobic vs. abiotic origin). By using this index without an additional metric for abiotic $CO_2$, we must assume that the proportion of abiotic $CO_2$ is small and invariant across sites and dates sampled. Richey et al. (1988)*

*justified this assumption with the following statement: "At ambient conditions (pH 6-7, alkalinity of 500-1000 ueq), with dissolved free $CO_2$ of 100-150uM or higher, the $CO_2$ produced through respiration remains primarily as dissolved $CO_2$. Thus ionic equilibrium reactions can be neglected." Richey et al (1998)'s justification is not valid in all cases for our study, as pH measurements varied widely from 4.81 to 8.9, and site-average $CO_2$ concentrations were lower than 100uM on 20 out of 152 sampling sites and dates, and alkalinity was not measured. $CO_2$ and pH were only both within this range on 36 out of 152 occasions. Among these observations, there remains a significant, positive linear relationship between xs $CO_2$ and xs$N_2O$ (p= 8.36 $x10^{-15}$, $r^2$ = 0.83) across all sites. In the next version of this paper, we will repeat analyses that include AOU for this subset of samples.*

*We would like to clarify that this index does account for external (non-in-stream) $CO_2$ and $O_2$ sources, and this was our main reason for using the index. Regardless of whether $CO_2$ and $O_2$ are produced within the stream, in the soil, or along groundwater flowpaths, the ratio of these two gases within the stream will represent the relative abundance of $CO_2$ production to $O_2$ consumption along that flowpath. Richey et al. (1988) and Daniels et al. (2002) are two examples of freshwater-based studies that used this index to evaluate anaerobic $CO_2$ production in freshwaters.*

R2, P11, L21: The term "anaerobic $CO_2$ concentration" seems erroneous. It does not make much sense. The same applies for anaerobic $N_2O$ or $CH_4$ concentrations.

*In the original draft of this manuscript, we define 'anaerobic $CO_2$ concentration' in the methods on page 8 lines 10-11 as follows: 'Anaerobic $CO_2$ concentrations were calculated as the difference between aerobically produced $CO_2$ (assumed equivalent to AOU) and measured $CO_2$ concentration.' Anaerobic $CO_2$, as we define it, is just $CO_2$ that was not produced by aerobic respiration, which could also be abiotic. In response to this reviewer's comment, we will change terminology of this $CO_2$ source to be non-aerobic $CO_2$. We would like to additionally clarify that AOU is not used for any other gases ($CH_4$ or $N_2O$) and we do not make mention to 'anaerobic $N_2O$' or 'anaerobic $CH_4$' because, unlike $CO_2$, these gases are not produced and consumed in direct proportion to $O_2$.*

R2, P7, L23-25: Unclear. Please explain better how Cesc was estimated from SF6 additions.

*$C_{esc}$ is introduced in our manuscript in equation 6 in our manuscript ($K_{20}$= Cesc * S * V). When this equation is rearranged to solve for Cesc, Cesc = S* V/$K_{20}$. $K_{20}$ was calculated using measurements of SF6 off gassing conducted by Pennino et al. (2014). Briefly, because $SF_6$ is an intert gas, the loss of SF6 along a reach is proportional to the gas escape velocity ($K_{20}$). S (slope) at these injection sites was estimated using Google Earth imagery (though we will re-calculate this with 2m resolution DEM, based on Reviewer 1's comments) and V (velocity) was measured in the field. We will update the methods section with this added detail.*

R2: Table 1: I do not think so many decimals are necessary for most of these variables.

Table 2: "0.000" = "<0.001" or "<0.0001"?

*We will make our reporting of p-values more consistent (<0.001) throughout the manuscript.*

R2, Table 4: If some variables were log-transformed (e.g. logDOC: NO3), this should be indicated in the methods section.

*We will add detail about log-transforming the $DOC:NO_3^-$ ratio to the methods section.*

**II. STATISTICAL ANALYSES**

**Comment 2: In your statistical methods (section 2.4, "Statistical Analyses,") you execute a number of models (linear mixed effects, stepwise linear regression, etc., yielding all the results in Table 2 and 5) testing similar or related things. This may constitute a statistical multiple comparisons problem, i.e. increased chance of Type I error (https://xkcd.com/882). Consider either combining models (e.g. in a structural equations modeling framework or similar) or correcting for this risk of error. At the very least, try to combine your categorical and continuous variables for into a single model for each gas.**

*Reviewer 1 expressed concern about the statistical approach of using two modeling approaches to examine controls on each gas species citing that this approach seems redundant. The authors acknowledge that using two separate approaches for the purpose of predicting gas saturation values would increase the chance of Type I error; however, this was not the aim of our approach. The two models were used to examine first, whether or not there was consistent variation in gases across the categorical comparisons of watersheds, and secondly to examine whether or not gases could be predicted based on broader gradients in physical or chemical constituents that existed across all sampling dates and locations. In response to this reviewer's comments we have additionally incorporated a bonferroni correction to the p-value by dividing the 0.05 significance threshold by the number of models (6 models total, three for each gas), so that only tests with p< 0.0083 are considered significant. This does not change our results.*

**III. INTERPRETATION OF RESULTS**

**R1 Comment 3: Some interpretations of your results, most but not all minor, don't entirely make sense, or seem incomplete. For example:**

R1: 12(16-17): Are you sure the "influence" is actually "indirect" on "biogeochemical processes in streams," or does the "indirect" part really only apply to GHGs? It seems like those things listed are directly related to biogeochemistry in general.

*We will clarify here that, while watershed infrastructure was not a statistically significant*

*predictor of GHG saturation in streams, the gradients in DOC: $NO_3^-$ that we found across all infrastructure types was strongly correlated with GHG saturation. We interpreted this to mean that infrastructure may directly influence DOC and NO3- loading to streams, and that this C:N stoichiometry is likely to be an important controller of GHG abundance downstream. This is not to say that GHGs produced within sewers, stormwater wetlands, etc., are not important, but rather that the strongest correlations exist with continuous dissolved parameters rather than categorical.*

R1: P12 L23: Plain "nitrogen" or "*inorganic* nitrogen?"

*We will change the wording here to 'inorganic nitrogen'*

R1: P13 L9-10: "stoichiometric conditions more favorable for denitrification" would be a DOC: nitrate ratio closer to 1:1? If that ratio is different in incoming groundwater, wouldn't the $N_2O$:$CO_2$ ratio from that groundwater be correspondingly different as well?

*We are not sure we follow the reviewer's question here, however we see the need here to clarify our interpretation of DOC:$NO_3^-$ and $CO_2$:$N_2O$ ratios.*

*DOC: $NO_3^-$ stoichiometry is one way to examine whether biogeochemical conditions are favorable for one microbial process over another, as Taylor and Townsend (2010) describe in their in-depth metadata analysis of DOC: $NO_3^-$ stoichiometry across a wide range of ecosystems. Helton et al. (2015) also provide a comprehensive review of the ways in which stoichiometry between inorganic N and organic C can be interpreted in various ecosystems. The implications of this stoichiometry at small spatial scales, such as the stream-groundwater interface of headwater streams, can be more complicated, however, and we agree with the reviewer that our interpretation could be explained more clearly.*

*As noted by Taylor and Townsend (2010), a DOC: $NO_3^-$ ratio of 1:1 is ideal for denitrification, while DOC: $NO_3^-$ much below 1:1 signifies conditions favorable for nitrification. While this ratio reflects the biogeochemical condition at the location/time the sample was collected, it is the result of processes occurring along the upstream flowpath. In predominantly groundwater-fed streams, for instance, heterotrophic denitrification may consume significant proportion of DOC along groundwater flowpaths of a septic plume, thus drawing down the DOC: $NO_3^-$ of upwelling groundwater. Denitrification converts DOC to $CO_2$ and $NO_3^-$ to $N_2$ and $N_2O$. Numerous studies have*

*shown septic plumes to have high concentrations of NO3- (e.g. Aravena et al. 1993).*
*DOC concentrations are variable, but tend attenuate with depth in the aquifer and/or*
*flow distance along the plume (Aravena and Robertson 1998; Pabich et al. 2001). For*
*instance, Pabich et al. found high concentrations of DOC (>20 mg/L) in the upper part of*
*a septic plume, with an exponential pattern of attenuation with depth. Consistently high*
*$NO_3^-$- paired with attenuating DOC can result in a very low DOC: $NO_3^-$ ratio by the time*
*groundwater reaches stream. These conditions at the stream-scale are is more favorable*
*for nitrification. Since nitrification is a chemoautotrophic process, consuming $CO_2$ while*
*producing $N_2O$, we would expect to see a negative relationship, or no relationship*
*between $CO_2$ and $N_2O$ if nitrification were the dominant $N_2O$ production pathway in a*
*given watershed. Instead, we find positive correlations between $CO_2$ and $N_2O$ in nearly*
*all watershed sites (Figure 4a). We suggest therefore that denitrification may be*
*producing $N_2O$ in the groundwater in our septic-dominated sites, and drawing down*
*DOC: $NO_3^-$ along groundwater flowpaths. This interpretation remains hypothetical,*
*however due to a number of biotic and abiotic processes occurring at the same time.*
*Further work measuring solutes and gases along a groundwater flowpath is necessary to*
*identify the mechanisms producing high concentrations of $N_2O$.*

R1: P13 L24-25: You've made a big jump here, from relatively high emissions in certain
places to "globally significant." Consider reminding your reader ("reminding" insofar as
this should go in the introduction first; currently it's all just missing) what it would take
for these locally high emissions to be globally significant- what's the relative global
contribution of streams in general; how much of global streams is urban stream, etc. It
might make more sense to think of the impacts of $NO_2$ emissions in the city in terms of
local air pollution than global GHGs. You might also think about if your findings suggest
anything new for general biogeochemistry, as opposed to just the GHG emission
application.

*Rather than focusing on global emissions, we will point out here, and in other parts of the*
*manuscript, that diffuse emissions from urban streams constitute a previously*
*unaccounted for source of $N_2O$ and $CH_4$. It is currently unknown how significant this*
*source is, although one study shows that, for $N_2O$, sanitary sewers could emit as much*
*$N_2O$ per capita as current estimates for secondary WWTP plants (Short et al. 2014).*
*There is evidence that most of the $N_2O$-N found in these streams originates as*
*wastewater, and our study adds insight into the magnitude and variability of biogenic*
*gases in streams draining septic and sewer infrastructure.*

*We will also emphasize the point that greenhouse gas emissions from urban streams may*

*represent an important export pathway, for C and N from stream networks. Our results suggest that gaseous losses may need to be considered in urban watersheds from the perspective of mass transport and watershed C and N budgets.*

*Since we did not measure $NO_2$ emissions, we are unable to comment on whether or not streams are a source of that gas in our study sites.*

R1: Missing: How did you analyze "longitudinal variability," or the effect of "distance from watershed outlet," on any of the response variables, i.e., the output of the method described in section 2.1.3? You make claims about the results of this survey in section 3.6 and display graphs derived from the data in Fig. 5, and then about the significance of these findings in 14(10-18). However, it's never apparent that you did more than eyeball that data to assess spatial trends. Moreover, my eyeballing does not match your eyeballing; I don't see Fig. 5 as reflecting the patterns you describe in the text.

R1: P14 L10-18: See comment after "missing," in Comment 1; it is unclear if you did a statistical analysis to support these claims.

*The above two comments address our longitudinal study. We agree with Reviewer 1 that a more statistical approach to interpreting this data set is necessary. Our purpose for sampling gaseous and dissolved C and N along these watersheds was to determine whether or not the high $N_2O$ and $CH_4$ saturation values found in headwaters was specific to headwaters or ubiquitous throughout the watershed. We will remedy the current lack of numerical interpretation as follows: For a given sampling date, we will compare the range and coefficient of variation in $CO_2$, $N_2O$ and $CH_4$ saturation values in headwater sites to the range and coefficient of variation in main-stem sites. This will address the basic question that we set out to answer regarding spatial variability. We will additionally remove the water balance information from Figure 5, as this does not add any insight to our interpretation.*

R1: P15 L17-18: "Variation in nonpoint sources and flowpaths" is not really an independent variable you tested; you don't know what in the watershed, but outside the stream, is driving anything, beyond a bit of inference about groundwater.

*We will remove this sentence in the conclusion.*

R1: Section 3.5 and Fig. 4b: Why do you think the slope directions of the lines in Fig. 4b so variable? Address this in discussion.

*Overall, the relationships between $CH_4$ and $CO_2$ were much weaker and more variable than the relationships between $CO_2$ and N2O. We show this figure in part to demonstrate that $N_2O$ and $CH_4$ do not behave similarly in relation to $CO_2$. In response to this reviewer's comment, we examined potential drivers of the ratio of $xsCH_4 : xsCO_2$. Total dissolved N (TDN) was negatively correlated with $xsCH_4:xsCO_2$, and TDN concentrations explained 66% of the variance in this ratio, while DOC:TDN ratio only explained 53%. Differences in N availability across infrastructure categories may explain why the slope values and directions are so variable. One mechanism of this could be competition between $NO_3^-$ and $CO_2$ as terminal electron acceptors during anaerobic respiration.*

R1: Table 5: You never interpret your K$_{20}$ results in the discussion.

*We will add a brief comparison of our $K_{20}$ values with the literature as follows. Our estimated $K_{20}$ for $O_2$ spanned a wide range, from 1.0x 10-8 to 548. Raymond et al. (2012) performed a metadata analysis of all measured gas transfer velocities currently in the literature. In order to compare our values to theirs, we converted our $K_{20}$ units from 1/day to m/day by multiplying $K_{20}$ by water depth. Our calculated $K_{20}$ (m/day) values span the full range of their metadata analysis (4.1 x $10^{-10}$ to 179 m/day), with 95% of our measurements falling on the low end (i.e. below 10 m/day). This lower end of the range reported by Raymond et al. (2012) is consistent with their result that gas transfer velocity scales with stream order, as our sites were located in first order streams. We provide detailed calculations of $K_{20}$ in order to describe how GHG flux estimates were performed, however we decided not to discuss the $K_{20}$ values in the discussion because we were not specifically interested in $K_{20}$ as a variable on its own.*

R1: P12 L26: Can you not distinguish (or at least venture an educated guess) between "C and N inputs and/or microbial metabolism," based on measurements/calculations of these gases individually, together with those of other gases?

*We will add to the discussion section our speculation that the degree to which DOC: $NO_3$ in streamwater is driven by variations in C and N loading to the landscape vs. microbial processing along flowpaths depends on infrastructure. For instance, samples from streams draining septic systems had the lowest $DOC:NO_3^-$ ratios, and we believe this is principally driven by the low starting C:N ratio of wastewater, and paucity of carbon sources that intersect the septic plume flowpath. On the other end of the spectrum, stream draining 'floodplain preservation' typologies also had newer development, and thus potentially reduced influx of low C:N sewage into streams. At the same time, these watersheds also had highly connected riparian banks with organic-rich soils at the stream-riparian zone interface may also be hot spots of $NO_3$- removal via denitrification.*

R2: P2, L4: Land use can alter GHG emissions from streams not only through changes in drivers of stream metabolism. Changes in external GHG sources (e.g. groundwater inputs, soil leaching, point sources) and some geochemical reactions may also be important. In general, only part of GHG emissions from streams come from in-stream metabolism. This relevant aspect is not made sufficiently clear in this manuscript.

*We agree with Reviewer 2 that land use can alter external GHG sources to the stream, along with changing in-stream metabolism. In the present form of the paper, we make mention of external GHG sources on several occasions, however we do not specifically attempt to differentiate between external vs. in-stream GHG production as we do not have data to back up this type of analysis. In terms of potential external sources, we mention the role of external GHGs via groundwater flowpaths in the introduction (page 2, line 26) referring to the buildup to GHGs in groundwater that is connected with wetlands, as well as in line 31 on the same page, referring to direct leakage of gas from sanitary sewer infrastructure. We also discuss the role of N$_2$O produced via denitrification or nitrification along subsurface flowpaths on page 13 (lines 8-12) in the discussion section. The use of the term 'watershed continuum' in this section and others refers to the suite of flowpaths (surface and subsurface) by which sources of GHGs from infrastructure and the landscape are connected to the stream. We acknowledge that this point can be made more clearly in the manuscript, and will edit the text accordingly.*

**IV.  DESCRIPTION OF STUDY DESIGN**

**R1 Comment 4: You refer several times to a gradient or continuum of stormwater infrastructure, but you never elucidate the relationships between or ordering of the infrastructure types that makes them constitute a gradient or continuum. Explain, up front and early. For example:**

R1: P2 L29: Is the "along the urban watershed continuum" significant? Does something change along this gradient about the effect of the wetlands, or do you just mean "in urban watersheds?"

*We will change the language from 'along the urban watershed continuum' to 'in urban watersheds' here, as recommended by the reviewer.*

R1: P1 L16: It is not immediately clear how these seemingly discrete categories constitute "a gradient of stormwater and sanitary infrastructure"- gradient along what axis, what variable?

*We will clarify in the text here that these are indeed discrete categories of infrastructure, across which we found gradients in C:N stoichiometry, dissolved oxygen, temperature, etc.*

R1: P3 L20-21: "Urban watershed continuum" again- is that just a way to refer to the stretch from the infrastructure in the headwater downstream a bit, or are the different kinds of infrastructure arranged along a continuum, or what?

*The reviewer is correct that it is a way to refer to the flowpath from the infrastructure to the headwater downstream a bit. It is a term that explicitly incorporates infrastructure as part of the stream network in urban watersheds – the infrastructure/stream interface may play a significant biogeochemical role at a watershed scale in urban ecosystems. We have focused our study on understanding the role of urban infrastructure on greenhouse gas dynamics in urban waterways. A growing body of work has shown that nutrient and carbon loads to streams, as well as the biogeochemical processes within flowing waters is related to not only to land cover (% impervious surface, urban density, etc) but also urban infrastructure. Connectivity between runoff-generating water sources (groundwater, overland flow, shallow subsurface flow) and urban infrastructure (sewer lines, stormwater conveyance pipes, drinking water pipes, constructed wetlands, etc). is likely to influence not only the anthropogenic inputs of C and N to waterways but also the relative importance of biotic interactions on C and N removal along flowpaths. Kaushal and Belt (2012) describe a conceptual framework of how urban-impacted flowpaths may influence downstream export of nutrients as the 'Urban Watershed Continuum.'*

R1: P5 L22: This is the closest thing to an explanation you've made so far, and it still doesn't really make sense.

*In this section we are describing our sampling along the stream network. We will use the term 'stream network' here to clarify meaning. These specific changes are cited below.*

**V. VAGUE WORDING CHOICES**

**R1 Comment 5: You could improve this paper by reducing vague and occasionally careless diction. Sometimes this problem makes your meaning somewhat unclear. For example:**

*Both reviewers had concerns about some of the vague and unclear phrasing in sections of this paper. We respond here to their general comments as well as the specific examples from their line-by-line comments. Generally, we will clarify the key ideas underlying this paper in the introduction, provide more concrete details to back up statements about the literature, and link our interpretation of results more clearly to the figures and tables provided. The key ideas will be clarified in introduction include as follows:*

R2: Title: I have the feeling that something is missing in the title. Maybe the word "of" before "urban"?

*We will change the title to 'Influence of infrastructure on water quality and greenhouse*

*gas dynamics in urban streams'*

R1: P2 L3-4: Consider fleshing out "globally significant" with some actual numbers? Also, if you have space, it might not hurt to explain very briefly how this impact of rivers and streams on GHGs was determined. It is unclear here whether the figures you cite include urban streams or not, and why. In other words, could knowing about urban stream GHGs make these fluxes more or less "globally significant?" Without this piece of information, it is unclear if all of the potentially contributing factors to urban stream GHG emissions that you describe in the rest of the paragraph are already accounted for in the currently accepted stream GHG numbers and you're just partitioning sources, or if you might revise the numbers on stream GHG fluxes as a result of this study.

*We will flesh out the claim that streams and rivers emit globally significant quantities of greenhouse gases as follows: 'Flowing waters transport significant quantities of carbon and nitrogen from terrestrial ecosystems to the ocean. Along these flowpaths, rivers also emit significant quantities of biogenic gases. Inland waters, including rivers, lakes and reservoirs emit 1.2 Pg C $yr^{-1}$ of $CO_2$, equivalent to about half of the annual terrestrial carbon sink (Cole et al. 2007; Battin et al. 2009). Bastviken et al. 2011 recently estimated that inland waters emit 103 Tg $CH_4$-C $yr^{-1}$, the greenhouse warming equivalent to 0.65Pg $CO_2$-C $yr^{-1}$. Seitzinger et al. (2000) estimated that rivers, estuaries and continental shelves emit 1.6 Tg $N_2O$ N $yr^{-1}$, which is equivalent to nearly half of all $N_2O$ emissions from the ocean.'*

*We will additionally clarify that some of these studies do take into account $N_2O$ emissions from urban areas indirectly, by using population to estimate N inputs to watersheds (Seitzinger et al. 2000). There remain significant uncertainties in 1) the amount of N entering waterways from urban areas, 2) the proportion of N that is converted to $N_2O$ along groundwater and surfacewater flowpaths, especially in urban areas. These uncertainties may not be important at the global scale, but do impact watershed N budgets. For instance, Gardner et al. (2015) conducted a nitrogen input-output budget based on the difference between estimated anthropogenic N loading to the watershed and fluvial N export from streams. They found that outgassing of N ($N_2O$ + $N_2$) from the stream accounted for all of the missing N.*

R2 P1, L17: Unclear what is meant by "watershed continuum". I think it would be more correct to speak about river network. This study focuses on the river and not on the whole watershed. This should be clear throughout the manuscript.

R1 P3 (20-21): "Urban watershed continuum" again- is that just a way to refer to the stretch from the infrastructure in the headwater downstream a bit, or are the different kinds of infrastructure arranged along a continuum, or what?

*These two comments are related to the term 'urban watershed continuum.' We agree that, as presented in this paper, the urban watershed continuum is not clearly defined.*

*We will clarify in the introduction that this term is meant to describe expanded connectivity between infrastructure, landscape, and streams, which can influence biogeochemical functions (particularly at infrastructure/stream interfaces). We will additionally clarify throughout the paper which scale of connectivity we are referring to in order to more precisely present our findings.*

R1: P1(27-29): Your concluding sentence is rather vague; for a start, "influenced" could mean almost anything. Could you be a bit more specific about what the "influence" was and what the "implications" are?

R2: P1, L29: This last sentence of the abstract does not seem appropriate. It refers to emissions, which are not the focus of the manuscript. I would rather include a more conclusive sentence here.

*These two comments refer to the final sentence of the abstract. We agree with the reviewers that our study does not focus on emissions and will remove the last part of this sentence starting with 'with significant implications…'*

R2: P3, L20-24: Yes, but how much do streams contribute to whole watershed GHG fluxes?
R1: To put your results in context a bit better, see Gallo et al. 2014 ("Physical and biological controls on trace gas fluxes in semi-arid urban ephemeral waterways" in Biogeochemistry 121(1) pp.189-207). They did related measurements in ephemeral streams in urbanized deserts, with similar results. For just nitrous oxide emissions from urban streams, there are several more relevant papers; try searching "nitrous oxide urban stream," in Web of Science if you can. (No, I am not Gallo et al.)

*While it is beyond the scope of this manuscript to robustly quantify emissions from streams in this region we acknowledge that more context is necessary here to justify the scalability of our results. We will incorporate a 'back of the envelope' scaling exercise based on 1) range of flux estimates, 2) estimate of stream surface area and 3) soil GHG emissions from ongoing work at the Baltimore Ecosystem Study (Groffman et al. 2000; Groffman et al. 2009; Smith et al. in prep), and 4) a recent estimate of anthropogenic GHG emissions in Baltimore County (Brady and Fath 2008) in order to place stream GHG emissions in context with other notably larger sources. We will additionally note that, while GHG emissions from flowing waters are small compared to other anthropogenic GHG sources, they are clearly linked to water quality. It is therefore important to note that updating infrastructure may provide the dual benefit of improving water quality and reducing GHG emissions. We will certainly also incorporate citations to Gallo et al. (2014), as this study is highly relevant to the growing understanding of greenhouse gas production in urban aquatic environments.*

R2, P15, L25-28: I suggest the authors try to include more results-based conclusions and implications at the end of the paper. It also seems confusing that the authors emphasize wastewater here, when the paper is about streams and GIs.

*The paper examines both stormwater and sanitary infrastructure. We will clarify that we think our results present evidence that N loading and GHG emissions are related to sanitary infrastructure. We will introduce this idea a bit more clearly in the introduction and discussion.*

The following two comments refer to the same two sentences in the abstract:

R1: 1 L22: "These variables" refers to the "drivers of GHG dynamics," "infrastructure categories," or both? If it's the former, I guess this line just verifies that "nitrogen stoichiometry" etc. *are* in fact "drivers of GHG dynamics" in this context (as expected); if "these variables" are the "infrastructure categories," then it's a much more novel finding.

R2:  P1, L23:  Not sure these $r^2$ values are helpful here. It is not clear which statistical test was used.

*These two comments refer to the same two sentences in the abstract. On page 1, line 23, we are referring to the relationship between drivers of GHG dynamics (meaning the previously listed variables: C:N stoichiometry, dissolved $O_2$, dissolved nitrogen, and temperature) and $N_2O$ $CO_2$, and $CH_4$ from linear mixed effects models. We will clarify our meaning in the text as follows:*

*'While categorical analysis of infrastructure type vs. GHG saturation did not show significant differences among the pairs of watersheds, watersheds draining different types of infrastructure did yield strong gradients in continuous variables such as C:N stoichiometry, dissolved oxygen, dissolved N concentrations, and water temperature. Taken together in linear mixed effects models, these continuous variables explained 78%, 78% and 50% of variability in $N_2O$, $CO_2$, and $CH_4$ respectively.'*

R2: P1, L26: Again, unclear use of $r^2$ value.
*This line contains an error, as we did find significant differences in the relationship between $CO_2$ and $N_2O$ amongst infrastructure categories. The line will be changed to reflect the different $r^2$ values the relationship between $N_2O$ and $CO_2$ for each infrastructure category.*

R1: P2 L16-17: When you talk about GI here, are you proposing that all GI will have the same effects, at least in terms of direction of change in GHGs, or might effects differ depending on GI type?

*We will change the wording here from 'GI' to 'constructed wetlands and riparian*

*preservation' because that is what we are referring to here. We mean to hypothesize here that both of these practices may potentially increase $CH_4$ while reducing $N_2O$ production and emissions from streams.*

R1: P2 L20: "Source of uncertainty" for what? Do you just mean "uncertain," or do you mean that this role could change our understanding of global fluxes from rivers, or what?

*We will change this sentence as follows: "Despite considerable funds spent on restoring aging infrastructure and improving water quality in cities globally (Doyle et al. 2008), the role of urban infrastructure on in-stream GHG emission remains under-studied."*

R1: P3 L10: Specify *anaerobic* nitrification; this is unclear until 12(29). With plain "nitrification," it at first seems like $N_2O$ must be a typo for $NO_2^-$. You also need a source here for the description of nitrification; I don't think Taylor and Townsend 2010 suffices.

*We are not sure we understand the reviewer's comment here, as we do not use the term 'anaerobic nitrification' in this paper. As described on page 3, lines 9-10, nitrification is a chemoautotrophic process, which oxidizes $NH_4^+$ to $NO_3^-$. $CO_2$ is consumed during this process, and $N_2O$ is also produced as an intermediate in the $NO_3^-$ oxidation process. We have cited Taylor and Townsend (2010) because they provide an excellent framework for determining whether an environment is more favorable to nitrification over denitrification based on the ratio of $NO_3^-$ to DOC. We will add a more general reference about nitrification in aquatic systems to the text (Schlesinger 1997).*

R1: P3 L18: "GHG emissions"- what about them? "Increased GHG emissions?"

*We will change the wording here to "Increased GHG emissions"*

R1: P12 L27 & P15 L13: Provide a citation for "'hot spots'" if you're going to put it in quotes, so we can verify which definition of "hot spot" you mean. Also, decide if you're going to say, "'hot spot'" or just "hotspot;" be consistent.

*Upon reflection on Reviewer 1's suggestions about this analysis, we plan to remove the term(s) 'hot spot' in this section the paper and consistently separate it into two words where we do use the term.*

R1: P15 L23: "Role" or "influence?" Sometimes your point could be stronger if you provided concrete numbers to back up your assertions. For example:

R1: P2 L13: What does "substantially" mean? Can you provide numbers as to the relative contributions of nonpoint and point sources?

*We will change this sentence as follows, by moving some background information from the discussion into the introduction. "Several studies have documented that wastewater leakage from municipal sewers often accounts for more than 50% of dissolved N in urban streams (Kaushal et al. 2011; Pennino et al. 2016; Divers et al. 2013). While sewer lines are known to leak dissolved N, $N_2O$ losses are not accounted for in greenhouse gas budgets of large WWTPs that these pipes feed into. Short et al. (2014) measured intake lines from three municipal WWTPs and estimated that $N_2O$ emissions from gravity sewer lines alone on the same order of magnitude (1.7g $N_2O$ person $yr^{-1}$) as current IPCC estimates for per-capita emissions from secondary WWTPs. Their study demonstrates the importance of constraining biogenic gas emissions from streams which flow alongside aging sewer lines."*

R1: P3 L22-24: How is human population relevant? Also, please contextualize "fastest form of land use change;" that statement alone isn't really enough to ascertain significance. Is the magnitude of the change (i.e. first derivative of land use rather than second derivative) large? Is urban land use large, relative to other uses? Or do you think urban watersheds contribute disproportionately much to GHGs for their size, and so are significant globally even if small?

*We will remove this sentence and clarify as follows:*

*'Our study investigates patterns in GHG abundance and emissions from urban streams. This source of GHG emissions remains poorly constrained due to 1) heterogeneity of aquatic ecosystems within urban watersheds, and 2) uncertainties in emission factors (i.e. the percent of N added from a particular source that becomes $N_2O$) due to a range of N sources in urban streams (wastewater, atmospheric deposition, fertilizer). All of these sources, but especially wastewater, tend to increase with population rather than land cover explicitly. Wastewater N loading to rivers is projected to more-than double between 2000 and 2050 (van Drecht et al. 2009). While wastewater currently only comprises about 3% of anthropogenic $N_2O$ emissions globally (IPCC 2006), Strokal and Kroeze (2014) demonstrate that increasing population and thus N loading will almost certainly lead to higher $N_2O$ emissions, regardless of increased water treatment. It is therefore crucial to evaluate the ways in which highly managed urban watersheds process excess N and produce GHGs.*

R1: P12 L6: Which were the "three high-flow sampling dates?" Sometimes you waste

valuable space by not going ahead and saying what you actually mean.

W*e will list the three dates when high-flow conditions were sampled here.*

R1: P2 L9: Again, on "implications," try to be less vague if you can do so concisely. "Increase or decrease" or "change the magnitude of?" "Alter seasonality of?" Etc.

R1: P15 L1: By "typologies" you mean "types?"

*In the methods section (page 4, lines 20-24), we describe the combinations of sanitary and stormwater infrastructure in each pair of similar watersheds as 'typologies.' This could be changed to 'categories' if this helps to clarify our categorization of these watersheds based on infrastructure types.*

**VI. TRANSITIONS, DEFINING TERMS, ETC**

**R1, Comment 6: Remember to maintain coherence and clarity of the paper through clear transitions, linking similar ideas, defining terms the first time you mention them, etc. For example:**

**R2 Comment 1) Some strange terms are used throughout the text that could be avoided (e.g. "watershed continuum", anaerobic concentration)**

*We have taken these reviewers comments into consideration and will make changes to the wording, definition of terms and transitions of ideas throughout the manuscript. We have included examples of our response to these concerns below.*

R2: Abstract: You don't describe your "longitudinal" results here (the ones along stream length).

*Following the changes to our statistical analysis and interpretation of these results that we have described above, we will add a sentence to the abstract describing differences in the variance GHG saturation in headwater streams, compared with main channel sites.*

R1 P2 L21-23: How do these numbers/methods for calculating global fluxes that you cite here compare to the ones in 2(2-3)?

R1 P3 L29 - P4 L2: The final sentence in this paragraph seems out of place. Maybe shift it to the start of the next paragraph and end with, ", which facilitated site selection," or something? If you don't move the sentence, at least go ahead and explain why this information store matters. I mean, I can guess, but I shouldn't have to do so, or to wait

until you bring it up again later. Maybe just collapse the first two paragraphs into one?

*We will remove the last sentence of this paragraph.*

R1 P4 L5-6: Clarify timing. Everything was put in place in the 1950s-1970s, and the aging and cracking is now (or rather, when this study was conducted)? Also, "between" or "from?"

R1 P4 L13: Remind us *which* eight streams- "...the eight streams *studied* drained...?"

*We will clarify that we are referring to the headwater stream sampling sites, which are paired across eight infrastructure categories.*

R1 P4 L14-20: Some of this description of what types of infrastructure were built when might go better in the introduction. Or at least, you might want to introduce the concept of change in design through time in the introduction.

*We will incorporate more background related to urban sanitary and stormwater infrastructure into the introduction and pare down our methods section accordingly.*

R1  P4 L12-16: This sentence has a bit of a run-on feel; consider breaking down. Also, does "stormwater infrastructure... encompass older designs" *and* the newer GI ones? The way the sentence breaks doesn't suggest so. You could say, "We define stormwater infrastructure broadly to encompass older designs such as stormwater drainage networks and newer forms of 'green' stormwater infrastructure (GI)," and then define each in a sentence (or so) each.

*Reviewer 1 is correct in his/her reading of this sentence. We will clarify our meaning here as the reviewer has recommended.*

R1: P5 L20: Unclear how GIS calculations in previous sentence are used; abrupt transition back to "these surveys" is hard to follow.

*We used the latitude and longitude of sampling sites to delineate watersheds and also calculate distance along the stream network for each location. We will clarify this in the text.*

R1 P5 L25: "Relative contributions of inflow" *to groundwater*?

*This sentence should say 'relative contributions of groundwater inflow to discharge at a given sampling point.' However, upon reflection and based on the reviewer's comments, we will remove this hydrologic mass-balance analysis from the paper since it is not directly discussed or used to interpret GHG results.*

R1 P12 L30 - P13 L1: Consider referencing figures here (and more elsewhere in the discussion) to make it easy for readers to look back at the ratios etc. that you mention.

*We will add a citation for figure 4 here.*

R1 P13 L27 -P14 L9 & P15 L5-9: Most of this information should go in the introduction. You can refer back to it here insofar as your findings update or add to it, but it's unclear that they do. It does not seem entirely relevant here.

*We will move this background information about N loading to urban streams in Baltimore to the introduction.*

R1 P14 L31-32: You do not make it clear how this information about plants is relevant. Are you saying that some other type of plant within the waters you surveyed might be releasing methane in this way, but you didn't measure it? There are no transitions into or out of this part about the plants, either.

*We meant here to provide some potential explanation for why streams in Wilcok and Sorrell (2008) found such high $CH_4$ emissions compared with our streams. We can shorten this section, to simply say that their methane emissions are not quite comparable because they included measurements of fluxes from aquatic plants, which would skew the comparison. To add more detail, we could include that these aquatic plants increase methane fluxes from sediment to atmosphere because of holes in their stems (aerenchyma), which allow for diffusive gas exchange between the atmosphere and rooting zone.*

R1 P15 (26-27): It is unclear how exactly this part about wastewater relates to your results. Either make your transitions more clear, or move this sentence to a different section.

*We will re-frame these concluding paragraphs to clarify the transitions here as follows. We present evidence in this paper that N from septic plumes and sewer lines is the principal source of $N_2O$ saturation in our study sties. Dissolved inorganic N is highly correlated with $N_2O$ in our study sites, and the highest values are only present in*

*watersheds with aging sewer infrastructure or septic systems. Our observations of $N_2O$ saturation and emissions from urban and suburban headwater streams are some of the highest reported in the literature, comparable with streams and ditches in intensive agricultural watersheds (Harrison and Matson. 2003; Outram et al. 2012). These results suggest that streams draining low-density suburban or exurban land cover may be comparable to those in intensively fertilized agricultural areas in terms of $N_2O$ emissions, however further study is necessary to constrain emission factors in non-agricultural landscapes.*

R1 P15(28): You have not brought up the concept of mitigation before, and it isn't immediately obvious if mitigation per se is the goal, or how your results translate to doing mitigation. Elaborate.

*We will remove the mention of mitigation since this is not the focus of this study, however we will add emphasis on the importance of accounting for $N_2O$ urban streams (see our response to the previous comment).*

**R2 Comment 4) The dynamics of $CO_2$ are not considered in the discussion section**

*We did not focus on the $CO_2$ results in the discussion for two reasons. Firstly, $CO_2$ was strongly correlated with $N_2O$ (as mentioned in the abstract and elsewhere), so additional descriptions of the spatial and temporal patterns seemed redundant. Secondly, as Reviewer 2 points out, we do not have the data to take into account abiotic sources of $CO_2$ and are therefore cautious to compare absolute values across systems.*

**R2 Comment 5) Reference to relevant recent studies on GHG dynamics in urban streams are missing (e.g. see Alshboul et al. 2016 Environmental Science & Technology 50: 5555-5563 DOI: 10.1021/acs.est.5b04923 and references therein).**

*We will incorporate this reference and papers cited therein it into the section of our discussion linking wastewater to patterns in aquatic GHGs.*

**References Cited**

Aravena R, ML E, JA C (1993) Stable Isotopes of Oxygen and Nitrogen in Source Identification of Nitrate from Septic Systems. Ground Water 31:180–186.

Brady, P.A., Fath, B.D. (2008) Baltimore County Government Greenhouse Gas Inventory 2002-2006 Projections for 2012. Updated June 18, 2015, Accessed December 31, 2015. http://www.baltimorecountymd.gov/Agencies/environment/sustainability/ghgproject.html

Cole JJ, Prairie YT, Caraco NF, et al (2007) Plumbing the Global Carbon Cycle: Integrating Inland Waters into the Terrestrial Carbon Budget. Ecosystems 10:172–185. doi: 10.1007/s10021-006-9013-8

Daniel MHB, Montebelo AA, Bernardes MC, et al (2001) Effects of urban sewage on dissolved oxygen, dissolved inorganic and organic carbon, and electrical conductivity of small streams along a gradient of urbanization in the Piracicaba River basin. Water, Air, and Soil Pollution 189–206.

Divers MT, Elliott EM, Bain DJ (2012) Constraining nitrogen inputs to urban streams from leaking sewer infrastructure using inverse modeling: Implications for DIN retention in urban environments. Environmental Science & Technology 47:1816–1823.

Doyle MW, Stanley EH, Havlick DG, et al (2008) Aging Infrastructure and Ecosystem Restoration. Science 319:286–287.

Dubnick A, Barker J, Sharp MJ, et al (2010) Characterization of dissolved organic matter (DOM) from glacial environments using total fluorescence spectroscopy and parallel factor analysis. Annals of Glaciology 51:111–122. doi: 10.3189/172756411795931912

Gabor RS, Eilers K, McKnight DM, et al (2014) From the litter layer to the saprolite: Chemical changes in water-soluble soil organic matter and their correlation to microbial community composition. Soil Biology and Biochemistry 68:166–176. doi: 10.1016/j.soilbio.2013.09.029

Gallo EL, Lohse KA, Ferlin CM, et al (2014) Physical and biological controls on trace gas fluxes in semi-arid urban ephemeral waterways. Biogeochemistry 121:189–207. doi: 10.1007/s10533-013-9927-0

Gardner JR, Fisher TR, Jordan TE, Knee KL (2016) Balancing watershed nitrogen budgets: accounting for biogenic gases in streams. Biogeochemistry 1–23. doi: 10.1007/s10533-015-0177-1

Google Earth (Version 5.1.3533.1731) [Software]. Mountain View, CA: Google Inc. (2009). Available from https://www.google.com/earth/

Groffman PM, Gold AJ, Addy K (2000) Nitrous oxide production in riparian zones and its importance to national emission inventories. Chemosphere - Global Change Science 2:291–299.

Groffman PM, Pouyat R V. (2009) Methane uptake in urban forests and lawns. Environmental Science and Technology 43:5229–5235. doi: 10.1021/es803720h

Harrison J, Matson, Pamela A (2003) Patterns and controls of nitrous oxide emissions from waters draining a subtropical agricultural valley. Global Biogeochemical Cycles. doi: 10.1029/2002GB001991

Helton AM, Ardón M, Bernhardt ES (2015) Thermodynamic constraints on the utility of ecological stoichiometry for explaining global biogeochemical patterns. Ecology letters 18:1049–56. doi: 10.1111/ele.12487

Huguet A, Vacher L, Relexans S, et al (2009) Properties of fluorescent dissolved organic matter in the Gironde Estuary. Organic Geochemistry 40:706–719. doi: 10.1016/j.orggeochem.2009.03.002

Kaushal SS, Groffman PM, Band LE, et al (2011) Tracking nonpoint source nitrogen pollution in human-impacted watersheds. Environmental Science & Technology 45:8225–8232.

Kaushal SS, Belt KT (2012) The urban watershed continuum: evolving spatial and temporal dimensions. Urban Ecosystems 15:409–435. doi: 10.1007/s11252-012-0226-7

Outram FN, Hiscock KM (2012) Indirect nitrous oxide emissions from surface water bodies in a lowland arable catchment: a significant contribution to agricultural greenhouse gas budgets? Environmental Science & Technology 46:8156–8163.

Pabich WJ, Valiela I, Hemond HF (2001) Relationship between DOC concentration and vadose zone thickness and depth below water table in groundwater of Cape Cod, U.S.A. Biogeochemistry 55:247–268. doi: 10.1023/A:1011842918260

Pennino MJ, Kaushal SS, Beaulieu JJ, et al (2014) Effects of urban stream burial on nitrogen uptake and ecosystem metabolism: implications for watershed nitrogen and carbon fluxes. Biogeochemistry 121:247–269. doi: 10.1007/s10533-014-9958-1

Pennino MJ, Kaushal SS, Mayer PM, et al (2016) Stream restoration and sewers impact sources and fluxes of water, carbon, and nutrients in urban watersheds. Hydrology and Earth System Sciences 20:3419–3439. doi: 10.5194/hess-20-3419-2016

Raymond PA, Zappa CJ, Butman D, et al (2012) Scaling the gas transfer velocity and hydraulic geometry in streams and small rivers. Limnology & Oceanography: Fluids & Environments 2:41–53. doi: 10.1215/21573689-1597669

Richey JE, Devol AH, Wofsy SC, et al (1988) Biogenic Gases and the Oxidation and Reduction of Carbon in Amazon River and Floodplain Waters. Limnology and Oceanography 33:551–561.

Short MD, Daikeler A, Peters GM, et al (2014) Municipal gravity sewers: An unrecognised source of nitrous oxide. Science of the Total Environment 468–469:211–218. doi: 10.1016/j.scitotenv.2013.08.051

Strokal M, Kroeze C (2014) Nitrous oxide (N2O) emissions from human waste in 1970–2050. Current Opinion in Environmental Sustainability 9–10:108–121. doi: 10.1016/j.cosust.2014.09.008

Singh S, Dutta S, Inamdar S (2014) Land application of poultry manure and its influence on spectrofluorometric characteristics of dissolved organic matter. Agriculture, Ecosystems and Environment 193:25–36. doi: 10.1016/j.agee.2014.04.019

Singh S, Inamdar S, Mitchell M (2015) Changes in dissolved organic matter (DOM) amount and composition along nested headwater stream locations during baseflow and stormflow. Hydrological Processes 29:1505–1520. doi: 10.1002/hyp.10286

Townsend-Small A, Czimczik CI (2010) Carbon sequestration and greenhouse gas emissions in urban turf. Geophysical Research Letters 37:n/a-n/a. doi: 10.1029/2009GL041675

Battin TJ, Kaplan LA, Findlay SEG, et al (2008) Biophysical controls on organic carbon fluxes in fluvial networks. Nature Geoscience 1:95–100. doi: 10.1038/ngeo101

Bastviken D, Tranvik LJ, Downing JA, et al (2011) Freshwater Methane Emissions Offset the Continental Carbon Sink. Science Brevia 331:50.

Seitzinger SP, Kroeze C, Styles R V. (2000) Global distribution of N2O emissions from aquatic systems: Natural emissions and anthropogenic effects. Chemosphere - Global Change Science 2:267–279. doi: 10.1016/S1465-9972(00)00015-5

Van Drecht G, Bouwman AF, Harrison J, Knoop JM (2009) Global nitrogen and phosphate in urban wastewater for the period 1970 to 2050. Global Biogeochemical Cycles 23:1–19. doi: 10.1029/2009GB003458

Wilcock RJ, Sorrell BK (2008) Emissions of Greenhouse Gases CH4 and N2O from Low-gradient Streams in Agriculturally Developed Catchments. Water, Air, and Soil Pollution 188:155–170. doi: 10.1007/s11270-007-9532-8

---

## Author Response (AR1)

**BG 2016-380**
**Responses to Comments by Anonymous Reviewers 1 & 2**
**Resubmission of manuscript following open discussion & Associate Editor review**
**R. Smith et al.**
*March 10, 2017*

*We would like to thank the reviewers and the associate editor for their time in providing detailed, constructive comments regarding this manuscript. We have combined our responses to both reviewers' comments below, and believe that their contributions will lead to significant improvements. Both reviewers raised concerns about: 1) methodological details, 2) interpretation of results, and 3) terminology and clarity of ideas. As requested by the reviewers and the associate editor, we have responded to each of these comments below, and have incorporated all changes into the manuscript.*

**I.   METHODOLOGICAL INFORMATION**

*Both reviewers have expressed concern about the level of detail provided in the methods section, and certain specific methodologies used. We have compiled their general comments and replied to their examples where more than a simple textual response was deemed necessary.*

**R1 Comment 1: This paper appears to lack some methodological information, some of which is important and makes it difficult to assess what you did. Some of these examples of this are listed below.**

*And*

**R2 Comment 3) Some parts of the methods need clarification (e.g. supersaturation, DOM sample preservation). In addition some parts of the methods seem unnecessary given the results that are presented**

*In the submitted version of this paper, we described the different terminology for gas saturation in stream water (saturation ratio, or $xsCO_2$, $xsCH_4$, and $xsN_2O$) on page 7, lines 19-24. We have added the following text to clarify this section:*
*"Super-saturation is defined as having a saturation ratio >1 or when $xsCO_2$, $xsCH_4$, or $xsN_2O$ is >0." (Page 10, line 12)*

**R2, P6, L16: 0.7micron-filtered samples stored for 2 weeks seems inappropriate for a DOM composition analysis. 0.2 micron filtering is usually preferred.**

*In the original manuscript, we describe DOM sample preservation and analysis on page 6, lines 16-21. Following filtration through pre-combusted 0.7μM glass fiber filters, samples were stored in amber glass vials at 4°C and analyzed within 2 weeks following collection.*

*To the authors' knowledge, this is an appropriate and commonly utilized filtration procedure for DOM fluorescence metrics. Glass fiber filter pore sizes are not available below 0.7μM, and smaller filter materials (such as 0.2 μM nylon) have the potential to leach out fluorescently active compounds and/or measureable amounts of dissolved organic carbon during filtration.*

*We have added the following text to clarify this method:*
"Detailed methodology for optical properties and fluorescence indices can be found in Smith and Kaushal (2015), and numerous other studies have followed a similar filtration and storage procedure (Singh et al. 2014, Sing et al. 2015, Huguet et al 2009, Dubnick et al. 2010, Gabor et al. 2014). Fluorescently active DOM constitutes a wide range of lability. While some highly labile compounds may break down within hours of sample collection, more recalcitrant forms can remain stable for months. The 'two week window' is a convention meant to facilitate comparisons between sites, rather than a biologically based limit to storage *(Personal communication, Rachel Gabor, Shuiwang Duan).* "
(Page 9, lines 14-20).

**R1: P4 L22-24 and Table 1: You may want to explain why you decided to treat these watersheds as four categories of two replicates each, rather than eight watersheds varying continuously along a few axes (impervious surface cover, development age, etc.). I think the reason of different discrete stormwater infrastructure design types going with developments built at different times makes sense; you just might want to state it a little more explicitly.**

*The reviewer's understanding of our reasoning for treating watersheds as replicates of different categories is correct. We do attempt to explain the reasoning for development of infrastructure types (page 4, lines 22-24), and we have clarified this section as follows (Page 6, lines 16-21):*

"We identified four categories based on distinct combinations of stormwater and sanitary infrastructure dominating the greater Baltimore region, based on maps of stormwater control structures, housing age, and intensive field scouting. We then selected eight first-order streams paired across the four categories. The first order stream sites each were located in half in Red Run and half in Dead Run, sub-watersheds of the Gwynns Falls (Fig. 1). We have abbreviated the categories based on the dominant infrastructure feature as follows: 1) stream burial, 2) inline stormwater management (SWM) wetlands, 3) riparian/floodplain preservation, and 4) septic systems (Table 1)." (Page 6, lines 16-21)

*We have additionally reviewed the remainder of the text to ensure that the infrastructure groupings are not described as a gradient.*

**R1: P4 L26-28: Over what time period (i.e. year(s), season(s)/month(s)?, times of day?) Actually, you should probably give much of this this information earlier than this section, and I don't think you did.**

*We have added this information to the "Temporal Sampling of dissolved gases and stream chemistry" section* (Page 7, lines 7-10)*: "Headwater stream sites were sampled every two weeks for both water chemistry and dissolved gas concentrations. Chemistry sampling took place for two years, between January 2013 and December 2014. Dissolved gas sampling took place between July 2013 and July 2014. Sites were visited between the hours of 9 AM and 2 PM."*

**R1:  P5 L5: How did you define a study reach? Approximately how long were study reaches? This information should come up in the previous section.**

**R2, P5, L5: Unclear what is meant by "study reach". It has not been defined.**

"Five dissolved gas samples were collected per stream on each date, along an established 20m study reach either upstream adjacent to the gaging station. Gas samples were collected at 0, 5, 10, 15, and 20m from the fixed starting point of the study reach.*" (Page 7, lines 9-10)*

**R1: P5 L17  & P8 L21: "Estimated using Google Earth software" sounds a bit sketchy. If you must mention Google Earth, include a citation for the program. Ditto at 8(21)), and also, what's the precision on the Google DEM, and why didn't you use the lidar one mentioned in 5(19-20); is it not more precise?**

*This point was brought up by reviewers regarding 1) sampling locations, and 2) channel gradient (S) of our headwater stream sites, as well as the reaches where SF6 injection by Pennino et al. (2014) took place.*

*We did use Google Earth to identify the latitude and longitude of sampling locations, with reasonable confidence since sites were co-located with road crossings. We have cited Google Inc. here (Page 8, line 15).*

*In the original manuscript, we did estimate channel gradient of Pennino et al (2014)'s sites. In the revised version, we have re-calculated the channel gradient using a 1-meter resolution DEM derived from LiDAR surveys. This changed S for these sites, which in turn affected our estimate and uncertainty surrounding Cesc. Gas flux estimates have all increased in response to this change. Our description of these methods has changed as follows:*

"We estimated S of headwater streams with GHG sampling sites by measuring the change in elevation along the stream above and below stream gaging stations. We determined the latitude and longitude of the stream gage, which was co-located with GHG sampling sites in Red Run and Dead Run using a Trimble GeoXH handheld 3.5G edition GPS unit (10cm accuracy). We then plotted this location atop a 1-m resolution LiDAR-based digital elevation model (DEM, Baltimore County Government, 2002) in ArcMap 10. Using low points in the DEM to represent the stream channel, we then selected one point above and one point below the stream gaging station and measured the distance between these two points along the stream channel with the 'Measure' tool. We then calculated S based on the change in elevation divided by distance. The slope

measurement reach overlapped with, but did not coincide exactly with the gas sampling reach in order to ensure measureable differences in elevation. We followed the same protocol to estimate S for reaches in Pennino et al (2014), except rather than estimating points above and below a gaging station, we determined the change in elevation over the specific reach where $SF_6$ injections took place. Latitude and longitude for the upstream injection point and distance downstream were provided by Pennino et al (2014) provided data on the latitude and longitude of their $SF_6$ injection reaches. " (Page 12, line 23- Page 13 line 7)

**R2: P5 L17-20: There are multiple ways to make these calculations; what actual commands or tools did you use to do this?**

*We have clarified these methods as follows:*

" Sampling locations were designated pour points in the hydrology tools workflow. Because sampling points were always co-located with road crossings, we were able to acquire the latitude and longitude of sampling sites using Google Earth software (Google Inc. 2009). Watersheds were delineated using a 2-meter resolution DEM (Baltimore County Government, 2002). We first corrected the DEM for spurious depressions using the "Fill" tool in the ArcMap10.0 hydrology toolbox. Next, we calculated flow direction for each pixel of this filled DEM raster. We then used the Flow Accumulation tool to evaluate the number of pixels contributing to each downstream pixel. After ensuring that each pour point was co-located on the map streams (i.e. areas with flow accumulation >500 pixels), we used the 'Watershed' tool to delineate the pixels draining into each sampled location." (Page 8, lines 13-20)

**R1: P8 L14, 17, & 24) & P9 L1-4: What is $K_{20}$? You did not previously explain what GT (from $K_{GT}$) means in general terms, so if that explanation was supposed to translate; it does not do so effectively. Ditto with $K_{SF6}$ and plain K; are those at ambient temperature?**

*We have clarified our description reaeration coefficient (K) for a given gas (G) and temperature (T), as well as the gas transfer (k600) in the following lines, starting on page 11:*

We calculated the gas flux rate using Eq. (5) where $F_{GT}$ is the flux (g m$^{-2}$ d$^{-1}$) of a given gas (G) at ambient temperature (T) and d is water depth (m). $K_{GT}$ (day$^{-1}$) is the re-aeration coefficient for a given G at ambient T. Measured and equilibrium gas concentrations [$C_{str}$] and [$C_{eq}$] were calculated following equations 3 and 4, then converted to units of g m$^{-3}$.

$$F_{GT} = K_{GT} * d * ([C_{str}] - [C_{eq}]) ,$$  (5)

We modeled $K_{GT}$ for each site and sampling date using the energy dissipation model (Tsivoglou and Neal 1976). The energy dissipation model predicts K from the product of

water velocity (V, m day$^{-1}$), water surface gradient (S), and the escape coefficient, $C_{esc}$, (m$^{-1}$, Eq. 6).

$$K = C_{esc} * S * V \qquad (6)$$

$C_{esc}$ is a parameter related to additional factors other than streambed slope and velocity that affect gas exchange, such as streambed roughness and the relative abundance of pools and riffles. The $C_{esc}$ value used in this study was derived from 22 measurements of K, made using the SF$_6$ gas tracer method, carried out across a range of flow conditions in four streams within 5 km of our study sites and reported in Pennino et al. (2014). $C_{esc}$ was calculated as the slope of the regression of K vs. S*V from data in Pennino et al (2014) and was assumed to be representative of our headwater stream sites in Dead Run and Red Run.

We calculated $C_{esc}$ to be 0.653 m$^{-1}$ (n=22, r$^2$=0.42, p= 0.001). The 95% confidence interval of this $C_{esc}$ based on measured $K_{20,O2}$ values was ±0.359 m$^{-1}$, which corresponds to ±55% of a given gas flux estimate. This estimate of $C_{esc}$ from these nearby sites was assumed to be representative of the 8 stream reaches investigated in this study. Given the moderate range of uncertainty in $C_{esc}$, as well as additional uncertainties associated with slope estimation and relating $C_{esc}$ to different stream sites, gas flux estimates must be interpreted with caution.

Measurements of K were converted to K for each GHG (as well as O$_2$ for general comparisons) by multiplying by the ratio of their Schmidt numbers (Stumm and Morgan 1981). K measured at ambient temperature was converted to K at 20C ($K_{20}$) following Eq. 7.

$$K_{20} = \frac{KT}{11.0421^{T-20}} \qquad (7)$$

In order to compare re-aeration rates across sites and prior studies, we calculated the gas transfer velocity, $k_{600,}$ which is defined as $K_{20,O2}$ multiplied by water depth, with units of m d$^{-1}$. (Page 11 line 18- page 12 line 15)

**R1: P8 L20 You say you, "measure[ed] the change in elevation over a reach with a handheld GPS unit." Isn't elevation from GPS units usually rather unreliable? Describe the precision of your GPS unit.**

We have clarified our description of how the stream channel slopes at our gaging stations were determined as follows: (Page 12, line 18- Page 13 line 2):

"We estimated S at each GHG sampling site by measuring the change in elevation over a reach. We determined the latitude and longitude of the stream gaging using a Trimble GeoXH handheld 3.5G edition GPS unit (10cm accuracy). We then plotted this location atop a 2m resolution LiDAR-based digital elevation model (DEM, Baltimore County Government, 2002) in ArcMap 10. Using low points in the DEM to represent the stream channel, we then selected one point above and one point below the stream gaging station and measured the distance between these two points with the 'Measure' tool. We then calculated S based on the change in elevation divided by distance."

**R1: P3 L28: Go ahead and be more specific than "water chemistry" if you can do so concisely.**

*We have changed this to "were sampled every two weeks for dissolved carbon and nitrogen concentrations as well as and dissolved gases." (Page 7, lines 7- 8)*

**R2, P5, L1: Please specify what blanks are here.**
*We have clarified that we collected three gas blanks by filling vials with 25mL of helium in the field (Page 7, line 17)*

**R2, P5, L26: Not sure this equation and the associated text are necessary according to the results shown later.**
*Reviewer 2 is correct that we do not discuss the results from this mass balance calculation later on and it could justifiably be removed. We have removed this, as well as panels (e) and (f) from Figure 5.*

**R2, P5, L29: What about minor tributaries? Define better what you mean by major tributary.**
*We have clarified in the text that we sampled tributaries contributing more than 5% of the discharge to the main channel at a given point along the stream network, however minor tributaries, contributing less than 5%, were not measured. (Page 8, line 5)*

**R2, P6, L10-12: Specify how TDN and DOC were analyzed.**

*We have clarified that 'TDN' was measured using the 'TDN' method, which consists of high temperature combustion in the presence of a platinum catalyst, and clarify that the 'Shimadzu' instrument was a "TOC Analyzer." (Page 9, lines 7-10).*

**R2, P6, L29: Why use a new name for this index if BIX is the name normally used?**

*We have replaced 'index of autochthonous inputs' with 'BIX' throughout the manuscript.*

**R2 Comment 2) The role of external (non-in stream) and non-biological sources of GHG is not well considered in the manuscript. This may also make some calculations such as the index of aerobic and anaerobic respiration inaccurate.**

We have added clarification of the potential non-in-stream sources of GHGs in the introduction (Page 3 line 2) and discussion (Page 21, line 3) of the revised manuscript.

**R2, P7, L25 to P8, L11: This index seems controversial and needs clarifications. Not sure it can be really applied because apparently, it does not take into account external (non-in-stream) GHG sources and non-biological GHG sources.**

*Reviewer 2 is correct that AOU does not account for non-biological sources of GHGs. We will clarify this assumption about using the index on page 8, lines 10-11 where we define AOU. We have clarified that that AOU differentiates between aerobic $CO_2$ and all*

*other anaerobic or abiotic sources (and not anaerobic specifically). By using this index without an additional metric for abiotic $CO_2$, we must assume that the proportion of abiotic $CO_2$ is small and invariant across sites and dates sampled. Richey et al. (1988) justified the assumption that abiotic CO2 was minimal in their systems as follows "At ambient conditions (pH 6-7, alkalinity of 500-1000 ueq), with dissolved free $CO_2$ of 100-150uM or higher, the $CO_2$ produced through respiration remains primarily as dissolved $CO_2$. Thus ionic equilibrium reactions can be neglected." Richey et al (1998)'s justification is not valid in all cases for our study, as pH measurements varied widely from 4.81 to 8.9, and site-average $CO_2$ concentrations were lower than 100uM on 20 out of 152 sampling sites and dates, and alkalinity was not measured. $CO_2$ and pH were only both within this range on 36 out of 152 occasions. Among these observations, there remains a significant, positive linear relationship between xs $CO_2$ and xs$N_2O$ (p= 8.36 $x10^{-15}$, $r^2$ = 0.83) across all sites. We have thus changed the terminology to "xs$CO_2$-AOU" rather than anaerobic $CO_2$ throughout the text.*

*We have clarified in the text (Page 11, lines 4-16) that this index does account for external (non-in-stream) $CO_2$ and $O_2$ sources, and this was our main reason for using the index. Regardless of whether $CO_2$ and $O_2$ are produced within the stream, in the soil, or along groundwater flowpaths, the ratio of these two gases within the stream will represent the relative abundance of $CO_2$ production to $O_2$ consumption along that flowpath. Richey et al. (1988) and Daniels et al. (2002) are two examples of freshwater-based studies that used this index to evaluate anaerobic $CO_2$ production in freshwaters.*

**R2, P11, L21: The term "anaerobic $CO_2$ concentration" seems erroneous. It does not make much sense. The same applies for anaerobic $N_2O$ or $CH_4$ concentrations.**

*In the original draft of this manuscript, we defined 'anaerobic $CO_2$ as xs$CO_2$-AOU*1.2. We now refer to this metric as (xs$CO_2$-AOU) throughout the manuscript instead of anaerobic $CO_2$ in order to acknowledge potential abiotic sources of $CO_2$.*
*We would like to additionally clarify that AOU is not used for any other gases ($CH_4$ or $N_2O$) and we do not make mention to 'anaerobic $N_2O$' or 'anaerobic $CH_4$' because, unlike $CO_2$, these gases are not produced and consumed in direct proportion to $O_2$.*

**R2, P7, L23-25: Unclear. Please explain better how Cesc was estimated from SF6 additions.**
*We have clarified the definition of Cesc in the methods as follows: (Page 11, line 21-Page 12, line 6)*
"We modeled $K_{GT}$ for each site and sampling date using the energy dissipation model (Tsivoglou and Neal 1976). The energy dissipation model predicts K from the product of water velocity (V, m day$^{-1}$), water surface gradient (S), and the escape coefficient, $C_{esc}$, (m$^{-1}$, Eq. 6).

$$K = C_{esc} * S * V$$
$$(6)$$

$C_{esc}$ is a parameter related to additional factors other than streambed slope and velocity that affect gas exchange, such as streambed roughness and the relative abundance of

pools and riffles. The $C_{esc}$ value used in this study was derived from 22 measurements of K, made using the $SF_6$ gas tracer method, carried out across a range of flow conditions in four streams within 5 km of our study sites and reported in Pennino et al. (2014). $C_{esc}$ was calculated as the slope of the regression of K vs. S*V from data in Pennino et al (2014) and was assumed to be representative of our headwater stream sites in Dead Run and Red Run."

**R2: Table 1: I do not think so many decimals are necessary for most of these variables. Table 2: "0.000" = "<0.001" or "<0.0001"?**

*We have determined the significance limit for statistical tests in this study to be 0.008 using a bonferroni correction, in which the number of tests performed (6) is divided by the normal 95% significance limit (0.05) yielding 0.008. We have therefore changed the number of significant digits in all p-value reporting to match this limit.*

**R2, Table 4: If some variables were log-transformed (e.g. logDOC: NO3), this should be indicated in the methods section.**

*We have clarified that the log of DOC: NO3- was used for statistical comparisons (Page 14, line 16)*

**II.     STATISTICAL ANALYSES**

Comment 2: In your statistical methods (section 2.4, "Statistical Analyses,") you execute a number of models (linear mixed effects, stepwise linear regression, etc., yielding all the results in Table 2 and 5) testing similar or related things. This may constitute a statistical multiple comparisons problem, i.e. increased chance of Type I error (https://xkcd.com/882). Consider either combining models (e.g. in a structural equations modeling framework or similar) or correcting for this risk of error. At the very least, try to combine your categorical and continuous variables for into a single model for each gas.

*Reviewer 1 expressed concern about the statistical approach of using two modeling approaches to examine controls on each gas species citing that this approach seems redundant. The authors acknowledge that using two separate approaches for the purpose of predicting gas saturation values would increase the chance of Type I error; however, this was not the aim of our approach. The two models were used to examine first, whether or not there was consistent variation in gases across the categorical comparisons of watersheds, and secondly to examine whether or not gases could be predicted based on broader gradients in physical or chemical constituents that existed across all sampling dates and locations. In response to this reviewer's comments we have additionally incorporated a bonferroni correction to the p-value by dividing the 0.05 significance threshold by the number of models (6 models total, three for each gas), so that only tests with p< 0.0083 are considered significant. This does not change our results.*

**III.    INTERPRETATION OF RESULTS**

**R1 Comment 3: Some interpretations of your results, most but not all minor, don't entirely make sense, or seem incomplete. For example:**

**R1: 12(16-17): Are you sure the "influence" is actually "indirect" on "biogeochemical  processes in streams," or does the "indirect" part really only apply to GHGs? It  seems like those things listed are directly related to biogeochemistry in general.**

*We have clarified here that, while watershed infrastructure was not a statistically significant predictor of GHG saturation in streams, the gradients in DOC: $NO_3^-$ that we found across all infrastructure types was strongly correlated with GHG saturation. We interpreted this to mean that infrastructure may directly influence DOC and $NO_3^-$ - loading to streams, and that this C:N stoichiometry is likely to be an important controller of GHG abundance downstream. (Page 18, lines 10-16)*

*We have also added mention of GHGs produced along groundwater flowpaths, or entering streams directly from leaky sewer lines (Page 19, line 9)*

**R1: P12 L23: Plain "nitrogen" or *inorganic* nitrogen?"**

*We have changed the wording here to 'inorganic nitrogen' (P18, line 20)*

**R1: P13 L9-10: "stoichiometric conditions more favorable for denitrification" would be a  DOC: nitrate ratio closer to 1:1? If that ratio is different in incoming groundwater, wouldn't the $N_2O:CO_2$ ratio from that groundwater be correspondingly different as well?**

*We are not sure we follow the reviewer's question here, however we see the need here to clarify our interpretation of DOC:$NO_3^-$ and $CO_2:N_2O$ ratios.*

*DOC: $NO_3^-$ stoichiometry is one way to examine whether biogeochemical conditions are favorable for one microbial process over another, as Taylor and Townsend (2010) describe in their in-depth metadata analysis of DOC: $NO_3^-$ stoichiometry across a wide range of ecosystems. Helton et al. (2015) also provide a comprehensive review of the ways in which stoichiometry between inorganic N and organic C can be interpreted in various ecosystems. The implications of this stoichiometry at small spatial scales, such as the stream-groundwater interface of headwater streams, can be more complicated, however, and we agree with the reviewer that our interpretation could be explained more clearly.*

*As noted by Taylor and Townsend (2010), a DOC: $NO_3^-$ ratio of 1:1 is ideal for denitrification, while DOC: $NO_3^-$ much below 1:1 signifies conditions favorable for nitrification. While this ratio reflects the biogeochemical condition at the location/time the sample was collected, it is the result of processes occurring along the upstream*

*flowpath. In predominantly groundwater-fed streams, for instance, heterotrophic denitrification may consume significant proportion of DOC along groundwater flowpaths of a septic plume, thus drawing down the DOC: $NO_3^-$ of upwelling groundwater. Denitrification converts DOC to $CO_2$ and $NO_3^-$ to $N_2$ and $N_2O$. Numerous studies have shown septic plumes to have high concentrations of NO3- (e.g. Aravena et al. 1993). DOC concentrations are variable, but tend attenuate with depth in the aquifer and/or flow distance along the plume (Aravena and Robertson 1998; Pabich et al. 2001). For instance, Pabich et al. found high concentrations of DOC (>20 mg/L) in the upper part of a septic plume, with an exponential pattern of attenuation with depth. Consistently high $NO_3^-$- paired with attenuating DOC can result in a very low DOC: $NO_3^-$ ratio by the time groundwater reaches stream. These conditions at the stream-scale are is more favorable for nitrification. Since nitrification is a chemoautotrophic process, consuming $CO_2$ while producing $N_2O$, we would expect to see a negative relationship, or no relationship between $CO_2$ and $N_2O$ if nitrification were the dominant $N_2O$ production pathway in a given watershed. Instead, we find positive correlations between $CO_2$ and $N_2O$ in nearly all watershed sites (Figure 4a). We suggest therefore that denitrification may be producing $N_2O$ in the groundwater in our septic-dominated sites, and drawing down DOC: $NO_3^-$ along groundwater flowpaths. This interpretation remains hypothetical, however due to a number of biotic and abiotic processes occurring at the same time. Further work measuring solutes and gases along a groundwater flowpath is necessary to identify the mechanisms producing high concentrations of $N_2O$.*

**R1: P13 L24-25: You've made a big jump here, from relatively high emissions in certain places to "globally significant." Consider reminding your reader ("reminding" insofar as this should go in the introduction first; currently it's all just missing) what it would take for these locally high emissions to be globally significant- what's the relative global contribution of streams in general; how much of global streams is urban stream, etc. It might make more sense to think of the impacts of NO2 emissions in the city in terms of local air pollution than global GHGs. You might also think about if your findings suggest anything new for general biogeochemistry, as opposed to just the GHG emission application.**

*Rather than focusing on global emissions, have re-framed our results to point out here, and in other parts of the manuscript, that diffuse emissions from urban streams constitute a previously unaccounted for source of $N_2O$ and $CH_4$. It is currently unknown how significant this source is, although one study shows that, for $N_2O$, sanitary sewers could emit as much $N_2O$ per capita as current estimates for secondary WWTP plants (Short et al. 2014). There is evidence that most of the $N_2O$-N found in these streams originates as wastewater, and our study adds insight into the magnitude and variability of biogenic gases in streams draining septic and sewer infrastructure.*

*We have also emphasized the point that greenhouse gas emissions from urban streams may represent an important export pathway, for C and N from stream networks. Our results suggest that gaseous losses may need to be considered in urban watersheds from the perspective of mass transport and watershed C and N budgets, citing Gardner et al. (2015). (Page 21, line 5)*

*Since we did not measure $NO_2$ emissions, we are unable to comment on whether or not streams are a source of that gas in our study sites.*

**R1: Missing: How did you analyze "longitudinal variability," or the effect of "distance from watershed outlet," on any of the response variables, i.e., the output of the method described in section 2.1.3? You make claims about the results of this survey in section 3.6 and display graphs derived from the data in Fig. 5, and then about the significance of these findings in 14(10-18). However, it's never apparent that you did more than eyeball that data to assess spatial trends. Moreover, my eyeballing does not match your eyeballing; I don't see Fig. 5 as reflecting the patterns you describe in the text.**

**R1: P14 L10-18: See comment after "missing," in Comment 1; it is unclear if you did a statistical analysis to support these claims.**

*We have added statistical analysis of longitudinal surveys as follows: (Page 15, lines 2-8)*

"We analyzed longitudinal data using multiple linear regressions in order to evaluate whether patterns observed in headwater sites were representative of the broader stream network. We compiled data from four surveys – Red Run and Dead Run in spring and fall – and used a stepwise linear regression approach to determine the significant drivers for each gas (Table 6). Covariates included log of drainage area above each point, watershed (Red Run vs. Dead Run), season (spring vs. fall), DOC concentration, DIC concentration, TDN concentration, log of discharge, location (tributary vs. main stem), DOC: TDN molar ratio, a TDN by Drainage are interaction term, and a DOC by drainage are interaction term. We used the stepAIC() function in R to determine the optimal model formulation, selecting the model with minimum AIC."

**R1: P15 L17-18: "Variation in nonpoint sources and flowpaths" is not really an independent variable you tested; you don't know what in the watershed, but outside the stream, is driving anything, beyond a bit of inference about groundwater.**

*We have removed this sentence in the conclusion.*

**R1: Section 3.5 and Fig. 4b: Why do you think the slope directions of the lines in Fig. 4b so variable? Address this in discussion.**

We have added the following discussion of this result: *(Page 20, lines 3-6)*

*"Overall, the relationships between $CH_4$ and $CO_2$ were much weaker and more variable than the relationships between $CO_2$ and $N_2O$ (Figure 4). While $CO_2$ and $CH_4$ are sometimes correlated in wetlands and rivers with low oxygen (Richey et al. 1998), this was not the case for our study sites. Instead, $CO_2$ and $N_2O$ were highly coupled, suggesting prevalence of $NO_3^-$ as a terminal electron acceptor over $CO_2$."*

**R1: Table 5: You never interpret your K$_{20}$ results in the discussion.**

*We have added description and discussion of our modeled K$_{600}$ values in the results (Page 17, lines 15-22) as follows:*

"GHG emission rates were sensitive to differences in modeled k$_{600}$. Despite having medium to low gas saturation ratios compared with other sites, DRKV had the highest GHG emission rates on all dates. This is due in part to having the highest slope (0.10 m/m), and thus the highest modeled k$_{600}$ (m day$^{-1}$). Our 37 estimates of k$_{600}$ ranged from 2.4 to 122.6.1 m d$^{-1}$. Site-averages for k$_{600}$ varied from 5.39± 0.73 to 28.0± 7.0 m day$^{-1}$. The median value for all k$_{600}$ estimates was 13.24 m day$^{-1}$. This range of values and site-averaged values extends beyond that measured by Pennino et al. (2014) of 0.5 to 9.0 m d$^{-1}$. The discrepancy between Pennino et al. (2014)'s k$_{600}$ measurements is driven by differences in channel gradient. Gradients in the present study ranged from 0.01 to 0.1, while Pennino's ranged from 0.001 to 0.016 m d$^{-1}$. Channel gradient (S) is also the parameter with the greatest uncertainty, thus warranting cautious interpretation of our gas emission estimates.

*We have added discussion of the modeled k$_{600}$ values as they relate to gas fluxes as follows: (Page 20 line 24)*

"While our measured N$_2$O saturation ratios were highly correlated solute concentrations and redox conditions (Table 4), emission rates sensitive to the gas transfer velocity (k$_{600}$), which varied by two orders of magnitude in our study (Table 6). "

**R1: P12 L26: Can you not distinguish (or at least venture an educated guess) between "C and N inputs and/or microbial metabolism," based on measurements/calculations of these gases individually, together with those of other gases?**

We have added the following text regarding this point in the discussion: (Page 19, lines 3-9)

"We speculate that the location of infrastructure on the landscape may affect the relative importance of direct anthropogenic loading vs. microbial processes on DOC: NO$_3^-$ ratios of stream water. For instance we found high concentrations of and NO$_3^-$ and low DOC in streams draining septic systems. Much of this excess NO$_3^-$ is likely from septic plumes, but the lack of DOC may be the result of microbial C mineralization along subsurface flowpaths.  On the other end of the spectrum, very low NO$_3^-$ and TDN in streams draining watersheds in the floodplain preservation category, which were also newly developed. In this case, the higher C:N may have been driven by lower N leakage rates as

well as improved ecological function of the preserved floodplain wetlands to remove any N that does enter the groundwater from stormwater or sewage leaks."

**R2: P2, L4: Land use can alter GHG emissions from streams not only through changes in drivers of stream metabolism. Changes in external GHG sources (e.g. groundwater inputs, soil leaching, point sources) and some geochemical reactions may also be important. In general, only part of GHG emissions from streams come from in-stream metabolism. This relevant aspect is not made sufficiently clear in this manuscript.**

*We agree with Reviewer 2 that land use can alter external GHG sources to the stream, along with changing in-stream metabolism. In the present form of the paper, we make mention of external GHG sources on several occasions (i.e. 'Section 1.2. Role of Sanitary Infrastructure section), however we do not specifically attempt to differentiate between external vs. in-stream GHG production as we do not have data to back up this type of analysis.*

**IV.     DESCRIPTION OF STUDY DESIGN**

**R1 Comment 4: You refer several times to a gradient or continuum of stormwater infrastructure, but you never elucidate the relationships between or ordering of the infrastructure types that makes them constitute a gradient or continuum. Explain, up front and early. For example:**

**R1: P2 L29: Is the "along the urban watershed continuum" significant? Does something change along this gradient about the effect of the wetlands, or do you just mean "in urban watersheds?"**

*We have removed this mention of the 'urban watershed continuum' in this case, as the paragraph was re-written.*

**R1: P1 L16: It is not immediately clear how these seemingly discrete categories constitute  "a gradient of stormwater and sanitary infrastructure"- gradient along what axis, what variable?**

*We have clarified here, and throughout the text that the watersheds were compared as infrastructure categories rather than a continuum. For example, (Page 1, lines 15-20)*

"We hypothesized that urban infrastructure significantly alters downstream water quality and contributes to variability in GHG saturation and emissions.  We measured gas saturation and estimated emission rates in headwaters of two urban stream networks (Red Run and Dead Run) of the Baltimore Ecosystem Study Long-Term Ecological Research Project.. We identified four combinations of stormwater and sanitary infrastructure present in these watersheds, including: 1) stream burial, 2) inline stormwater wetlands, 3) riparian/ floodplain preservation, and 4) septic systems."

**R1: P3 L20-21: "Urban watershed continuum" again- is that just a way to refer to the stretch from the infrastructure in the headwater downstream a bit, or are the different kinds of infrastructure arranged along a continuum, or what?**

*The reviewer is correct that it is a way to refer to the flowpath from the infrastructure to the headwater downstream a bit. We have removed this instance of using the term in order to clarify meaning. We intend to describe the ways in which infrastructure is part of the stream network in urban watersheds – and how the infrastructure/stream interface may play a significant biogeochemical role at a watershed scale in urban ecosystems. We have focused our study on understanding the role of urban infrastructure on greenhouse gas dynamics in urban waterways. A growing body of work has shown that nutrient and carbon loads to streams, as well as the biogeochemical processes within flowing waters is related to not only to land cover (% impervious surface, urban density, etc) but also urban infrastructure. Connectivity between runoff-generating water sources (groundwater, overland flow, shallow subsurface flow) and urban infrastructure (sewer lines, stormwater conveyance pipes, drinking water pipes, constructed wetlands, etc.) is likely to influence not only the anthropogenic inputs of C and N to waterways but also the relative importance of biotic interactions on C and N removal along flowpaths. Kaushal and Belt (2012) describe a conceptual framework of how urban-impacted flowpaths may influence downstream export of nutrients as the 'Urban Watershed Continuum.'*

**R1: P5 L22: This is the closest thing to an explanation you've made so far, and it still doesn't really make sense.**

*We have removed this sentence and reference to the urban watershed continuum throughout the manuscript.*

**V.     VAGUE WORDING CHOICES**

**R1 Comment 5: You could improve this paper by reducing vague and occasionally careless diction. Sometimes this problem makes your meaning somewhat unclear. For example:**

*Both reviewers had concerns about some of the vague and unclear phrasing in sections of this paper. We respond here to their general comments as well as the specific examples from their line-by-line comments. Overall, we have clarified the key ideas underlying this paper in the introduction, provide more concrete details to back up statements about the literature, and link our interpretation of results more clearly to the figures and tables provided. Specific examples can be found in the following responses to reviewers' comments below.*

**R2: Title: I have the feeling that something is missing in the title. Maybe the word "of" before "urban"?**

*We changed title to 'Influence of infrastructure on water quality and greenhouse gas*

*dynamics in urban streams'*

**R1: P2 L3-4: Consider fleshing out "globally significant" with some actual numbers? Also, if you have space, it might not hurt to explain very briefly how this impact of rivers and streams on GHGs was determined. It is unclear here whether the figures you cite include urban streams or not, and why. In other words, could knowing about urban stream GHGs make these fluxes more or less "globally significant?" Without this piece of information, it is unclear if all of the potentially contributing factors to urban stream GHG emissions that you describe in the rest of the paragraph are already accounted for in the currently accepted stream GHG numbers and you're just partitioning sources, or if you might revise the numbers on stream GHG fluxes as a result of this study.**

*We have re-structured much of the introduction and included more details to justify the measurement of GHGs from streams(Page 2, lines 11-18) as follows:*

"Streams and rivers are dynamic networks that emit globally significant quantities of $CO_2$, $CH_4$ and $N_2O$ to the atmosphere. $CO_2$ emissions via flowing waters are equivalent to half of the annual terrestrial carbon sink (1.2 Pg $CO_2$-C yr$^{-1}$, Cole et al. 2007; Battin et al. 2009). Stanley et al. (2016) recently demonstrated that flowing waters are significant $CH_4$ sources as well, emitting approximately 28 Tg yr$^{-1}$, which is equivalent to between 10 and 35% of emissions from wetlands globally (Bridgham et al. 2013). Approximately 10% of global anthropogenic $N_2O$ emissions are emitted from river networks due to nitrogen contamination of surface and groundwater (UNEP 2013; Ciais et al. 2013). There is evidence that these $N_2O$ estimates, based on IPCC guidelines, might be too low, given growing evidence of high denitrification rates in small streams with high $NO_3^-$ loads (Beaulieu et al. 2011). "

*We have additionally clarified that some of these studies do take into account $N_2O$ emissions from urban areas indirectly, by using population to estimate N inputs to watersheds (Page 2, lines 21-26):*

*"As urban land cover and populations continue to expand, it is critical to understand the impacts on waterways, including C and N loading and GHG emissions. While N2O emissions from both urban and agricultural sources are taken into account in models based on estimated watershed DIN loading (Nevison et al. 2000; Seitzinger et al. 1998), measurements validating these estimates or estimates of CO$_2$ and CH$_4$ in urban watersheds are rare."*

**R2 P1, L17: Unclear what is meant by "watershed continuum". I think it would be more correct to speak about river network. This study focuses on the river and not on the whole watershed. This should be clear throughout the manuscript.**
**&**
**R1 P3 (20-21): "Urban watershed continuum" again- is that just a way to refer to the stretch from the infrastructure in the headwater downstream a bit, or are the**

**different kinds of infrastructure arranged along a continuum, or what?**

*These two comments are related to the term 'urban watershed continuum.' As noted above, we have removed references to the urban watershed continuum in this paper.*

**R1: P1(27-29): Your concluding sentence is rather vague; for a start, "influenced" could mean almost anything. Could you be a bit more specific about what the "influence" was and what the "implications" are?**

**R2: P1, L29: This last sentence of the abstract does not seem appropriate. It refers to emissions, which are not the focus of the manuscript. I would rather include a more conclusive sentence here.**

*These two comments refer to the final sentence of the abstract. We agree with the reviewers that our study does not focus on emissions and will remove the last part of this sentence starting with 'with significant implications…'*

*We have replaced this section with the following concluding sentence:*

*"Despite a decline in gas saturation from the headwaters, streams remained saturated with GHGs throughout the drainage network, however, suggesting that urban streams are continuous sources of CO2, CH4, and N2O." (Page 2, lines 3- 5)*

**R2: P3, L20-24: Yes, but how much do streams contribute to whole watershed GHG fluxes?**
**R1: To put your results in context a bit better, see Gallo et al. 2014 ("Physical and biological controls on trace gas fluxes in semi-arid urban ephemeral waterways" in Biogeochemistry 121(1) pp.189-207). They did related measurements in ephemeral streams in urbanized deserts, with similar results. For just nitrous oxide emissions from urban streams, there are several more relevant papers; try searching "nitrous oxide urban stream," in Web of Science if you can. (No, I am not Gallo et al.)**

*While it is beyond the scope of this manuscript to robustly quantify emissions from streams in this region we acknowledge that more context is necessary here to justify the scalability of our results. We have added mention of the relative contribution of rivers to GHG emissions from terrestrial ecosystems (agricultural and otherwise) in the introduction (Page 2, lines 11-18). We have also incorporated a citation to Gallo et al. (2014), as this study is highly relevant to the growing understanding of greenhouse gas production in urban aquatic environments. (Page, 2, line 21)*

**R2, P15, L25-28: I suggest the authors try to include more results-based conclusions and implications at the end of the paper. It also seems confusing that the authors emphasize wastewater here, when the paper is about streams and GIs.**

*We have changed the conclusions to reflect the extent of our paper- the relationship*

*between GHG saturation and water quality metrics, as well as the relatively high GHG emission rates which are equivalent to intensively managed agricultural landscapes (Page 22, lines 15-25)*

**R1: 1 L22: "These variables" refers to the "drivers of GHG dynamics," "infrastructure categories," or both? If it's the former, I guess this line just verifies that "nitrogen stoichiometry" etc. *are* in fact "drivers of GHG dynamics" in this context (as expected); if "these variables" are the "infrastructure categories," then it's a much more novel finding.**

*We have clarified which variables we are referring to in the text as follows:*

*"Multiple linear regressions including DOC: $NO_3^-$ and other variables (DO, TDN, and temperature) explained much of the statistical variation in nitrous oxide ($N_2O$, $r^2$= 0.78), carbon dioxide ($CO_2$, $r^2$=0.78), and methane ($CH_4$, $r^2$=0.50) saturation in stream water."* *(P1, line 23).*

**R2: P1, L23: Not sure these $r^2$ values are helpful here. It is not clear which statistical test was used.**

*We have clarified that the $r^2$ values refer to the results from multiple linear regression models for each gas. (P1, line 23-24).*

**R2: P1, L26: Again, unclear use of $r^2$ value.**
*We have removed the $r^2$ value in this line (P1, line 26).*

**R1: P2 L16-17: When you talk about GI here, are you proposing that all GI will have the same effects, at least in terms of direction of change in GHGs, or might effects differ depending on GI type?**

*We have clarified the potential varying roles of different forms of GI on GHGs as follows (Page 4, lines 14-17)*
*"The form of GI (i.e. stormwater control wetland vs. riparian/floodplain preservation) may also influence GHGs due to 1) differences in water residence time and oxygen depletion in wetland vs. floodplain soils, and 2) differences in watershed-scale N removal capacity of the two different approaches. Newcomer Johnson et al. (2014) found that riparian/ floodplain reconnection was more effective at reducing N export from streams, compared with stormwater wetlands in Baltimore.*

**R1: P2 L20: "Source of uncertainty" for what? Do you just mean "uncertain," or do you mean that this role could change our understanding of global fluxes from rivers, or what?**

*We have removed this sentence from the manuscript.*

**R1: P3 L10: Specify *anaerobic* nitrification; this is unclear until 12(29). With plain "nitrification," it at first seems like $N_2O$ must be a typo for $NO_2^-$. You also need a source here for the description of nitrification; I don't think Taylor and Townsend 2010 suffices.**

*We are not sure we understand the reviewer's comment here, as we do not use the term 'anaerobic nitrification' in this paper. As described on page 5, lines 9-10, nitrification is a chemoautotrophic process, which oxidizes $NH_4^+$ to $NO_3^-$. $CO_2$ is consumed during this process, and $N_2O$ is also produced as an intermediate in the $NO_3^-$ oxidation process. We have cited Taylor and Townsend (2010) because they provide an excellent framework for determining whether an environment is more favorable to nitrification over denitrification based on the ratio of $NO_3^-$ to DOC. We have additionally added a more general reference about nitrification in aquatic systems to the text (Schlesinger 1997).*

**R1: P3 L18: "GHG emissions"- what about them? "Increased GHG emissions?"**

*We have changed the wording here to "Increased GHG emissions"(Page 6, line 4)*

**R1: P12 L27 & P15 L13: Provide a citation for "'hot spots'" if you're going to put it in quotes, so we can verify which definition of "hot spot" you mean. Also, decide if you're going to say, "'hot spot'" or just "hotspot;" be consistent.**

*Upon reflection on Reviewer 1's suggestions about this analysis, we have removed the term(s) 'hot spot' in this section the paper and consistently separate it into two words where we do use the term.* In place of using the term 'hot spot', we have changed the wording as follows:

"Understanding the spatial variability in $N_2O$ concentrations, as well as the processes responsible for $N_2O$ production and $NO_3^-$ removal in watersheds is useful for informing watershed management." (Page 19, line 10).

**R1: P15 L23: "Role" or "influence?"  Sometimes your point could be stronger if you provided concrete numbers to back up your assertions. For example:  P2 L13: What does "substantially" mean? Can you provide numbers as to the relative contributions of nonpoint and point sources?**

*We have backed up the claim that 'Sewage may also contribute substantially to N2O emissions from urban streams' with the following text addition: (Page 3, lines 20-26)*

" Several studies have documented that wastewater leakage from municipal sewers often accounts for more than 50% of dissolved N in urban streams (Kaushal et al. 2011; Pennino et al. 2016; Divers et al. 2013). While sanitary sewer lines are known to leak dissolved N, $N_2O$ losses are not accounted for in greenhouse gas budgets of large WWTPs that these pipes feed into. Short et al. (2014) measured intake lines from three municipal WWTPs and estimated that $N_2O$ emissions from sewer lines alone on the same order of magnitude (1.7g $N_2O$ person $yr^{-1}$) as current IPCC estimates for per-capita

emissions from secondary WWTPs. Their study demonstrates the importance of constraining biogenic gas emissions from streams, which flow alongside and may receive gaseous inputs from aging sanitary sewer lines."

**R1: P3 L22-24: How is human population relevant? Also, please contextualize "fastest form of land use change;" that statement alone isn't really enough to ascertain significance. Is the magnitude of the change (i.e. first derivative of land use rather than second derivative) large? Is urban land use large, relative to other uses? Or do you think urban watersheds contribute disproportionately much to GHGs for their size, and so are significant globally even if small?**

*We will remove this sentence and clarify as follows: (Page 6, lines 10-12)*

"An improved understanding of the relationship between infrastructure type and biogeochemical functions is critical for minimizing unintended consequences of water quality management, especially as growing urban populations place greater burden on watershed infrastructure (Doyle et al. 2009; Foley et al. 2005; Strokal and Kroeze 2014)."

**R1: P12 L6: Which were the "three high-flow sampling dates?" Sometimes you waste valuable space by not going ahead and saying what you actually mean.**

We have changed the methods here slightly, removing the need to identify specific high flow sampling dates. (Page 13 lines 3-7)

"Pennino et al's (2014) measurements of V during gas injections ranged from 0.02 to 0.15 m s$^{-1}$. V measured at headwater gaging stations in our sites ranged from undetectable to 0.34 m s$^{-1}$. In order to avoid extrapolation, we limited our estimation of gas fluxes to sampling sites and dates with V in the range measured by Pennino et al. (2014). These conditions corresponded to 37 measurements total, spread unevenly across the four headwater sites with complete rating curves (DRAL, DRKV, RRRB, DRGG). K estimates were restricted to five dates at DRAL, 18 dates at DRKV, 11 dates at RRRB, and three dates at DRGG. "

**R1: P2 L9: Again, on "implications," try to be less vague if you can do so concisely. "Increase or decrease" or "change the magnitude of?" "Alter seasonality of?" Etc.**

During our re-structuring of the introduction, we have removed this sentence and clarified whether we expect an increase or decrease in GHG emission from streams in other instances.

**R1: P15 L1: By "typologies" you mean "types?"**

*In the methods section (page 4, lines 20-24), we originally described the combinations of sanitary and stormwater infrastructure in each pair of similar watersheds as 'typologies.' This has been changed to 'categories' throughout the manuscript.*

**VI. TRANSITIONS, DEFINING TERMS, ETC.**

**R1, Comment 6: Remember to maintain coherence and clarity of the paper through clear transitions, linking similar ideas, defining terms the first time you mention them, etc. For example:**

**R2 Comment 1) Some strange terms are used throughout the text that could be avoided (e.g. "watershed continuum", anaerobic concentration)**

*We have taken these reviewers comments into consideration and have made changes to the wording, definition of terms and transitions of ideas throughout the manuscript. We have included examples of our response to these concerns below.*

**R2: Abstract: You don't describe your "longitudinal" results here (the ones along stream length).**

*We have included the following description of our longitudinal results in the abstract:* (Page 2 lines 1-7)

"Longitudinal surveys extending form headwaters to third order outlets of Red Run and Dead Run took place in spring and fall. Linear regressions of this data yielded significant negative relationships between each gas with increasing watershed size, as well as consistent relationships between solutes (TDN or DOC, and DOC: TDN ratio) and gas saturation. Despite a decline in gas saturation between the headwaters and stream outlet, streams remained saturated with GHGs throughout the drainage network, suggesting that urban streams are continuous sources of $CO_2$, $CH_4$, and $N_2O$. "

*R1 P2 L21-23: How do these numbers/methods for calculating global fluxes that you cite here compare to the ones in 2(2-3)?*

*We have removed this sentence from the manuscript.*

**R1 P3 L29 - P4 L2: The final sentence in this paragraph seems out of place. Maybe shift it to the start of the next paragraph and end with, ", which facilitated site selection," or something? If you don't move the sentence, at least go ahead and explain why this information store matters. I mean, I can guess, but I shouldn't have to do so, or to wait until you bring it up again later. Maybe just collapse the first two paragraphs into one?**

*We have remove the last sentence of this paragraph.*

**R1 P4 L5-6: Clarify timing. Everything was put in place in the 1950s-1970s, and the aging and cracking is now (or rather, when this study was conducted)? Also, "between" or "from?"**

We have removed this paragraph in order to streamline our description of the study

design.

**R1 P4 L13: Remind us *which* eight streams- "...the eight streams *studied* drained...?"**

*We have clarified that we are referring to the headwater stream sampling sites, which are paired across eight infrastructure categories (Page 6, lines 15-21).*

**R1 P4 L14-20: Some of this description of what types of infrastructure were built when might go better in the introduction. Or at least, you might want to introduce the concept of change in design through time in the introduction.**

*We have rearranged this section and removed the description of stormwater and sanitary infrastructure designs to the introduction (Pages 2 -4)*

**R1 P4 L12-16: This sentence has a bit of a run-on feel; consider breaking down. Also, does "stormwater infrastructure... encompass older designs" *and* the newer GI ones? The way the sentence breaks doesn't suggest so. You could say, "We define stormwater infrastructure broadly to encompass older designs such as stormwater drainage networks and newer forms of 'green' stormwater infrastructure (GI)," and then define each in a sentence (or so) each.**

*We have removed this sentence in order to streamline the description of our study sites.*

**R1: P5 L20: Unclear how GIS calculations in previous sentence are used; abrupt transition back to "these surveys" is hard to follow.**

*We have clarified this in the text as follows: (Page 8 lines 12-20)*
 "We calculated the watershed contributing area above each sampling point and flow length from each sampling point to the watershed outlet using Hydrology toolbox in ArcMap 10. Sampling locations were designated pour points in the hydrology tools workflow. Because sampling points were always co-located with road crossings, we were able to acquire the latitude and longitude of sampling sites using Google Earth software (Google Inc. 2009). Watersheds were delineated using a 2-m resolution DEM (Baltimore County Government, 2002). We first corrected the DEM for spurious depressions using the "Fill" tool in the ArcMap10.0 hydrology toolbox. Next, we calculated flow direction for each pixel of this filled DEM raster. We then used the Flow Accumulation tool to evaluate the number of pixels contributing to each downstream pixel. After ensuring that each pout point was co-located on the map streams (i.e. areas with flow accumulation > 500 pixels), we used the 'Watershed' tool to delineate the pixels draining into each sampled location."

**R1 P5 L25: "Relative contributions of inflow" *to groundwater*?**

*Upon reflection and based on the reviewer's comments, we have decided to remove this hydrologic mass-balance analysis from the paper since it is not directly discussed or used*

*to interpret GHG results.*

**R1 P12 L30 - P13 L1: Consider referencing figures here (and more elsewhere in the discussion) to make it easy for readers to look back at the ratios etc. that you mention.**

*We have added figures to this section of the discussion as follows:*

*"*We found a strong positive relationship between $N_2O$ saturation and $CO_2$ concentrations, suggesting that denitrification was the primary source of $N_2O$ (Figure 5a). By contrast, very low DOC: $NO_3^-$ ratios (Figure 2) in stream water with highest $N_2O$ saturation (Figure 3a) suggest that nitrification was the dominant process at these sites." (Page 19, line 13-15)

**R1 P13 L27 -P14 L9 & P15 L5-9: Most of this information should go in the introduction. You can refer back to it here insofar as your findings update or add to it, but it's unclear that they do. It does not seem entirely relevant here.**

*We have removed the paragraph about drivers of methane and moved background information about nitrogen loading and $N_2O$ to the introduction. (Starting Page 4, line 18)*

**R1 P14 L31-32: You do not make it clear how this information about plants is relevant. Are you saying that some other type of plant within the waters you surveyed might be releasing methane in this way, but you didn't measure it? There are no transitions into or out of this part about the plants, either.**

We have removed this section on plants as it is not relevant to the paper.

**R1 P15 (26-27): It is unclear how exactly this part about wastewater relates to your results. Either make your transitions more clear, or move this sentence to a different section.**

*We have included more detail about the role of wastewater in our study as follows (Page 22, lines 19-25)*

*"*Our results suggest that N from septic plumes and sanitary sewer lines is the principal source of $N_2O$ saturation in our study sties. Dissolved inorganic N is highly correlated with $N_2O$ in our study sites, and the highest values are only present in watersheds with aging sanitary sewer infrastructure or septic systems. Our observations of $N_2O$ saturation and emissions from urban and suburban headwater streams are comparable with streams and ditches in intensive agricultural watersheds (Harrison and Matson. 2003; Outram et al. 2012). These results suggest that streams draining medium to low-density suburban or exurban land cover are comparable to those in intensively managed agricultural areas in terms of $N_2O$ emissions. *"*

**R1 P15(28): You have not brought up the concept of mitigation before, and it isn't immediately obvious if mitigation per se is the goal, or how your results translate to doing mitigation. Elaborate.**

*We have removed the mention of mitigation in the conclusions, as this is not the focus of this study.*

**R2 Comment 4) The dynamics of $CO_2$ are not considered in the discussion section**

*We have added discussion of $CO_2$ dynamics as they correspond with $N_2O$ and $CH_4$ respectively. (Starting page 18, line 18)*

**R2 Comment 5) Reference to relevant recent studies on GHG dynamics in urban streams are missing (e.g. see Alshboul et al. 2016 Environmental Science & Technology 50: 5555-5563 DOI: 10.1021/acs.est.5b04923 and references therein).**

*We have cited this paper in the introduction (Page 3, line 19)*

**Technical corrections: Again, numbers preceding comments refer to page number (line number). Please do not feel obligated to respond to all of these; just make sure you have them the way you want them in the final version.**

**1(30): "Infrastructure" misspelled. Also, consistent capitalization of keywords?**

*We have fixed this spelling error and capitalized all key words consistently (Page 2, line 8).*

**3(9-10): Instead of, "nitrification is a chemoautotrophic process that produces," you could just say, "nitrification chemoautotrophically produces," (and then switch ", and consumes" to "and consuming") for brevity.**

*We have changed the wording here as suggested by the reviewer (Page 5, line 14)*

**3(27), 5(20), & 7(22-23): Is just sticking a web link in here appropriate? For 5(20) and 7(22-23) especially, I think you need proper citations.**

We have included citations for these websites in the text and will move the websites to the references section. We have maintained the website reference to beslter.org, as this is a permanent link and not a specific data source. (Page 10, line 24)

**3(28): ", which" would be more grammatically appropriate than "that."**

*We have changed 'that' to 'which' in this sentence (now Page 9 Line 9)*

**4(7): "In-line?" Repeats throughout document- just make sure you want "in-line" and not "inline" or "in line."**

*We have changed all instances of 'in-line' to 'inline' in the document.*

**4(11-12): Maybe "and" instead of "that are;" the phrasing of this sentence is a bit awkward. Also, I think you could avoid the passive tense of "are located" ("exist?").**

In our broader re-writing of the discussion, we have removed this wording.

**4(26): "First-order streams" instead of "first order streams," yes?**

*We have changed 'first order' to 'first-order' here (Page 1, line 15 & page 6 line 19) and throughout the manuscript.*

**4(27-28): I'm not sure why you repeat all the categories when you just said them and even *said* that you just said them. Also, here you capitalized the categories and put apostrophes around them, whereas you didn't in the last sentence; pick a format, and be consistent.**

*We have re-written this section to clarify and consolidate descriptions of infrastructure categories (Page 6, lines 12-25)*

**4(32): "Septa" or "septum?"**

*We have changed the wording here to 'septum' (Page 7, line 15)*

**5(3), 7(16), 10(11), & 13(26): Remove tab for consistent paragraph formatting.**

We have removed tabs throughout the manuscript for consistent formatting.

**5(3-4): Consider rephrasing for clarity and brevity, e.g.: "A single stream water sample was collected in a 250 mL high-density polyethylene bottle at each site. One sample duplication rotated site each sampling date."**

We have rephrased this sentence as suggested by the reviewer (Page 7, line 20)

**5(10): Unnecessary "to."**

We have removed this 'to'

**5(15-16): Can shorten slightly by removing passive tense, i.e. "USGS provided discharge data." Also, consider providing a citation for the USGS data here.**

We have changed the wording of this sentence as follows:

"A minimum of 10 points was measured along each cross section. Discharge data was

provided by USGS when samples were co-located with a gaging station." (Page 8, line 10)

**6(9): "To *the* University?"**

We have added 'the' to this sentence (page 9, line 5)

**6(12): "Underestimates" or *underestimations*? Also, what "it" refers to is a bit unclear.**

We have changed 'under-estimates' to 'underestimation' and clarified that 'it refers to the NPOC method. (P9, line 8)

**6(13 & 24), 10(3), 11(23), & 13(7): "*Via*" and "*vs.*" need not be italicized.**

*We have removed italicization of these words throughout the manuscript.*

**6(16 & 19): Move "(DOM)" up to first use.**

We have defined dissolved organic matter (DOM) in the methods section (page 9, line 4) and refer to 'DOM' for the remainder of the manuscript.

**6(19-20 & 27-28): You essentially describe what molecular weight characterizes which source twice in a row, and do it better the second time; condense.**

We have removed the first mention of molecular weight and condensed the explanation as follows (Page 9, line 24- page 10 line 4)

 "The humification index (HIX) is defined as the ratio of emission intensity of the 435-480 nm region of the EEM to the emission intensity of the 300-345 nm region of the EEM at the excitation wavelength of 254 nm (Zsolnay et al. 1999; Ohno 2002). HIX varies from 0 to 1, with higher values signifying high-molecular weight DOM molecules characteristic of humic terrestrial sources. Lower HIX indicates DOM of bacterial or aquatic origin (Zsolnay et al. 1999). The autochthonous inputs index (BIX) is defined as the ratio of fluorescence intensity at the emission wavelength 380 nm to the intensity emitted at 430 nm at the excitation wavelength of 310 nm (Huguet et al. 2009). Lower BIX values (< 0.7) represent terrestrial sources, and higher BIX values (> 0.8) represent algal or bacterial sources (Huguet et al. 2009)."

**7(4): "Eq.'s?" Maybe just write it out.**

We have changed "Eq's (2-4)" to "equations 3 and 4 here. (Page 10, line 6)

**7(4): "Rations" or "ratios?" (Pretty sure you mean "ratios.")**

*We have changed 'rations' to 'ratios' here (page 10, line 6)*

**7(5): If you must put a comma before "(µmol L$^{-1}$), I think you need one after too.**

*We have removed the comma before* µmol L$^{-1}$ *(Page 9, line 25)*

**7(11) & Table 1: Combine things in parentheses in "(Eq. 3) (Stumm and Morgan 1981)." Similar change needed at end of caption for Table 1.**

We have done so here and in the caption of Table 1.

**7(19): "-" may be unnecessary.**

We have removed the hyphen here.

**8(4): "In" or "at?"**

We have changed 'in' to 'at here (page 11, line 9)

**8(7): "From" or "by?"**

We have changed 'from' to 'by' here. (Page 8, line 4)

**8(8): "Were," not "where."**

*We have changed 'where' to were here (page 12, line 14).*

**8(8-9): "Would be indicative of" can be shortened to "would indicate" or even "indicates." You could also remove, "other CO$_2$ sources, namely."**

*We have changed the wording here to 'indicates' (Page 10, line 1)*

**8(26): "P= " or "p=?"**

We changed 'p' to lower case here. (P 12, line 8)

**8(27): Provide units again for "±0.058."**

We have clarified that this value corresponds to 'Cesc, units of m-1 (P 12, line 8)

**9(13): Escaped ")."**

We have fixed the escaped ')'.

**9(19-20): Lost sentence fragment.**

We have removed this sentence fragment in our broader edits of this section.

**11(9): Second comma unnecessary. Also, why "may be," and only in second alternative explanation?**

We have removed the second comma and chanted 'may be' to 'is' (Page 18, line 8)

**12(16): "Typologies however," should probably be, "typologies, however."**

We have changed the comma usage in this sentence and changed 'typologies' to 'categories' throughout the text.

**12(22): You can shorten, "were present across all four infrastructure typologies (Fig. 4c), which suggests," to "present across all four infrastructure typologies (Fig. 4c) suggest."**

We have changed the wording here to reflect the reviewer's suggestion (P18, line 4)

**12(30): "Concentrations suggest that" should be, "concentrations, suggesting that."**

We have made this change to the text  (Page 18, line 20)

**13(24): "Warrants," not "warrant."**

we have changed 'warrant' to 'warrants' (Page 21, line 6)

**14(23): "With $DOC:NO3-$ while other" could use a comma in the middle (i.e. "with $DOC:NO^{3-}$, while other."**

We have removed this sentence during broader edits of the discussion section.

**15(1-2): Isn't there just the one negative relationship? ("The negative relationship" instead of "negative relationships.")**

We have changed the wording here to 'the negative relationship between $CH_4$ saturation and TDN, suggest…' (Page 22, line 11)

**Table 1: Header word spacing is awkward.**

We have consolidated the wording of headers for better alignment.

**Table 4: In caption, "* Indicate" should be something like, "A '*' indicates," based on  comparable sentences elsewhere.**

We have changed the wording of this figure caption as recommended.

**Table 5: You may be missing some commas towards the end of the list in the**

**caption.**

We have added commas to the caption as recommended.

**Figure 1: "Sampling sites and black dots signify" should have a comma after "sites."**

We have added a comma here as recommended.

**Figure 2: "Points signify data points," in the caption is a bit confusing; consider removing the second "points."**

We have removed the second 'points' as recommended in this figure caption.

**Figure 3c: I know it will mess with the clarity of your outliers, but consider some kind of log scale here; the differences between the actual boxes and whiskers are almost completely unapparent.**

We have changed the $CH_4$ panel on this figure (now Figure 3b) to have a log-scale.

**Figure 3: In caption, "box and whiskers signify the median, first and third quartiles," is unclear phrasing. At minimum, I think "box" needs to be plural.**

We have clarified the meaning of the box plots in this figure caption.

**Figure 5: Consider combining identical keys for panels (e) and (f), and perhaps some of the identical axes across panels as well. Unpunctuated letters representing figure panels within the caption text, e.g. "in panels a through d signify a saturation," are confusing; "a" is also a word. Also, more specific date here?**

We have removed the panels related to hydrologic mass-balance in this figure, as this analysis was not used to describe patterns in GHGs.

**15(16): "Of aquatic ecosystems" is in the middle of a list which relates to it (either end would make more sense), and the "as well as" and "significantly alter" seem unnecessary; commas would do.**

We have changed the wording here as follows: "Variations in urban infrastructure (i.e. SWM wetlands, riparian connectivity, septic systems) influenced C:N stoichiometry and redox state of urban streams. These in-stream variables, along with potential direct sources from leaky sanitary sewer lines may contribute to increased GHG production and/or delivery to streams. " (Page 22, lines17-19).

**15(25): "Include" not "includes."**

We have changed this to 'include' (Page 20, line 25)

**R2, P11, L1: This subtitle is repeated 3 times in this page.**
We have fixed the subtitles to match the text of each section.

**R2, P7, L13: Remove "and" before "flux"?**
We were not able to find this error in the submitted manuscript.

**R2, Table 4: If some variables were log-transformed (e.g. logDOC:NO3), this should be indicated in the methods section.**
We have added mention of the log DOC:NO3 ratio (Page 14, line 16)

**R2: For greater clarity, I suggest keeping the same order for the 3 solutes ($CO_2$, $CH_4$ and**
**$N_2O$) in all tables and figures as well as in the text**
**We have changed the order of gases throughout the figures and text.**
**&**
**2(25-26): For clarity, consider something like, "*In urban watersheds, these factors likely vary with* stormwater and sanitary sewer..."**

*Both comments refer to the same sentence. We have removed this sentence from the manuscript during our re-write of the introduction section.*

**3(5): Consider ending this sentence with an "as well," or similar to tie back to previous sentence.**

*We have added 'as well to the end of this sentence (Page 11, line 16).*

**2(24-25): Consider "Some key differences *between the watershed types that might affect this relationship* include," for clarity. Alternatively, "...may differ substantially *between* urban and agricultural watersheds *due to contrasting biogeochemistry and hydrology*. Some key differences..."**

We have removed this sentence from the manuscript during our re-write of the introduction.

**14(16): "Detailed information" is not in itself a "step;" you need a verb, e.g. "*Finding* detailed information."**

We have removed this sentence during the process of editing the discussion section.

**15(9-11): This sentence goes with the end of the last paragraph.**

We have removed this redundant sentence.

**14(5): By "relative proportion of different gases," do you actually mean "methane production?"**

We have changed the wording here as follows:
"As with $CO_2$). and $N_2O$, $CH_4$ saturation was negatively correlated with DO, however $CH_4$ was positively correlated with DOC: $NO_3^-$. $CO_2$ and $N_2O$, by contrast, were more strongly and positively correlated with TDN (Table 4). These patterns suggest that, along with redox conditions, carbon availability may modulate $CH_4$ production as well. (Page 21, lines 21-26).

**12(25): Instead of "Varying forms," just "form."**

We have moved this sentence to the introduction, and changed the wording to 'The form' rather than 'Varying forms' (Page 4, line 14)

**12(30): "The source," or just "the primary source," or "a source?"**

We have moved this sentence to the introduction (Page 3, line 20) and changed our wording here to be 'the primary source.'

**15(1): By "variations" you mean "differences?"**

We have changed 'variations' to 'differences' here. (Page 21, line 19)

**15(6): "Methodology" or "assumptions" (or "methodological assumptions")?**

We have removed this sentence during other edits of the discussion.

**15(20): "Ecological?" What does that mean here?**

We have removed the mention of GHG emissions from urban ecosystems in the concluding paragraph in order to focus more on results-based conclusions (i.e. connections between water quality and GHGs)

4(8): "Reflects" what? I think you mean the timing of development. Maybe rephrase: "...developed in the 2000s *with* more infiltration-based designs..."

We have re-written this section and removed the term 'reflects' from our description of this infrastructure category.

**4(18): Maybe "...exists in various forms*, including* gravity sewers and septic systems*, as well as a gradient...*" or "...exists *as both* gravity sewers and septic systems *along* a gradient..."rather than the current, more ambiguous, "...exists in varying forms (gravity sewers and septic systems) as well as a gradient...."**

We have removed this paragraph in our restructuring of this section.

**11(28): "Consistent along the drainage network for Red Run and Dead Run": do you mean looking intra-Red Run drainage network and intra-Dead Run drainage**

**network, or are you looking at both together as part of a larger drainage network? I think you mean the former, but your phrasing is unclear?**

We have edited this paragraph as follows (Page 21, line 15):

"Synoptic surveys of $N_2O$ saturation in Red Run and Dead Run in this study provide evidence that the entire network is a net source of $N_2O$ (Fig. 5). $N_2O$ saturation shows a significant decline with increasing drainage area (Table 6, Fig. 5), suggesting that emissions outpace new sources to the water column."

**References Cited**

Aravena R, ML E, JA C (1993) Stable Isotopes of Oxygen and Nitrogen in Source Identification of Nitrate from Septic Systems. Ground Water 31:180–186.

Battin TJ, Kaplan LA, Findlay SEG, et al (2008) Biophysical controls on organic carbon fluxes in fluvial networks. Nature Geoscience 1:95–100. doi: 10.1038/ngeo101

Bastviken D, Tranvik LJ, Downing JA, et al (2011) Freshwater Methane Emissions Offset the Continental Carbon Sink. Science Brevia 331:50.

Brady, P.A., Fath, B.D. (2008) Baltimore County Government Greenhouse Gas Inventory 2002-2006 Projections for 2012. Updated June 18, 2015, Accessed December 31, 2015. http://www.baltimorecountymd.gov/ Agencies/environment/sustainability/ghgproject.html

Cole JJ, Prairie YT, Caraco NF, et al (2007) Plumbing the Global Carbon Cycle: Integrating Inland Waters into the Terrestrial Carbon Budget. Ecosystems 10:172–185. doi: 10.1007/s10021-006-9013-8

Daniel MHB, Montebelo AA, Bernardes MC, et al (2001) Effects of urban sewage on dissolved oxygen, dissolved inorganic and organic carbon, and electrical conductivity of small streams along a gradient of urbanization in the Piracicaba River basin. Water, Air, and Soil Pollution 189–206.

Divers MT, Elliott EM, Bain DJ (2012) Constraining nitrogen inputs to urban streams from leaking sewer infrastructure using inverse modeling: Implications for DIN retention in urban environments. Environmental Science & Technology 47:1816–1823.

Doyle MW, Stanley EH, Havlick DG, et al (2008) Aging Infrastructure and Ecosystem Restoration. Science 319:286–287.

Dubnick A, Barker J, Sharp MJ, et al. Characterization of dissolved organic matter (DOM) from glacial environments using total fluorescence spectroscopy and parallel factor analysis. Annals of Glaciology 51:111–122. doi: 10.3189/172756411795931912. 2010.

Gabor RS, Eilers K, McKnight DM, et al (2014) From the litter layer to the saprolite: Chemical changes in water-soluble soil organic matter and their correlation to microbial community composition. Soil Biology and Biochemistry 68:166–176. doi: 10.1016/j.soilbio.2013.09.029

Gallo EL, Lohse KA, Ferlin CM, et al (2014) Physical and biological controls on trace gas fluxes in semi-arid urban ephemeral waterways. Biogeochemistry 121:189–207. doi: 10.1007/s10533-013-9927-0

Gardner JR, Fisher TR, Jordan TE, Knee KL (2016) Balancing watershed nitrogen budgets: accounting for biogenic gases in streams. Biogeochemistry 1–23. doi: 10.1007/s10533-015-0177-1

Google Earth (Version 5.1.3533.1731) [Software]. Mountain View, CA: Google Inc. (2009). Available from https://www.google.com/earth/

Groffman PM, Gold AJ, Addy K (2000) Nitrous oxide production in riparian zones and its importance to national emission inventories. Chemosphere - Global Change Science 2:291–299.

Groffman PM, Pouyat R V. (2009) Methane uptake in urban forests and lawns. Environmental Science and Technology 43:5229–5235. doi: 10.1021/es803720h

Harrison J, Matson, Pamela A (2003) Patterns and controls of nitrous oxide emissions from waters draining a subtropical agricultural valley. Global Biogeochemical Cycles. doi: 10.1029/2002GB001991

Helton AM, Ardón M, Bernhardt ES (2015) Thermodynamic constraints on the utility of ecological stoichiometry for explaining global biogeochemical patterns. Ecology letters 18:1049–56. doi: 10.1111/ele.12487

Huguet A, Vacher L, Relexans S, et al (2009) Properties of fluorescent dissolved organic matter in the Gironde Estuary. Organic Geochemistry 40:706–719. doi: 10.1016/j.orggeochem.2009.03.002

Kaushal SS, Groffman PM, Band LE, et al (2011) Tracking nonpoint source nitrogen pollution in human-impacted watersheds. Environmental Science & Technology 45:8225–8232.

Kaushal SS, Belt KT (2012) The urban watershed continuum: evolving spatial and temporal dimensions. Urban Ecosystems 15:409–435. doi: 10.1007/s11252-012-0226-7

Outram FN, Hiscock KM (2012) Indirect nitrous oxide emissions from surface water bodies in a lowland arable catchment: a significant contribution to agricultural greenhouse gas budgets? Environmental Science & Technology 46:8156–8163.

Pabich WJ, Valiela I, Hemond HF (2001) Relationship between DOC concentration and vadose zone thickness and depth below water table in groundwater of Cape Cod, U.S.A. Biogeochemistry 55:247–268. doi: 10.1023/A:1011842918260

Pennino MJ, Kaushal SS, Beaulieu JJ, et al (2014) Effects of urban stream burial on nitrogen uptake and ecosystem metabolism: implications for watershed nitrogen and carbon fluxes. Biogeochemistry 121:247–269. doi: 10.1007/s10533-014-9958-1

Pennino MJ, Kaushal SS, Mayer PM, et al (2016) Stream restoration and sewers impact sources and fluxes of water, carbon, and nutrients in urban watersheds. Hydrology and Earth System Sciences 20:3419–3439. doi: 10.5194/hess-20-3419-2016

Raymond PA, Zappa CJ, Butman D, et al (2012) Scaling the gas transfer velocity and hydraulic geometry in streams and small rivers. Limnology & Oceanography: Fluids & Environments 2:41–53. doi: 10.1215/21573689-1597669

Richey JE, Devol AH, Wofsy SC, et al (1988) Biogenic Gases and the Oxidation and Reduction of Carbon in Amazon River and Floodplain Waters. Limnology and Oceanography 33:551–561.

Seitzinger SP, Kroeze C, Styles R V. (2000) Global distribution of N2O emissions from aquatic systems: Natural emissions and anthropogenic effects. Chemosphere - Global Change Science 2:267–279. doi: 10.1016/S1465-9972(00)00015-5

Short MD, Daikeler A, Peters GM, et al (2014) Municipal gravity sewers: An unrecognised source of nitrous oxide. Science of the Total Environment 468–469:211–218. doi: 10.1016/j.scitotenv.2013.08.051

Strokal M, Kroeze C (2014) Nitrous oxide (N2O) emissions from human waste in 1970–2050. Current Opinion in Environmental Sustainability 9–10:108–121. doi: 10.1016/j.cosust.2014.09.008

Singh S, Dutta S, Inamdar S. Land application of poultry manure and its influence on spectrofluorometric characteristics of dissolved organic matter. Agriculture, Ecosystems and Environment 193:25–36. doi: 10.1016/j.agee.2014.04.019, 2014

Singh S, Inamdar S, Mitchell M. Changes in dissolved organic matter (DOM) amount and composition along nested headwater stream locations during baseflow and stormflow. Hydrological Processes 29:1505–1520. doi: 10.1002/hyp.10286, 2015

Townsend-Small A, Czimczik CI (2010) Carbon sequestration and greenhouse gas emissions in urban turf. Geophysical Research Letters 37:n/a-n/a. doi: 10.1029/2009GL041675

Van Drecht G, Bouwman AF, Harrison J, Knoop JM (2009) Global nitrogen and phosphate in urban wastewater for the period 1970 to 2050. Global Biogeochemical Cycles 23:1–19. doi: 10.1029/2009GB003458

[revised manuscript text omitted]

**2.4 Statistical Analyses**

**2.4.1 Role of infrastructure and seasonality**

**Rose Smith 3/9/2017 4:39 PM**

**Rose Smith 3/9/2017 4:39 PM**

**Beaulieu, Jake 3/5/2017 10:23 PM**
**Moved up [2]:** The energy dissipation model uses the relationship between measurements of $K_{20,O2}$ from SF$_6$ gas injections and the product of water velocity (V, m day$^{-1}$) and water surface gradient (S) to calculate the escape coefficient, $C_{esc}$, (m$^{-1}$, Eq. 7). $C_{esc} = K_{20,O2} / (S * V)$ . . . . . . . . . . . . .(7)$C_{esc}$ is a parameter related to additional factors other than streambed slope and velocity that affect gas exchange, such as streambed roughness and the relative abundance of pools and riffles. Pennino et al. (2014) measured K during 22 SF$_6$ gas tracer injection experiments carried out across a range of flow conditions in four streams within 5 km$^2$ of our study sites. We used these measurements to estimate $C_{esc}$, assuming their measurements to be representative of our headwater stream sites in Dead Run and Red Run.

**Rose Smith 3/9/2017 3:42 PM**

**Rose Smith 2/23/2017 7:42 PM**

**Rose Smith 2/22/2017 5:05 PM**

**Rose Smith 2/22/2017 5:12 PM**

**Rose Smith 3/10/2017 1:49 PM**

**Rose Smith 2/22/2017 5:12 PM**

**Rose Smith 2/22/2017 5:12 PM**

A linear mixed effects modeling approach was used to determine the significant drivers of each gas across streams in different headwater infrastructure categories. Due to uncertainties in the gas flux parameters, GHG saturation ratios were used rather than GHG emissions to compare spatial and temporal patterns across sites. Mixed effects modeling was carried out using R (R Core Team, 2014) and the *nlme* package (Pinheiro et al. 2012) following guidance outlined in Zurr et al. (2009). Separate mixed

5    effects models were used to detect the role of infrastructure category and date on each response variable. Response variables included saturation ratios for each gas ($CO_2$, N2O, and CH4), solute concentrations (DOC, DIC, TDN, $NO_3^-$), and organic matter source indices (HIX, BIX). Fixed effects were 'infrastructure category' and 'sampling date,' as well as an interaction term for the two.  The effect of a random intercept for 'site' was included in each model. The statistical assumptions of normality, and equal variances were validated by inspecting model residuals. When necessary, variances were weighted based on infrastructure

10    category to remove heteroscedasticity in model residuals (Zuur et al. 2009). The assumption of temporal independence was examined by testing for temporal autocorrelation in each response variable. This test was performed using the function 'corAR1(),' which is part of the package 'nlme' in R. The significance of random effects,  weighting variances, and  temporal autocorrelation was tested by comparing Akaike information criterion (AIC) scores for models with and without each of these attributes. Additionally, pairwise ANOVA tests were run to determine whether each additional level of model complexity

15    significantly reduced the residual sum of squares. Final model selection was based on meeting model assumptions, minimizing the AIC value, and minimizing residual standard error. Pairwise comparisons among infrastructure categories were examined using the Tukey HSD post-hoc test (*lsmeans* package, Lenth, 2016) for each response variable where 'infrastructure category' had a significant effect.  Where 'infrastructure category' did not have a significant effect on a response variable after incorporating 'site' as a random effect, a separate set of linear models was run with 'site' and 'date' as main effects rather than

20    'infrastructure category'. The role of 'site' was evaluated in these cases to determine the degree to which site-specific factors overwhelmed the effect of infrastructure category.

**2.4. Role of environmental variables on gas saturation**

A stepwise linear regression approach was used to examine the role of multiple environmental variables on $CO_2$, N2O, and $CH_4$ saturation across sites and dates. Predictor variables were selected via backward stepwise procedure, using the 'Step' function in

25    R. This involves first running a model that includes all potential driving factors, then running sequential iterations of that model after removing one variable at a time until the simplest and most robust combination of predictors was achieved. Model fit at

each step was evaluated using the AIC score. Parameters that did not reduce AIC when comparing models were removed until the model had the best fit with the minimum number of factors. The initial list of potential drivers included temperature, DO, DOC, TDN, DIC, HIX, and the BIX. Prior to the stepwise regression, we calculated the variance inflation factor (VIF) for each response variable to test for multicolinearity. VIF > 3 was the cut off for assessing multicolinearity. All variables in this study were below the VIF > 3 threshold (Zuur et al. 2010).

Analysis of covariance (ANCOVA) was carried out to determine whether relationships among gases ($CO_2$ vs. $N_2O$, $CO_2$ vs. $CH_4$) and solutes (log of DOC:$NO_3^-$ ratio) varied systematically across infrastructure categories. ANCOVA involved comparing two generalized least squares models. The first linear model included an interaction term between one of the predictor variables (i.e. DOC or $CO_2$) and infrastructure category to predict the response variable ($N_2O$ or $CH_4$). The second was a linear model with the same two independent variables but no interaction term. When infrastructure category had a significant influence on both the intercept (first model) and slope (second model) of a relationship, this refuted the null hypothesis that infrastructure category had no influence on a relationship.

[revised manuscript text omitted]

Rose Smith 2/23/2017 8:43 PM
Rose Smith 2/23/2017 8:43 PM
Rose Smith 2/23/2017 8:43 PM
Rose Smith 2/22/2017 10:31 PM
Rose Smith 2/22/2017 10:31 PM
Rose Smith 2/28/2017 2:49 PM
Formatted Table ... [41]
Rose Smith 2/22/2017 10:31 PM
Rose Smith 2/22/2017 10:31 PM
Rose Smith 2/22/2017 10:31 PM
Rose Smith 2/22/2017 10:32 PM
Formatted ... [42]
Rose Smith 2/22/2017 10:32 PM
Formatted ... [43]
Rose Smith 2/22/2017 10:32 PM
Rose Smith 2/16/2017 8:31 PM
Rose Smith 2/16/2017 8:31 PM
Rose Smith 2/22/2017 10:32 PM
Rose Smith 2/22/2017 10:32 PM
Formatted ... [44]
Rose Smith 2/22/2017 10:32 PM
Rose Smith 2/22/2017 10:32 PM
Rose Smith 2/22/2017 10:32 PM

Table 2 Summary of results (main effects p-values) from mixed effects models examining the role of infrastructure typology and date on the following response variables: $CO_2$, $N_2O$ and $CH_4$ saturation ratios; TDN and DOC concentrations (mg L$^{-1}$), BIX, and HIX (unitless).

[revised manuscript text omitted]

Emissions during three high-flow sampling dates (over 0.015 m3 $s^{-1}$ for all sites) increased the variance of overall mean gas emission rates estimates.

| Page 17: [29] Deleted | Rose Smith | 2/27/17 3:29 PM |

When these high emission rates were removed, average daily $CO_2$ emissions ($\pm$ standard error) was twenty to 100-fold higher at DRKV 39.5 ($\pm$15.5) g C $m^{-2}$ $day^{-1}$ than the other sites, due in part to the tenfold high stream surface slope at DRKV.

| Page 19: [30] Deleted | Rose Smith | 3/10/17 3:20 PM |

s

| Page 19: [30] Deleted | Rose Smith | 3/10/17 3:20 PM |

s

| Page 19: [30] Deleted | Rose Smith | 3/10/17 3:20 PM |

s

| Page 19: [30] Deleted | Rose Smith | 3/10/17 3:20 PM |

s

| Page 19: [30] Deleted | Rose Smith | 3/10/17 3:20 PM |
|---|---|---|

s

| Page 19: [30] Deleted | Rose Smith | 3/10/17 3:20 PM |
|---|---|---|

s

| Page 19: [30] Deleted | Rose Smith | 3/10/17 3:20 PM |
|---|---|---|

s

| Page 19: [30] Deleted | Rose Smith | 3/10/17 3:20 PM |
|---|---|---|

s

| Page 19: [30] Deleted | Rose Smith | 3/10/17 3:20 PM |
|---|---|---|

s

| Page 19: [30] Deleted | Rose Smith | 3/10/17 3:20 PM |
|---|---|---|

s

| Page 19: [30] Deleted | Rose Smith | 3/10/17 3:20 PM |
|---|---|---|

s

| Page 19: [30] Deleted | Rose Smith | 3/10/17 3:20 PM |
|---|---|---|

s

| Page 19: [30] Deleted | Rose Smith | 3/10/17 3:20 PM |
|---|---|---|

s

| Page 19: [30] Deleted | Rose Smith | 3/10/17 3:20 PM |
|---|---|---|

s

| Page 19: [30] Deleted | Rose Smith | 3/10/17 3:20 PM |
|---|---|---|

s

| Page 19: [30] Deleted | Rose Smith | 3/10/17 3:20 PM |
|---|---|---|

s

| Page 19: [30] Deleted | Rose Smith | 3/10/17 3:20 PM |
|---|---|---|

s

| Page 19: [30] Deleted | Rose Smith | 3/10/17 3:20 PM |
|---|---|---|

s

| Page 19: [31] Deleted | Rose Smith | 3/2/17 12:41 AM |
|---|---|---|

| | | |
|---|---|---|
| **Page 19: [31] Deleted** | **Rose Smith** | **3/2/17 12:41 AM** |

| | | |
|---|---|---|
| **Page 20: [32] Deleted** | **Rose Smith** | **3/10/17 1:17 PM** |

runoff,

| | | |
|---|---|---|
| **Page 20: [33] Deleted** | **Rose Smith** | **3/1/17 11:51 AM** |

rates

| Page 20: [33] Deleted | Rose Smith | 3/1/17 11:51 AM |

rates

| Page 20: [33] Deleted | Rose Smith | 3/1/17 11:51 AM |

rates

| Page 20: [33] Deleted | Rose Smith | 3/1/17 11:51 AM |

rates

| Page 20: [33] Deleted | Rose Smith | 3/1/17 11:51 AM |

rates

| Page 20: [33] Deleted | Rose Smith | 3/1/17 11:51 AM |

rates

| Page 20: [34] Deleted | Rose Smith | 3/1/17 11:39 AM |

$N_2O$ emissions from agricultural runoff are currently included in IPCC estimates, but emissions associated with urban ecosystems are not currently accounted for (Ciais et al. 2013). Urban and agricultural streams are similar in that they recei ve excess nitrogen inputs from the watershed, including N inputs from contaminated groundwater. Key differences arise when considering $N_2O$ budgets, however. Whereas agricultural stream emissions are estimated based on annual fertilizer inputs, N in urban streams is derived from diffuse, spatially heterogeneous nonpoint sources. For instance, studies in Baltimore have found that atmospheric deposition and human waste contribute approximately 25 % and 50 % of nitrate inputs, while the remainder is derived from soils and plant materials (Kaushal et al. 2011; Pennino et al. 2016). The proportion of these sources and others is likely to vary widely across and within watersheds.

Recent reviews have suggested that

| Page 20: [34] Deleted | Rose Smith | 3/1/17 11:39 AM |

| Page 20: [34] Deleted | Rose Smith | 3/1/17 11:39 AM |

are similar in that they recei ve excess nitrogen inputs from the watershed, including N inputs from contaminated groundwater. Key differences arise when considering $N_2O$ budgets, however. Whereas agricultural stream emissions are estimated based on annual fertilizer inputs, N in urban streams is derived from diffuse, spatially heterogeneous nonpoint sources. For instance, studies in Baltimore have found that atmospheric deposition and human waste contribute approximately 25 % and 50 % of nitrate inputs, while the remainder is derived from soils and plant materials (Kaushal et al. 2011; Pennino et al. 2016). The proportion of these sources and others is likely to vary widely across and within watersheds.

Recent reviews have suggested that

atmospheric deposition and human waste contribute approximately 25 % and 50 % of nitrate inputs, while the remainder is derived from soils and plant materials (Kaushal et al. 2011; Pennino et al. 2016). The proportion of these sources and others is likely to vary widely across and within watersheds.

Recent reviews have suggested that

| Page 21: [35] Deleted | Rose Smith | 3/1/17 12:19 PM |
|---|---|---|

, despite patterns of 'hot spots' in the headwaters

| Page 21: [36] Deleted | Rose Smith | 2/21/17 6:51 PM |
|---|---|---|

| | | |
|---|---|---|
| **Page 21: [36] Deleted** | **Rose Smith** | **2/21/17 6:51 PM** |

| | | |
|---|---|---|
| **Page 21: [36] Deleted** | **Rose Smith** | **2/21/17 6:51 PM** |

| | | |
|---|---|---|
| **Page 21: [36] Deleted** | **Rose Smith** | **2/21/17 6:51 PM** |

| | | |
|---|---|---|
| **Page 21: [36] Deleted** | **Rose Smith** | **2/21/17 6:51 PM** |

| | | |
|---|---|---|
| **Page 21: [36] Deleted** | **Rose Smith** | **2/21/17 6:51 PM** |

| | | |
|---|---|---|
| **Page 21: [36] Deleted** | **Rose Smith** | **2/21/17 6:51 PM** |

| | | |
|---|---|---|
| **Page 21: [37] Deleted** | **Rose Smith** | **3/1/17 12:31 PM** |

Measurements of $CH_4$

| | | |
|---|---|---|
| **Page 21: [37] Deleted** | **Rose Smith** | **3/1/17 12:31 PM** |

Measurements of $CH_4$

| | | |
|---|---|---|
| **Page 21: [37] Deleted** | **Rose Smith** | **3/1/17 12:31 PM** |

Measurements of $CH_4$

| | | |
|---|---|---|
| **Page 21: [37] Deleted** | **Rose Smith** | **3/1/17 12:31 PM** |

Measurements of $CH_4$

| | | |
|---|---|---|
| **Page 21: [37] Deleted** | **Rose Smith** | **3/1/17 12:31 PM** |

Measurements of $CH_4$

| | | |
|---|---|---|
| **Page 21: [37] Deleted** | **Rose Smith** | **3/1/17 12:31 PM** |

Measurements of $CH_4$

| | | |
|---|---|---|
| **Page 21: [37] Deleted** | **Rose Smith** | **3/1/17 12:31 PM** |

Measurements of $CH_4$

| | | |
|---|---|---|
| **Page 21: [37] Deleted** | **Rose Smith** | **3/1/17 12:31 PM** |

Measurements of $CH_4$

| | | |
|---|---|---|
| **Page 21: [37] Deleted** | **Rose Smith** | **3/1/17 12:31 PM** |

Measurements of $CH_4$

| | | |
|---|---|---|
| **Page 21: [37] Deleted** | **Rose Smith** | **3/1/17 12:31 PM** |

Measurements of $CH_4$

| **Page 21: [37] Deleted** | **Rose Smith** | **3/1/17 12:31 PM** |

Measurements of $CH_4$

| **Page 21: [37] Deleted** | **Rose Smith** | **3/1/17 12:31 PM** |

Measurements of $CH_4$

| **Page 21: [37] Deleted** | **Rose Smith** | **3/1/17 12:31 PM** |

Measurements of $CH_4$

| **Page 21: [37] Deleted** | **Rose Smith** | **3/1/17 12:31 PM** |

Measurements of $CH_4$

| **Page 21: [37] Deleted** | **Rose Smith** | **3/1/17 12:31 PM** |

Measurements of $CH_4$

| **Page 21: [37] Deleted** | **Rose Smith** | **3/1/17 12:31 PM** |

Measurements of $CH_4$

| **Page 21: [37] Deleted** | **Rose Smith** | **3/1/17 12:31 PM** |

Measurements of $CH_4$

| **Page 21: [37] Deleted** | **Rose Smith** | **3/1/17 12:31 PM** |

Measurements of $CH_4$

| **Page 21: [37] Deleted** | **Rose Smith** | **3/1/17 12:31 PM** |

Measurements of $CH_4$

| **Page 21: [37] Deleted** | **Rose Smith** | **3/1/17 12:31 PM** |

Measurements of $CH_4$

| **Page 21: [37] Deleted** | **Rose Smith** | **3/1/17 12:31 PM** |

Measurements of $CH_4$

| **Page 21: [37] Deleted** | **Rose Smith** | **3/1/17 12:31 PM** |

Measurements of $CH_4$

| **Page 22: [38] Deleted** | **Rose Smith** | **3/9/17 4:41 PM** |

UiS a,

Methane concentrations were consistent with prior studies, showing that streams are commonly super-saturated with CH4 (e.g. Jones and Mulholland 1998; Wilcock and Sorrel 2008; Baulch et al. 2011; Werner et al. 2012).   In contrast with IPCC methodology (Ciais et al. 2013), there is growing evidence that human impacts on watersheds influence CH4 emissions from streams (Kaushal et al. 2014b, Crawford and Stanley 2015; Stanley et al. 2015). Prior studies have found that CH 4 production tends to be elevated in streams with fine benthic sediments, an influx of organic matter, or significant wetland drainage (Dinsmore et al. 2009; Dawson et al. 2002; Baulch et al. 2011). Significant negative relationships between TDN and CH4

were detected in this study, and elevated CH4 concentrations in streams draining intact floodplains and/or stormwater management wetlands.
* * *
**Page 22: [39] Deleted**        **Rose Smith**        **3/1/17 12:44 PM**

An increasing number of scientific studies have compiled GHG budgets of anthropogenic and ecological emissions across cities (e.g., Brady and Fath, 2008; Hoornweg et al. 2011; Weissert et al. 2014). Understanding both the anthropogenic and ecological components of a regional GHG budget is crucial for setting GHG targets and managing ecosystem services (Bellucci et al. 2012). The role of human activities on GHG emissions from agriculturally impacted waterways is well recognized (Ciais et al. 2013; Nevison 2000). However, further studies examining the magnitude and variations in GHG emissions along the urban watershed continuum, which explicitly includes flowpaths from engineered infrastructure to streams and rivers (e.g. Kaushal and Belt 2012), are necessary. As cities and populations continue to expand globally, GHG emissions from wastewater are likely to rise. A greater understanding of the interplay between urban water infrastructure and biogeochemical processes is necessary to mitigate negative consequences of N2O, CH4, and CO2.
* * *
**Page 22: [40] Deleted**        **Rose Smith**        **3/9/17 4:43 PM**

M. Pennino provided feedback on multiple versions of the manuscript
* * *
**Page 36: [41] Formatted Table**        **Rose Smith**        **2/28/17 2:49 PM**

Formatted Table
* * *
**Page 36: [42] Formatted**        **Rose Smith**        **2/22/17 10:32 PM**

Left, None, Space Before: 0 pt, Don't keep with next, Don't keep lines together
* * *
**Page 36: [43] Formatted**        **Rose Smith**        **2/22/17 10:32 PM**

Left, Tabs:Not at 3.15" + 6.3"
* * *
**Page 36: [44] Formatted**        **Rose Smith**        **2/22/17 10:32 PM**

Left, Tabs:Not at 3.15" + 6.3"
* * *
**Page 40: [45] Deleted**        **Rose Smith**        **3/1/17 3:51 PM**

| Infrastructure typology | Site | Gas | Average | s.e. | n |
|---|---|---|---|---|---|
| Stream Burial | DRAL | | 0.50 | 0.18 | 17 |
| In-line SWM | DRGG | | 0.64 | 0.10 | 15 |
| In-line SWM | DRKV | $CO_2$ | 39.55 | 8.72 | 19 |
| Floodplain Preservation | RRRB | | 2.04 | 0.48 | 17 |
| Stream Burial | DRAL | | 0.18 | 0.09 | 17 |
| In-line SWM | DRGG | | 1.53 | 0.35 | 15 |
| In-line SWM | DRKV | $CH_4$ | 29.41 | 21.69 | 19 |
| Floodplain Preservation | RRRB | | 3.53 | 1.13 | 17 |
| Stream Burial | DRAL | $N_2O$ | 0.57 | 0.22 | 17 |

| | | | | | |
|---|---|---|---|---|---|
| In-line SWM | DRGG | | 0.53 | 0.09 | 15 |
| In-line SWM | DRKV | | 44.07 | 9.79 | 19 |
| Floodplain Preservation | RRRB | | 1.01 | 0.23 | 17 |
| Stream Burial | DRAL | | 2.29 | 0.75 | 17 |
| In-line SWM | DRGG | $K_{20}$ | 1.67 | 0.33 | 15 |
| In-line SWM | DRKV | | 57.56 | 9.63 | 19 |
| Floodplain Preservation | RRRB | | 9.36 | 1.67 | 17 |

| Page 45: [46] Formatted | Rose Smith | 2/16/17 9:41 PM |
|---|---|---|

Font:Not Italic

| Page 45: [47] Deleted | Rose Smith | 3/1/17 4:00 PM |
|---|---|---|

**anaerobic**

| Page 46: [48] Deleted | Rose Smith | 2/28/17 2:44 PM |
|---|---|---|

**Panels A:D show longitudinal variability in**